# Introduction to quantum error correction and fault tolerance

**Steven M. Girvin⋆**

Yale Quantum Institute, 17 Hillhouse Ave., PO Box 208334, New Haven, CT 06520-8263 USA

⋆ steven.girvin@yale.edu

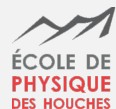

*Part of the Quantum Information Machines*
*Session 113 of the Les Houches School, July 2019*
*published in the Les Houches Lecture Notes Series*

## Abstract

These lecture notes from the 2019 Les Houches Summer School on 'Quantum Information Machines' are intended to provide an introduction to classical and quantum error correction with bits and qubits, and with continuous variable systems (harmonic oscillators). The focus on the latter will be on practical examples that can be realized today or in the near future with a modular architecture based on superconducting electrical circuits and microwave photons. The goal and vision is 'hardware-efficient' quantum error correction that does not require exponentially large hardware overhead in order to achieve practical and useful levels of fault tolerance and circuit depth.

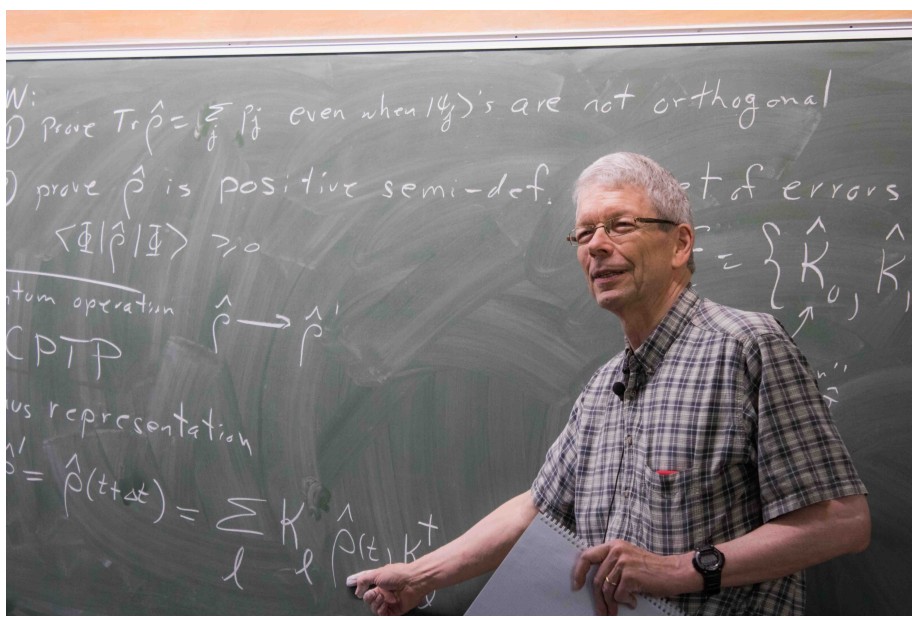

**Link to Video Lectures**

The website for the 2019 Les Houches School on Quantum Information Machines contains links to slides and videos for the various lectures:

**https://physinfo.fr/houches2019/program.html**

# 1 Introduction: The Grand Challenge

The last 20 years have seen spectacular experimental progress in our ability to create, control and measure the quantum states of superconducting 'artificial atoms' (qubits) and microwave photons stored in resonators. In addition to being a novel testbed for studying strong-coupling quantum electrodynamics in a radically new regime, 'circuit QED,' defines a fundamental architecture for the creation of all-electronic quantum computers based on integrated circuits with semiconductors replaced by superconductors. The artificial atoms are based on the Josephson tunnel junction and their relatively large size ($\sim$mm) means that they couple extremely strongly to individual microwave photons. This strong coupling yields very powerful state-manipulation and measurement capabilities, including the ability to create extremely large ($>$ 100 photon) 'cat' states and easily measure novel quantities such as the photon number parity. These new capabilities are enabling new schemes for 'continuous variable' quantum error correction based on encoding quantum information in superpositions of different Fock states of microwave photons.

The grand challenge facing us as we attempt to build large-scale quantum machines is fault tolerance. How do we build a nearly perfect machine from a large collection of imperfect parts? This question was addressed in the classical domain by von Neumann beginning after the second world war [1], in a series of lectures at Caltech in 1952 that were published in 1956 [2] and in the Silliman Lecture at Yale which he was unable to deliver but whose manuscript was published posthumously [3]. In addition to thinking about the crude and unreliable vacuum tube computers of his day, he was fascinated by the ability of the complex network of neurons in the brain to reliably compute. Claude Shannon, whose master's thesis first proved that circuits of switches and relays could perform arbitrary Boolean logic operations [4], was also keenly interested in this problem [5].

Von Neumann showed (not quite rigorously) that a Boolean function that can be computed by a network of $L$ reliable gates can be reliably (i.e., with high probability) computed by a network of $\mathcal{O}(L \log L)$ unreliable gates. This result was made rigorous by Dobrushin and Ortyukov [6]. Useful works to consult, to learn more about this field, include [7–10]. The modern perspective connects the problem of reliable computation with unreliable devices to Shannon's information theory [11] which describes how to reliably communicate over a noisy channel. As illustrated in Fig. 1, in Shannon's information theory, only the communication channel is considered unreliable and the encoding at the input and decoding at the output is taken to be perfect. A reliable computation can be performed by unreliable circuits by using circuit modules that operate on codewords designed for the Shannon communication problem and frequently checking them. The trick is to find ways to distinguish between differences in the output and input of the module that are intentional (i.e., due to the module correctly computing the intended function of the input) or are erroneous [10].

In addition to this key connection to information theory, there is also an important connection to control theory, illustrated in Fig. 2. A quantum computer is a dynamical system we are attempting to control despite noise and errors that occur continuously in time. Classical control theory, founded by Norbert Wiener, deals with a system (traditionally known as the 'plant' and which might actually represent a car manufacturing or chemical production plant) subject to errors. As illustrated in Fig. 3, sensors take continuous measurements of the state of the plant and a controller analyzes this information and uses it to provide (via an 'actuator') feedback to the plant to keep it running stably and reliably. Robust control systems are able to deal with the fact that the sensors, controller and actuator units can also be made of unreliable parts. We will find this a useful point of view but will have to deal with a number of subtleties in thinking about the control of quantum systems since we know that measurements on a quantum state disturb the state through measurement 'back action' (state collapse).

| message bits | ideal | noisy channel | ideal | recovered message |
| ancillae bits | encoder | | decoder | error information |

Figure 1: Shannon communication problem. The sender's message plus some redundant information about that message are encoded by an ideal encoder and transmitted over a noisy channel which introduces errors. The received erroneous message is then processed by an ideal decoder which outputs the recovered message plus ancillary information about the particular errors that the channel randomly introduced. In this model, only the noisy channel is imperfect. The computations needed for encoding and decoding are assumed to be carried out by error-free hardware.

The field of quantum machines has made tremendous experimental strides in the last two decades and we are just entering the era where the hardware is now reliable enough to begin to carry out quantum error correction. As we begin to scale up existing small quantum processors to ever larger numbers of components, it is essential that we learn how to do fault-tolerant design and control of these novel and powerful new machines. This is the grand challenge that must be met if the second quantum revolution is to succeed.

These lecture notes will discuss general issues in classical and quantum error correction and fault tolerance with a focus on quantum information processing with superconducting qubits and microwave photons using the 'circuit QED' architecture.

## 2 Classical Error Correction and Fault Tolerance

A significant benefit of the representation of information as discrete bits (with 0 and 1 corresponding to a voltage for example) is that one can ignore small noise voltages. That is, $V = 0.99$ volts can be safely assumed to represent 1 and not 0. Modern electronic logic chips deal with noise on input signals by processing them through a circuit whose voltage input output relation is shown schematically in Fig. (4). It is clear from this that input voltages less than 0.5 are clamped near zero on the output and input voltages larger than 0.5 are clamped near 1 on the output. This effectively erases errors from small voltage deviations and makes digital devices more robust than analog devices.

Even relatively 'small' digital devices today contain a vast number of components. For example, a typical cell phone contains more than 5 billion transistors in its processor chips plus even more components in its memory. Even if the individual components are highly reliable, they are never perfect. This means that such a complex system will almost certainly fail to operate correctly unless it is constructed using the principles of *fault tolerant design*. Roughly speaking we must design our systems to be tolerant of two kinds of faults–memory errors and gate operation errors. We will begin our analysis with memory errors.

To overcome the deleterious effects of more severe electrical and magnetic noise, cosmic rays, transistor failures and other hazards, modern digital computer memories and information storage devices rely on error correcting codes to store and correctly retrieve vast quantities of data. As illustrated in Fig. 5, it turns out that there is a deep connection between memory error correction and Claude Shannon's work [11] on using redundant encoding of messages for reliable communication of information over a noisy channel. In the communication problem, Alice sends a message to Bob and the message is corrupted by noise and loss during the

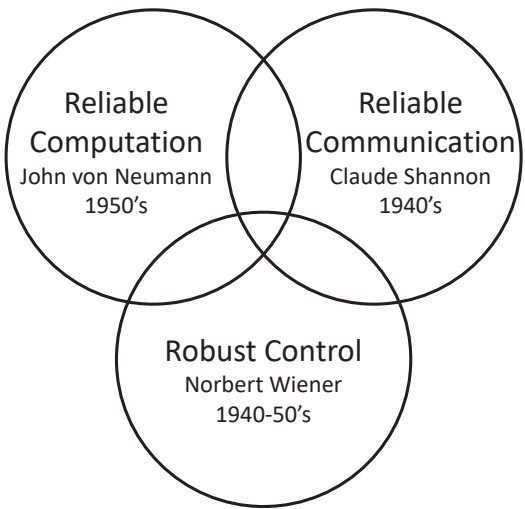

Figure 2: Three foundational topics in the study of complex classical systems. Information theory was developed to study reliable communication over a noisy channel using codes containing redundant information, but assuming reliable computation for the encoding and decoding of that information. Reliable computation requires understanding how to use redundant hardware to ensure proper functioning of the computer. Robust control theory studies the use of feedback to stabilize the operation of a complex system, even when the sensors, actuators and feedback controller are imperfect.

transmission through space. We can view the memory problem as Alice sending herself a message through time. She wants to store information and retrieve it later, but it may become corrupted during the passage of time.

Classical error correction works by introducing extra bits which provide redundant encoding of the information. A familiar everyday example is the phonetic alphabet used by air traffic control centers and pilots in their communications. 'Hold short of taxiway Tango Delta Bravo,' is a lot easier to understand in a noisy environment than 'Hold short of taxiway TDB.' The 'distance' in 'acoustic signal space' between 'T,' 'D,' and 'B' is much smaller than for the lengthier codewords 'Tango,' 'Delta,' and 'Bravo.' The latter are therefore more readily distinguished and correctly decoded when we hear them. More formally, classical error correction proceeds by measuring the bits and comparing them to the redundant information in the auxiliary bits.

All classical (and quantum) error correction codes are based on the assumption that the hardware is good enough that errors are rare. The goal is to make them even rarer. For classical bits there is only one kind of error, namely the bit flip which maps 0 to 1 or vice versa. This error channel is illustrated in Fig. 6.

As indicated in Fig. 6, in general the probability $p_{10}$ or an error when transmitting a 1 can be different than the error probability for transmitting a 0, $p_{01}$. For example, in transmitting a signal over an optical fiber, we might have a code in which the presence of a photon indicates a 1 and the absence of a photon encodes a 0. Attenuation (optical absorption) in the fiber can lead to photons being lost which contributes to $p_{10}$. The physical processes that contribute to $p_{01}$ are different and might include stray photons entering the fiber from the environmental so-called 'dark counts' in the photon detectors at the receiving end of the channel. Because the physical mechanisms leading to the two errors are different, in general the error rates will be different when transmitting 0's and 1's (with this particular encoding and physical transmission channel). If for example, we can only lose photons and never gain them, then 0

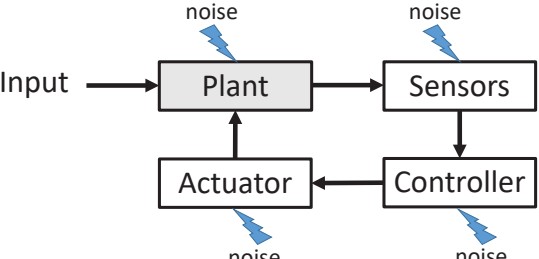

Figure 3: Classical control theory describes a system ('the plant') whose state is controlled by a feedback system consisting of sensors, a controller that decides how to respond to the sensor information by sending signals to an actuator which adjusts parameters in the plant to maintain it in the desired state despite perturbations. Robust control theory deals with the problem that the sensors, controller and actuator can be imperfect.

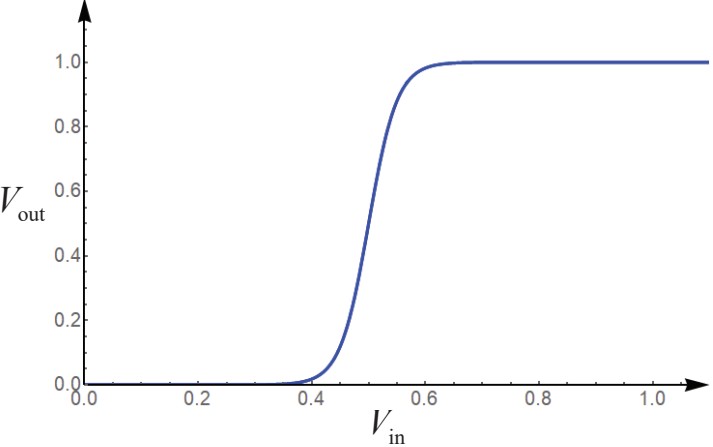

Figure 4: Input-Output relation for a voltage clamping circuit that eliminates noise in classical logic devices.

is always transmitted correctly, but 1 is sometimes received as a 0. This a highly asymmetric noise channel. Here we will assume for simplicity the binary symmetric channel (BSC) which is characterized by a single error probability $\epsilon = p_{10} = p_{01}$ (typically with $\epsilon \ll 1$), and has the important property that error occurrences are independent (uncorrelated) among different bits.

Another important error model is the *erasure channel*. In this case one knows which bits are erroneous, but not their correct values. For example, suppose that we are transmitting information via the polarization of individual photons: vertical polarization could represent 0 and horizontal polarization could represent 1. By passing the received photons through a polarizing beam splitter, vertically polarized photons will land on one detector and horizontally polarized photons will land on a different detector. If a message is transmitted as a sequence of vertically and horizontally polarized photons and one of those photons is absorbed by the transmitting medium (e.g., the optical fiber), then the receiver will detect neither polarization (neither detector will click) and we will know that particular photon has been lost. Thus photon loss in this particular code is an erasure error. Erasure errors are vastly easier to correct because we are given extra information–namely which bits are erroneous. This is in contrast to

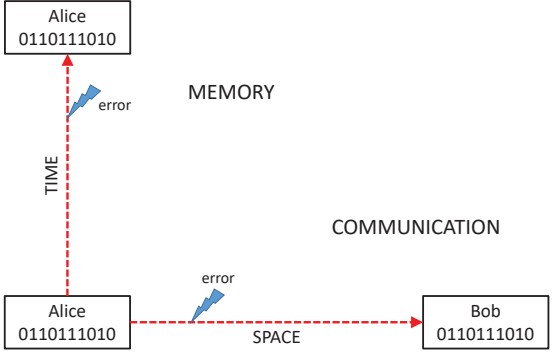

Figure 5: In the communication problem, Alice sends Bob a message over a noisy channel which may corrupt some of the bits. In the memory problem, Alice sends herself a message over time by storing it in a memory that may become corrupted. From the point of view of designing error correction codes to make the messages robust, communication and storage are exactly the same problem.

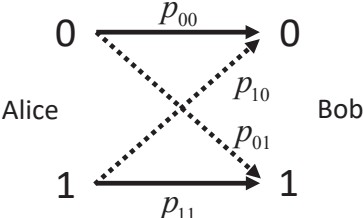

Figure 6: In classical communication there is only one type of error, namely the bit flip. If Alice sends a 0 to Bob, it correctly arrives as a zero with probability $p_{00}$ and incorrectly arrives as a 1 with probability $p_{01} = 1 - p_{00}$. If Alice sends a 1 to Bob it correctly arrives as a 1 with probability $p_{11}$ and incorrectly arrives as a 0 with probability $p_{10} = 1 - p_{11}$. The case $p_{10} = p_{01} = \epsilon$ is known in classical information theory as the 'binary symmetric channel' (BSC).

a physical encoding in which 1 is represented by the presence of a photon and 0 is represented by the absence of a photon. In this case, loss of a photon means that a 1 is incorrectly received as a 0 rather than as an erasure. This means that the error correction code has to be more sophisticated and be able to correct errors at unknown locations.

The most general definition of an erasure error is that it is an unknown error at a known location (i.e. on a known physical bit). Knowing which physical bit has gone bad is very useful information and this makes correcting such errors easier [12].

## 2.1 Error-detection Parity-check Code

Before considering codes which can correct errors, let us begin with a simple example of a code that can detect, but not correct, a single error. Imagine that the data is sent in blocks of $m$ bits. For example $m = 8$ would correspond to one byte of data. Now let Alice append an ancilla bit. This redundant bit is called the 'parity check' bit and she arranges its value to be such that the total parity of the $m + 1$ bits is (say) even. Thus if the number of 1's in the first $m$ bits is even, she makes the parity check bit have the value 0. Conversely if the number of 1's in the first $m$ bits is odd, she gives the parity check bit the value 1. Alice sends the $m + 1$ bits through a

noisy channel to Bob. It is possible that one or more errors has occurred, including perhaps in the parity check bit itself. Bob can then compute the parity of the received bit string. If he finds that the parity is even, he knows that the number of errors is even (and possibly 0). If he knows that errors are sufficiently rare that he can neglect the possibility of more than one error, then he can assume that with high probability there was no error. (If there were an even number of bit flips, he would be fooled, but the probability of 2 or more errors is very small.) If Bob sees that the parity is odd, he knows that there was an error (or more precisely an odd number of errors) and he can reject the information and ask Alice to resend it.[1] For the BSC, the probability of no errors occurring in a string of $m + 1$ bits is

$$P_0 = (1 - \epsilon)^{m+1} \sim 1 - (m + 1)\epsilon \,, \tag{1}$$

with the latter approximation being valid for $(m+1)\epsilon \ll 1$. The probability of a single bit flip error is

$$P_1 = {}_{m+1}C_1(1 - \epsilon)^m(\epsilon)^1 = (m + 1)\epsilon(1 - \epsilon)^m \sim (m + 1)\epsilon \,, \tag{2}$$

where ${}_N C_k$ is the binomial coefficient or combinatorial factor '$N$ choose $k$'

$$_N C_k = \frac{N!}{(N - k)!\, k!} \,. \tag{3}$$

Failure to detect an error occurs only if there are an even (but non-zero) number of errors, The most likely case for failure is the case with the smallest even number of errors, namely 2. The probability for this to occur is

$$P_2 = {}_{m+1}C_2(1 - \epsilon)^{m-1}(\epsilon)^2 = \frac{(m+1)m}{2}\epsilon^2(1 - \epsilon)^{m-1} \sim \frac{m(m+1)}{2}\epsilon^2 \,. \tag{4}$$

Assuming $m\epsilon/2 \ll 1$, we have $P_2 \ll P_1$ and so Bob can correctly detect errors with high probability. The actual failure probability is slightly larger than $P_2$ because $P_4, P_6, P_8, \ldots$ all contribute but these probabilities are very small.

## 2.2 Error Correction Parity Check Codes: Part 1

To actually correct the error, Bob would have to know the location of the error, but this information is not available to him in the simple parity check code described above. Before exploring error correction parity check codes in more detail, let us warm up with study of a very simple code, the repetition code.

### 2.2.1 Repetition Code

It is useful to review the very simple (and not very efficient) repetition code which is capable of not just detecting, but actually correcting a fixed number of errors. The repetition code is perhaps the easiest classical error correction code to understand and involves sending multiple copies of the data and decoding using majority rule. Versions of this code can correct multiple errors, but the code is very inefficient since it encodes only 1 error-correctable logical data bit in a string of $N$ physical bits and thus has code rate $r = 1/N$ which asymptotically goes to zero for large $N$.

To understand the repetition code, let us consider a register of $N = 2m + 1$ physical bits which we arrange to be in one of two 'logical codewords' (bit strings representing logical 0

---

[1]Notice that if the single error is an erasure error, then we can recover the correct bit value from the parity information in the remaining bits. This is a simple example of why erasure errors are easier to deal with. This code that can detect single errors at unknown locations, but can correct single erasure errors.

and logical 1)

$$0_{\mathrm{L}} = (000\ldots00), \tag{5}$$
$$1_{\mathrm{L}} = (111\ldots11). \tag{6}$$

We choose $N$ odd so that we can do majority voting in the decoding step. We can encode this logical bit by having a single original data bit and copying its value into the $2m$ additional ancilla bits in the register. Any state of the register not equal to one of the two logical codewords, is by definition, an error state.

The easiest repetition code to analyze is the smallest one which has $2m + 1 = 3$ physical bits: $M = 1$ original data bits and $R = 2$ ancilla bits. Assuming that the total number of errors in the register is limited to being either 0 or 1, there will be $M + R + 1 = 4$ possible error states (with the $+1$ accounting for the possibility of zero errors) that can be encoded in the $2^R = 4$ ancilla states, still leaving the original information available after decoding in the $2^M = 2$ states of the data bit. (It is useful at this point to review Fig. 1 to see that the encoding has to have enough degrees of freedom to allow the received version of the message to store both the intended message and the information on which errors occurred in the noisy channel.)

Suppose now that one of the three physical bits suffers an error. By examining the state of each bit it is a simple matter to identify the bit which has flipped and is not in agreement with the 'majority.' We then simply flip the minority bit so that it again agrees with the majority. This procedure succeeds if the number of errors is zero or one, but it fails if there is more than one error (because the majority is now wrong). Of course since we have replaced one imperfect bit with three imperfect bits, this means that the probability of an error occurring has increased considerably. For three bits the probability $P_n$ of $n$ errors is given by

$$P_0 = (1 - \epsilon)^3, \tag{7}$$
$$P_1 = 3\epsilon(1 - \epsilon)^2, \tag{8}$$
$$P_2 = 3\epsilon^2(1 - \epsilon), \tag{9}$$
$$P_3 = \epsilon^3. \tag{10}$$

Every error correction code begins by adding ancilla bits. This means that the probability of errors actually rises. For example, we see from the above that the probability of at least one error is

$$1 - P_0 = 1 - (1 - \epsilon)^3 \sim 3\epsilon, \tag{11}$$

where the approximation in the last term is valid for small $\epsilon$. We see that in this limit the error probability triples. Our error correction circuit must overcome this enhanced error probability if the error probability for the logical qubit is to be lower than that for a single physical qubit.

Because our error correction code only fails for two or more physical bit errors the error probability for our logical qubit is

$$\epsilon_{\mathrm{logical}} = P_2 + P_3 = 3\epsilon^2 - 2\epsilon^3. \tag{12}$$

As can be seen in Fig. 7, if $\epsilon < \epsilon^* = 1/2$, then the error correction scheme reduces the error rate (instead of making it worse). If for example $\epsilon = 10^{-6}$, then $\epsilon_{\mathrm{logical}} \sim 3 \times 10^{-12}$. Thus the lower the raw error rate, the greater the improvement. Note however that even at this low error rate, a petabyte ($8 \times 10^{15}$ bit) storage system would have on average 24,000 errors. Futhermore, one would have to buy three petabytes of storage since 2/3 of the disk would be taken up with ancilla bits!

**Box 1. Break-Even Point for Error Correction** *The particular value $\epsilon = \epsilon^*$ of the physical bit error probability at which $\epsilon_{\text{logical}} = \epsilon$ is known as the break-even point for error correction. For $\epsilon > \epsilon^*$, error correction makes things worse, while for $\epsilon < \epsilon^*$ the error correction increases the lifetime of the quantum information. (Specifically in the case described above, it is the break-even point for memory operation. The break-even point for gate operations is a separate matter.) In general, the physical qubits making up the logical bit have inhomogeneous properties. In this case, the break-even point is conservatively defined as that point at which the error probability of the logical bit is lower than that of the best physical bit comprising it.*

*A vivid example of the break-even point can be found in the first transatlantic solo airplane flight in 1927 by Charles Lindbergh. Many people thought that Lindbergh was foolish for using a single-engine airplane. However, he correctly understood that having two engines doubled the probability of engine failure in flight. He further understood that the 1927 engine technology was such that, unlike today, a twin-engine plane with one engine out could barely fly and would not be able to reach a safe landing place (such as Keflavik, Iceland). In short, the aircraft engine repetition code did not exceed break even in 1927.*

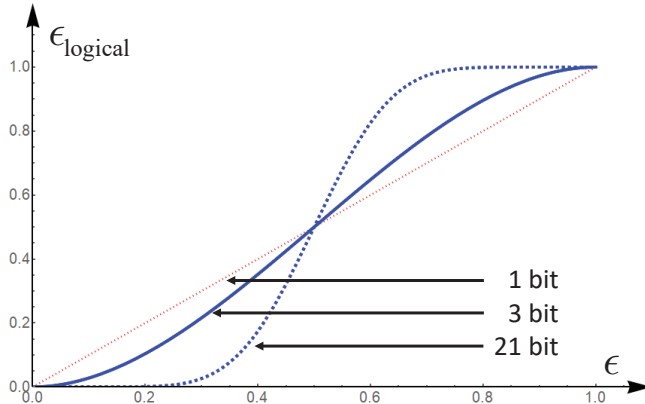

Figure 7: Logical error probability $\epsilon_{\text{logical}}$ for the classical repetition code vs. physical qubit error probability $\epsilon$. Solid line is the $2m + 1 = 3$ bit code. Dashed line is the $2m + 1 = 21$ bit code. The straight line is the single bit error probability. We see that the repetition codes have a break-even point of $\epsilon^* = \frac{1}{2}$. Below that value the logical error rate is lower than the physical error rate for a single bit even though the logical bits contain more than one physical bit. Notice that the larger the repetition code, the more sharp is the transition near the threshold and the greater is the error correction gain at low physical error rates.

The calculation above was for the case $2m + 1 = 3$ bit code. For general $m$, the repetition code fails only if there are more than $m$ errors (at which point the errors are in the majority). Hence for physical bit error probability $\epsilon$, the logical bit error probability is

$$\epsilon_{\text{logical}} = \sum_{n=m+1}^{2m+1} {}_{2m+1}C_n \, \epsilon^n (1-\epsilon)^{2m+1-n}, \tag{13}$$

where the summation is over the number of errors $n$. Keeping the full expression without approximation, we see that the error threshold is $\epsilon^* = \frac{1}{2}$ independent of the number of bits

$2m+1$ in the repetition code (visible graphically in Fig. 7 for the particular cases of $m = 1$ and $m = 10$). However the error correction 'gain'

$$G = \frac{\epsilon}{\epsilon_{\text{logical}}} \tag{14}$$

is greater and greater for larger $m$, especially for $\epsilon \ll 1$ because the code can tolerate up to $m$ errors and hence

$$\epsilon_{\text{logical}} \sim \frac{(2m+1)!}{m!(m+1)!} \epsilon^{m+1}, \tag{15}$$

in the limit of small $\epsilon$.

Notice that for large codewords of length $N = 2m + 1$, the number of errors $n$ (assuming as usual that they are uncorrelated) is also large on average:

$$\bar{n} = \langle n \rangle = \epsilon N. \tag{16}$$

If $\bar{n}$ is large then the probability distribution $P(n)$ for the number of errors is approximately a Gaussian with mean $\bar{n}$ and standard deviation $\sigma = \sqrt{\bar{n}}$. [See Exercise 1.] The code can correct up to $m$ errors which means that failure requires a statistical fluctuation in $n$ above the mean of $\Delta n \geq (m+1) - \bar{n}$. This is a fluctuation corresponding to a very large number of standard deviations above the mean

$$\frac{\Delta n}{\sigma} \sim \sqrt{\frac{m}{2\epsilon}}, \tag{17}$$

an extremely unlikely event[2] for large $m$.

---

**Exercise 1**

For large $m$, Fig. 7 shows a relatively sharp transition from low logical error rate to 50% as the physical error rate approaches 50% from below. Use the central limit theorem to estimate how the width of this transition scales with $m$ in the limit of large $m$.

---

We see that in the limit of large codeword length $N = 2m + 1$, the code failure probability can be made arbitrarily small. However the repetition code only stores one error-correctable logical bit in $N$ physical bits so the rate $r = 1/N$ vanishes (polynomially) as the error rate decreases exponentially with $N$. This simple code is very inefficient.

## 2.3 Error Correcting Parity Check Codes: Part 2

We can compute the minimum degree of redundancy needed to detect and correct a single bit flip in a string of $N$ bits by the following argument. Suppose that we have $M$ data bits we wish to protect using $R$ redundant bits. There are $N = M + R$ possible single-bit errors (because the error could be in one of the redundant bits, not just in the data bits). Including the case of zero errors, there are a total of $M + R + 1$ error states. If we are to be able to correct the error, we must know the error state and hence a necessary condition is that $R$ be large enough to encode the error state. Thus we must have

$$2^R \geq M + R + 1. \tag{18}$$

---

[2]Strictly speaking our approximation of the probability distribution as a Gaussian breaks down in the far tails of the distribution (where it becomes exponential rather than Gaussian), but this simple derivation gets across the main idea that the failure probability is extremely small, though not actually as small as the Gaussian approximation would suggest.

We previously discussed the three-bit repetition code which has $M = 1, R = 2$ for which the inequality in eqn (18) is satisfied as an equality. However we saw that for highly redundant repetition codes with $R \gg 1$, the LHS of eqn (18) is exponentially larger than the RHS, and yet the codes only hold a single logical bit ($M = 1$). For example with $R = 10$ we could encode 1024 error states. Hence the number of data bits that we should be able to protect against single errors is $M = 1013$, much greater than 1. This does not prove sufficiency, i.e., that there exists an encoding/decoding circuit that will do this, but it turns out there are parity check codes which will in fact work. Notice that such a code would be very efficient: Out of 1024 bits we send, 1013 of them are data bits and only 11 of them are ancilla bits. This efficiency is quantified in the *code rate*

$$r = \frac{\text{data bits}}{\text{total bits}} = \frac{1013}{1024} \approx 0.989 \,. \tag{19}$$

Because the number of redundant bits grows only logarithmically with the number of bits $R \sim \log_2[M + R + 1] \sim \log_2[N]$, the code rate

$$r = \frac{N - R}{N} \sim 1 - \frac{\log_2 N}{N} \tag{20}$$

asymptotically approaches unity as $N$ increases. Notice however that this class of codes only corrects a single error and for them to work with high probability, we would need $\epsilon[M+R] \ll 1$ and thus the error rate would have to fall like $\epsilon \sim \Lambda/M$ as $M$ increases where $\Lambda \ll 1$ is a constant.

For a given fixed error rate $\epsilon$, the average number of errors in an encoded message of length $N$ is of course $\epsilon N$ which becomes very large as $N$ grows. Thus to deal with this situation, we would need to have codes that for large $N$ can correct a large number of errors with very high success probability. We saw above that the simple repetition code can easily achieve arbitrarily high success probability but only at the cost of asymptotically zero rate (because they encode only a single logical bit). A remarkable 'noisy channel theorem' from Claude Shannon gives a (non-constructive) proof that codes exist which have arbitrarily low failure probability and have finite (i.e., not asymptotically zero) rate even when the physical error rate has a fixed non-zero value.

To work out the redundancy $R$ required when we have a fixed error rate $\epsilon$ we need to understand how much information $S$ is needed to specify the locations of the $N\epsilon$ errors that will (typically occur). This is very simply obtained from the Shannon entropy produced by probability distribution associated with the error channel

$$S = -N[\epsilon \log_2 \epsilon + (1 - \epsilon) \log_2 (1 - \epsilon)] \approx N\epsilon \log_2 \left(\frac{2}{\epsilon}\right), \tag{21}$$

where the last approximate equality is valid for sufficiently small $\epsilon$. The number of parity check bits $R$ needed to fully identify a typical set of errors is $\lceil S \rceil$. To provide (with high probability) security against upward fluctuations in the number of errors above the mean, one only needs to have $R = \lceil S + \omega \rceil$, where $\omega$ is subextensive (i.e., a sub-linear function of $N$). To gain some intuition for why $\omega$ is subextensive, imagine that there is a very rare large fluctuation in the number of errors above the mean $\bar{n} = \epsilon N$ by (say) $10\sqrt{\bar{n}}$ (i.e., 10 standard deviations). A simple and strict upper bound on the increase in the entropy of the error distribution (and therefore the size of $\omega$ needed to have a low failure probability) is $10\sqrt{\bar{n}} \log_2 N$. (This is obtained by ignoring the fact that there can be at most one error per bit and allowing the permutation of the locations of the errors to be counted as new error states even though they are not.) Hence $\omega/N$ becomes negligible in the asymptotic large $N$ limit. Thus, based

purely on this information theoretic argument, it should be possible to achieve a code rate (for asymptotically large $N$) approaching

$$r = \frac{M}{M+R} = 1 - \frac{R}{N} \approx 1 - \epsilon \log_2\left(\frac{2}{\epsilon}\right). \tag{22}$$

Of course this argument is not a proof that codes can be constructed which achieve this bound, but it does show that it is impossible for codes to exceed this bound because the information obtained from measuring the parity check bits must equal or exceed the entropy added to the message by the error channel. The point of this calculation is that it shows that satisfying this bound does not require a large fraction of the message to be parity check bits (i.e., we only need a low density of parity check bits within the code) if the error rate is low but finite and one can successfully encode and decode very long bit strings (of length $N = M + R$). Hamming was able to prove that such codes exist by showing that choosing codes with random arrangements of parity checks (random Tanner graphs in the language which will be introduced shortly below and illustrated in Fig. 8) almost always produces codes that approach this limit. (Presumably such random codes are not practical in terms of the computational cost of encoding and decoding them.)

There is a vast literature on practical parity-check codes that have non-zero rate and computationally tractable encoding and decoding protocols. We will not pursue these here, but will give one prototypical example of a simple parity-check code, the Hamming code, that is distinct from the simple repetition code.

## 2.4 The Hamming Code

Low-density parity-check codes (LDPCs) are used in classical communication to redundantly encode messages so that errors in transmission can be corrected by the receiver. They are also being explored theoretically for quantum error correction. To understand how these codes work it is useful to take advantage of a curious connection between these codes and the protocols that can be used to pool testing samples for the novel corona virus SARS COV-2 in order to reduce testing costs and increase speeds.

### 2.4.1 SARS COV-2 Sample Pooling

If the positivity rate is low, most tests come back negative. (For the moment we will assume tests are 100% reliable with no false negatives or false positives.) One can reduce the number of tests needed to find positive cases by dividing samples into a number of different portions and pooling them with other samples. This is illustrated in the so-called 'Tanner graph' in Fig. 8 showing how 7 samples $(S_1, \ldots, S_7)$ are combined into 3 pools $(P_1, P_2, P_3)$. The pooling is arranged so that each sample produces a unique pattern of positive results in the pools. If all of the samples are negative then all of the pools will be negative. If, for example, the first sample is positive, then only $P_1$ will be positive, while if the seventh sample is positive, then $P_1, P_2, P_3$ will all be positive. For simplicity, we assume that the sample positivity rate $\epsilon \ll 1/7$, so that the probability of getting two or more positive samples can be neglected to first approximation. In this case there are eight possible sample states: none are positive or any one of the seven are positive. These eight states can be encoded into the three 'pool bits'. A pool bit value of 0 means that no samples in the pool are positive. A pool bit value of 1 means that (at least) one of the samples in the pool is positive.

The connectivity of the Tanner graph can be conveniently represented by a rectangular matrix

$$H = \begin{pmatrix} 1 & 0 & 1 & 0 & 1 & 0 & 1 \\ 0 & 1 & 1 & 0 & 0 & 1 & 1 \\ 0 & 0 & 0 & 1 & 1 & 1 & 1 \end{pmatrix}. \tag{23}$$

$H_{jk}$ is the element of the matrix in row $j$, column $k$. $H_{jk} = 1$ means that sample $k$ is present in pool $j$. Each of the seven columns in $H$ represent a possible output bit string. For example if sample $k$ is the one which is positive then the output bit string is

$$\begin{pmatrix} P_1 \\ P_2 \\ P_3 \end{pmatrix} = \begin{pmatrix} H_{1k} \\ H_{2k} \\ H_{3k} \end{pmatrix}, \tag{24}$$

when the pools are measured. Assuming that at most one sample is positive, the input/output relation can be summarized by the matrix equation

$$\begin{pmatrix} P_1 \\ P_2 \\ P_3 \end{pmatrix} = HS = \begin{pmatrix} 1 & 0 & 1 & 0 & 1 & 0 & 1 \\ 0 & 1 & 1 & 0 & 0 & 1 & 1 \\ 0 & 0 & 0 & 1 & 1 & 1 & 1 \end{pmatrix} \begin{pmatrix} S_1 \\ S_2 \\ S_3 \\ S_4 \\ S_5 \\ S_6 \\ S_7 \end{pmatrix}, \tag{25}$$

where

$$S = \begin{pmatrix} S_1 \\ S_2 \\ S_3 \\ S_4 \\ S_5 \\ S_6 \\ S_7 \end{pmatrix} \tag{26}$$

is a vector describing the state of the samples. If none of the samples are positive, all of the entries in $S$ are 0. If sample $k$ is the positive, then $S_k = 1$ and the remaining entries are 0.

Notice that each row in $H$ contains four 1's indicating that every pool contains a mixture of four samples (as shown in the Tanner graph). Also notice that each column of $H$ is unique because the $k$th column is the binary representation for the number $k$. This uniqueness is what makes decoding possible. To understand this consider the following examples. The pool variable $P_j$ is zero if none of the four samples it contains is positive. The pool variable is one if one of the samples it contains is positive. (We assume here that there is at most one sample out of the seven that is positive.) Thus we can treat the $P_j$ as bits and the bit string $(P_3 P_2 P_1)$ as the binary representation of the integer number $k = P_1 + 2P_2 + 4P_3$ with $0 \le k \le 7$. As an example, examination of the Tanner graph in Fig. 8 shows that if (only) sample 5 is positive ($S_5 = 1$) then the pool 'bit string' is $(P_3 P_2 P_1) = (101)$ since sample 5 is present in pool 1 and pool 3, but not pool 2. Very conveniently (101) is the binary representation of 5 and from this we deduce that it was sample 5 that causes the unique pool pattern (101). Likewise if (only) sample 2 is positive, the pool pattern is (010) because sample two is not present in pool 1 or pool 3. Again (010) is the binary representation of 2 and we know that it was sample 2 that was positive. Finally if the pool pattern is (000) we know that none of the samples were positive.

Another way to see the uniqueness can be understood with the following example. Again suppose that the pool pattern is (101). Since $P_1 = 1$ the positive sample must be either $S_1, S_3, S_5,$ or $S_7$ since those samples are all present in pool 1. But $P_2 = 0$ so that means that $S_2, S_3, S_6$ and $S_7$ cannot be positive since they are all present in pool 2. The contradiction with the pool 1 result eliminates $S_3$ and $S_7$ as possibilities, leaving only $S_1$ and $S_5$. Now $P_3 = 1$ which means that either $S_4, S_5, S_6$ or $S_7$ must be positive. Since $S_5$ must be positive, $S_1$ is eliminated and we have uniquely found that it is sample 5 that is positive.

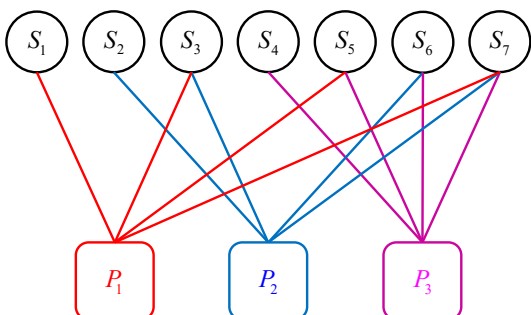

Figure 8: Tanner graph showing how 7 samples $S$ are pooled together into 3 pools $P$. Some samples go into a single pool, some are divided into two pools and one is divided into three pools. The pooling is arranged so that each sample produces a unique pattern of positive results in the pools with the binary number $(P_3 P_2 P_1)$ representing the sample number for the sample that is positive.

### 2.4.2 Example LDPC: The Hamming [7,4,3] Code

The very same mathematics above can be used to construct an LDPC for correcting errors in a message. Suppose we encode a message in a string of seven bits. Assume that the error probability $\epsilon$ per bit is low enough that there is at most one error in the string of seven bits. Then as before, there are eight possible error states. To successfully decode the error state we must reserve three of the seven bits as parity-check bits, leaving a total of four data bits. If we do this, then our code should be able to correct any single bit error (including errors that occur in the parity-check bits!). We will be constructing the $[n, k, d] = [7, 4, 3]$ Hamming code. In this notation $n$ is the number of physical bits, $k$ is the number of error correctable logical (data) bits and $d$ is the code 'distance,' the minimum Hamming distance between any pair of codewords, or equivalently, the minimum number of single qubit errors needed to convert two codewords into each other. (The minimum is taken over all pairs of codewords.) We will see below that the number of bit flip errors that the code can correct is monotonic in $d$ but is slightly less than $d/2$. The 'rate' of this code is the ratio of the number of logical qubits to physical qubits, $r = k/n = 4/7$.

Our code will use the very same matrix $H$ that appeared in the sample pooling protocol above but we will interpret eqn (25) differently. The seven-bit block of coded message we send will be represented by a column vector $M$ similar to $S$ in eqn (26), but now we will allow $M$ to have more than one non-zero entry. The allowed encoded messages are not arbitrary but rather given by the solutions of the equation

$$\begin{pmatrix} P_1 \\ P_2 \\ P_3 \end{pmatrix} = HM = \begin{pmatrix} 0 \\ 0 \\ 0 \end{pmatrix}. \tag{27}$$

Now however we interpret all quantities modulo 2. Thus

$$P_j = \left[ \sum_{k=1}^{7} H_{jk} M_k \right] \bmod 2. \tag{28}$$

That is, we replace all numbers by their parity (0 for even, 1 for odd).

We have 7 physical bits and 3 parity check constraints. We therefore expect there to be 7-3=4 logically encoded bits of information possible in the code. This requires $2^4 = 16$ allowed codewords and therefore 16 solutions to eqn (27). Inspection shows that setting $M$

to be any one of the three rows of $H$ gives a solution to eqn (27). A fourth solution is $M = \vec{0}$, where $\vec{0} = (0000000)^{\mathrm{T}}$. These four solutions give us the encoded versions of (some of) the four data bit combinations allowed in the message plus the three parity check bits determined by those four data bits. Let us label these four encoded messages $M_j$; $j = 0, 1, 2, 3$, with $M_0$ being $\vec{0}$ and $M_j$ for $j = 1, 2, 3$ being the $j$ row in $H$. Four other solutions can be found by taking linear combinations of $M_1, M_2, M_3$ (again using addition mod 2). (Codes in which linear combinations of allowed codewords are also allowed, are called linear codes.) That gives 8 solutions. It is straightforward to verify that the 1's complements of these 8 solutions gives the final 8 solutions. The easiest way to do this is to check that $\vec{1} = (1111111)^{\mathrm{T}}$ is a solution and then use linearity to find the remaining solutions by adding (mod 2) the solution $\vec{1}$ to the solutions already found. (This is what taking the 1's complement means.) Thus, as expected, there are altogether 16 distinct allowed codewords which means that each codeword holds $k = \log_2 16 = 4$ error correctable logical bits.

Now suppose that a single bit flip error occurs during transmission so that the encoded message is

$$M_j \rightarrow M_j' = M_j \oplus S\,, \tag{29}$$

where $S$ is a column vector with one value set to 1 (indicating the location of the corrupted bit) and the rest being 0. Here the notation $\oplus$ means bitwise addition mod 2. The error state is now easily decoded by simply computing

$$\begin{pmatrix} P_1 \\ P_2 \\ P_3 \end{pmatrix} = HM' = H[M \oplus S] = HM \oplus HS = \begin{pmatrix} 0 \\ 0 \\ 0 \end{pmatrix} \oplus HS = HS\,. \tag{30}$$

The key point here is that we have chosen codewords that have to satisfy all the parity checks. Hence locating the error in any codeword is the same problem as locating the error in $\vec{0}$. We can now see why locating the one bit flip is exactly the same problem as locating the one positive SARS COV-2 sample from the measurements on the pooled samples. Therefore we know that our decoder will work!

Since the code distance is $d = 3$, a single error can be corrected. Two errors leave you Hamming distance 2 from the correct word and only Hamming distance 1 from some other incorrect word. The decoder makes the assumption that errors are rare and that it is most likely that only one error occurred, not two. Thus it corrects the error state by moving it to the nearest (in Hamming distance) allowed codeword. This works perfectly for 1 error but fails for two errors (for this particular code). For general code distance $d$, the number of errors that can be corrected is $\lfloor (d-1)/2 \rfloor$.

The Hamming Code is just one particular example but it gets across the basic ideas behind low-density parity-check codes (LDPCs) and linear codes. Useful references include the following:

> https://errorcorrectionzoo.org/,
> https://errorcorrectionzoo.org/c/binary_linear,
> https://errorcorrectionzoo.org/c/hamming,
> https://en.wikipedia.org/wiki/Linear_code,
> https://en.wikipedia.org/wiki/Hamming_code,
> https://en.wikipedia.org/wiki/Hamming(7,4)#All_codewords.

## 2.5 Fault-Tolerant Classical Error Correction

The break-even point $\epsilon^* = \frac{1}{2}$ computed above for the repetition codes is technically known as the 'code-capacity' threshold. This is because we have implicitly assumed that we have

no errors in measuring the states of the individual bits and no errors in the device that does the majority rule calculation and then based on that information flips the errant bit. Thus this threshold is a mathematical property of the code itself but is not representative of any realistic experimental situation. True fault tolerance requires good performance not only when physical bits fail, but also when the circuits we use for error correction are themselves imperfect and require additional (imperfect) circuitry to correct them.[3] Let us therefore reexamine the repetition code for the case $2m + 1 = 3$ bits and look into more detail about how we can build circuits to correct errors using imperfect correction operations. Such error correction circuits (ECC) employ what is known in electrical engineering as 'triple modular redundancy' (TMR) [7]. This is not an especially 'hardware-efficient' scheme (i.e., its implementation brings a large hardware overhead cost), but it is relatively simple and clearly illustrates the key principles and concepts of fault tolerance.

Let us define the *code space* for a bundle of 3 bits (or 3 wires in a circuit) as the states $\{000, 111\}$ and the *correctable space* as the union of the code space and the space of all single-bit errors. That is, the correctable space is the set of states with at most one error (and therefore can be corrected by majority voting). The remaining states with 2 errors are uncorrectable because application of majority voting leads to a logical bit flip error. States with 3 errors are actually back in the code space but differ from the intended state by a logical bit flip. Fig. 9 illustrates the control theory point of view that we have a dynamical system subject to noise which is driving it out of the code space and we are attempting to build a controller (out of noisy components) that keeps the system within the controllable space (i.e., the correctable space) for as long as possible. (It is also useful to review Fig. 3.) The essential idea of fault tolerance is that we recognize that our controller (i.e., the ECC) is itself imperfect. However if we can arrange things so that the errors produced by the imperfect ECC are (almost always) in the category of errors that the controller can fix in the next round of feedback, then the escape from the controllable region will be significantly slowed. For example, if a perfect controller can fix only single physical bit flips, it would be bad if the imperfect version of the controller produced two or more bit flips. It is not so bad however if the imperfect controller produces only single bit flips because the controller will (likely) be able to fix its own error in the next round of correction.

Let us now move from this high-level picture to the specifics of the triple modular redundancy. Fig. 10 illustrates two possible error correction circuits that can correct single bit-flip errors in a 3-bit repetition code. Each circuit uses one or more majority voting units (MAJ) whose output matches the majority of its inputs. Both circuits are capable of correcting a single bit-flip error in the bundle by means of majority voting. However, in circuit (a) the MAJ unit is a single point of failure. If it fails, its output corrupts all three bits in the repetition code, thereby producing an unrecoverable error. (All the bits in the new state of the bundle match and so it appears there is no error when it is examined in the next round of error correction.) If the input to the circuit in Fig. 10(a) lies within the controllable space (i.e., has at most one error), then the output will have zero errors with probability $P_0 = 1 - \epsilon_{\mathrm{M}}$, where $\epsilon_{\mathrm{M}}$ is the failure probability for the MAJ unit. The output will have three errors (i.e., a logical bit flip occurs) with probability $\epsilon_{\mathrm{M}}$.

The circuit shown in Fig. 10(b) is more complex and requires three MAJ units, increasing the probability of one of the units failing. However this circuit is fault-tolerant in the following important sense. If the input to the circuit lies within the controllable space, a single failure in one of the MAJ units, still leaves the output within the controllable space. Note that it is useful to think of an imperfect MAJ unit as a perfect MAJ unit followed by a stochastic unit that produces an identity operation with probability $1 - \epsilon_{\mathrm{M}}$ and a bit flip with probability $\epsilon_{\mathrm{M}}$. Then it is easy to see that, independent of whether there are zero or one errors on the input

---

[3]This is sometimes referred to as the 'Who watches the watchman?' problem.

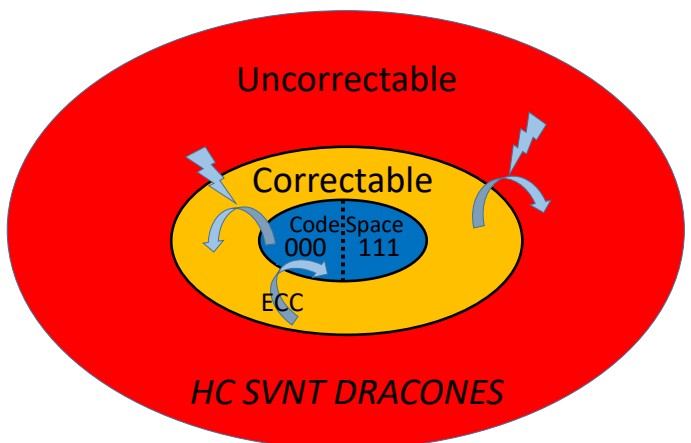

Figure 9: HC SVNT DRACONES ('Here be dragons') was the Latin inscription used to label the unknown and dangerous border regions at the 'edge of the world' on ancient maps. Here we illustrate the connection between error correction and robust control theory. The target region (code space) is a subset of the correctable or controllable region containing (in the case of the 3-bit repetition code) at most one error, and the surrounding dangerous uncontrollable region of two errors. The system is fault-tolerant if the controller (Error-Correction Circuit, ECC) can correct leading-order errors in the codewords and if leading-order imperfections in the controller circuit keep the system within the controllable space. If, due to higher-order errors, the controller fails to keep the system in the controllable region, there will be a transition back into the code space with a logical error. However this failure rate will be higher-order in the physical hardware error rate $\epsilon$.

lines, the three perfect MAJ units correct the error. The subsequent stochastic units produce $n$ errors with probabilities $P_n$ given by

$$P_0 = (1 - \epsilon_M)^3 \,, \tag{31}$$

$$P_1 = 3\epsilon_M (1 - \epsilon_M)^2 \,, \tag{32}$$

$$P_2 = 3\epsilon_M^2 (1 - \epsilon_M) \,, \tag{33}$$

$$P_3 = \epsilon_M^3 \,. \tag{34}$$

There are two critical features of this result. First, the probability to leave the controllable space is (for $\epsilon_M \ll 1$) second order in $\epsilon_M$

$$P_{\text{fail}} = P_2 + P_3 = 3\epsilon_M^2 (1 - \epsilon_M) + \epsilon_M^3 \approx 3\epsilon_M^2 \,. \tag{35}$$

Second this probability distribution for the errors is identical for the case of the input having no errors or having a single error. It does not matter where we are within the controllable space, so long as we are inside it. Thus if there is at most a single error on the input, it is corrected with probability $P_0 \approx 1 - 3\epsilon_M$ and we stay within the controllable space with probability $P_0 + P_1 = 1 - P_{\text{fail}} \approx 1 - 3\epsilon_M^2$.

## 2.6 Fault-Tolerant Memory Operations

Now that we understand how to do fault-tolerant error correction for the 3-bit repetition code. Let us apply it to the problem of error-corrected memory operations. Suppose that a physical memory bit flips randomly in time at rate $\kappa$. That is, the probability of a bit flip occurring in

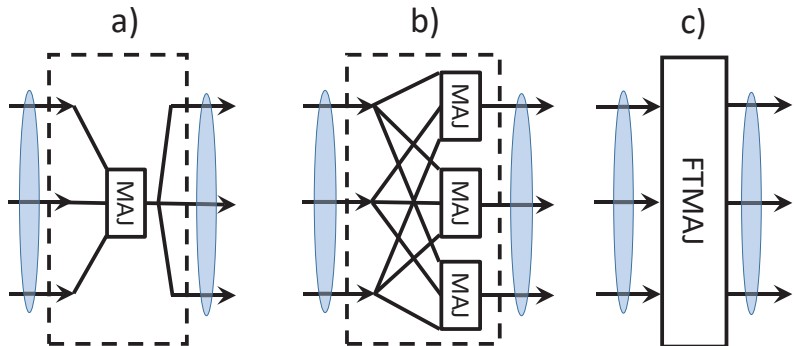

Figure 10: Two possible error correction circuits (ECC) for the 3-bit repetition code. The ellipses mark the bundle of three lines in the code going into and coming out of the ECC. MAJ indicates a majority voting unit whose output is designed to match the majority of its inputs. If the input suffers no more than a single bit flip, the MAJ unit(s) correct this error. Circuit (a) is not fault-tolerant because it has a single point of failure–the MAJ unit. If the MAJ unit fails, its erroneous output fans out and corrupts all three bits in the code, producing an unrecoverable error. Circuit (b) is fault tolerant. If one of the MAJ units fails, it corrupts only a single bit in the 3-bit code, keeping the system within the space of correctable errors and avoiding a logical error. Circuit (c) gives the schematic symbol that will be used to represent the fault-tolerant majority voting ECC in (b).

any small time interval $dt$ is $\kappa\, dt$. Let $\rho_0(t), \rho_1(t)$ be the probabilities that the bit is in state 0 and state 1 respectively at time $t$. The dynamics of the continuous-time Markov process describing the bit flips is captured by the following differential equations

$$\frac{d\rho_0(t)}{dt} = -\kappa\rho_0(t) + \kappa\rho_1(t), \tag{36}$$

$$\frac{d\rho_1(t)}{dt} = +\kappa\rho_0(t) - \kappa\rho_1(t). \tag{37}$$

If the bit starts in state 0 and $t = 0$, the solution to these equations is

$$\rho_0(t) = \frac{1}{2}\left[1 + e^{-2\kappa t}\right], \tag{38}$$

$$\rho_1(t) = \frac{1}{2}\left[1 - e^{-2\kappa t}\right]. \tag{39}$$

Suppose we pick a time $t_0$ at which we can define the memory error probability $\epsilon$ (for that value of $t_0$. Since the bit was in state 0 initially, the error probability is simply

$$\epsilon = \rho_1(t_0) = \frac{1}{2}\left[1 - e^{-2\kappa t_0}\right]. \tag{40}$$

We see that the error probability rises linearly from zero at rate $\kappa$ for short times and then saturates at 50% for long times such that $2\kappa t_0 \gg 1$. Because an error rate of 50% is the same as random guessing, the memory *fidelity* is defined to be the 'true positive' minus the 'false positive' probability

$$F(t_0) = \rho_0(t_0) - \rho_1(t_0) = 1 - 2\epsilon = e^{-2\kappa t_0}. \tag{41}$$

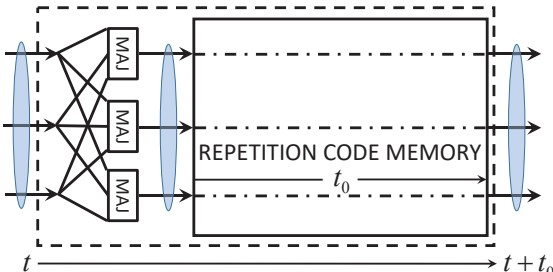

Figure 11: Fault-tolerant memory operation for the 3-bit repetition code. Shaded ellipses indicate wire bundles that represent logical bit in the repetition code. Time increases from left to right. The dashed rectangle outlines one cycle of error correction followed by memory waiting time $t_0$ before the correction cycle repeats. FTMAJ is the fault-tolerant ECC shown in Fig. 10(b-c). The lifetime of the memory is computed by assuming that the input at time $t$ at the beginning of the cycle (far left edge of the dashed rectangle) lies within the controllable space (i.e., has at most one error) and then computing the probability that the memory errors have not taken the system out of the controllable space at time $t + t_0$.

This appropriately starts at unity and decays to zero rather than 50%. An appropriate measure of the memory lifetime is simply the area under the fidelity curve

$$\tau = \int_0^\infty dt\, F(t) = \frac{1}{2\kappa}. \tag{42}$$

Our goal now is to try to increase the memory lifetime using the 3-bit repetition code. From the single bit result we can now compute the probability $P_n$ that $n$ errors have occurred in a 3-bit repetition code memory. The results are of course still given by eqns (7-10). The only difference now is that the value of $\epsilon$ depends how long the logical bit has been stored in the memory. If our FTMAJ error-correction circuit were perfect, we would want to apply it as frequently as practically possible to keep the 3-bit repetition code within the controllable space. However, in the context of fault tolerance, we are forced to recognize that the ECC itself makes errors with probabilities given by eqns (31-34). When $\epsilon_M$ is non-zero, there is an optimum waiting time between error corrections. If we correct too frequently, the imperfections of the ECC dominate. If we wait too long between corrections, the natural error rate in the memory dominates.

To analyze this situation, let us define the time interval $t_0$ to be the time delay between applications of the FTMAJ error correction circuit to fix any errors that have occurred. One cycle of the memory ECC is illustrated in Fig. 11. Following [7], it is convenient to assume that the FTMAJ is applied at the beginning time $t$ and this is followed by the memory storage time $t_0$ when the cycle repeats. We can then use $t_0$ as a variational parameter to optimize the performance of the memory.

Let us assume that the system state at time $t$ is within the controllable space (that is, there is at most 1 error among the three bits). We then apply the FTMAJ circuit and obtain an initial state with an error distribution given by eqns (31-34). The bits then idle for a time $t_0$ before the cycle repeats. We want to know the probability that the system is still within the controllable space at time $t + t_0$ before the cycle repeats.

Following [7], it is useful to define the reliability (success probability) of each individual majority voter

$$R_M = 1 - \epsilon_M. \tag{43}$$

The reliabilty of each bit of the memory is dependent on the waiting time and from eqn (38) is given by

$$R_0 = 1 - \epsilon = \frac{1}{2}\left[1 + e^{-2\kappa t_0}\right]. \tag{44}$$

Because the majority voter and the memory bit are in 'reliability series' and their failures are statistically independent, the reliability of the combination is simply the product of the individual reliabilities [7]

$$R_{M0} = R_M R_0. \tag{45}$$

The reliability of the FT memory operation over the time interval $t_0$ is the probability that the system remains within the controllable space

$$R = R_{M0}^3 + 3R_{M0}^2(1 - R_{M0}). \tag{46}$$

The first term describes the case of zero failures and the second term describes the case of one failure. Using eqns (43-44) and expanding eqn (46) to second order in $\epsilon$ and $\epsilon_M$ yields

$$R \approx 1 - 3(\epsilon + \epsilon_M)^2. \tag{47}$$

We can define an effective logical bit flip rate $\kappa_{\text{eff}}$ via

$$R = \frac{1}{2}\left[1 + e^{-2\kappa_{\text{eff}}t_0}\right] \approx 1 - \kappa_{\text{eff}}t_0, \tag{48}$$

where the approximation in the second equality will be justified later.

For small $\epsilon$ we have from eqn (40) that $\epsilon \approx \kappa t_0$ and hence

$$\frac{\kappa_{\text{eff}}}{\kappa} \approx 3(\epsilon + 2\epsilon_M + \epsilon_M^2/\epsilon). \tag{49}$$

Minimizing this with respect to $t_0$ is the same as minimizing this with respect to $\epsilon$ (for the case of small $\epsilon$) and so the optimal value of $t_0$ is set by

$$\epsilon \approx \epsilon_M, \tag{50}$$

as can be seen in Fig. 12. This makes sense because if $\epsilon \ll \epsilon_M$, we are correcting too often and errors from the ECC are dominating. Conversely if $\epsilon \gg \epsilon_M$, we are waiting too long and memory errors are taking us out of the controllable space and the ECC is unable to help. Finally, we note that if $\epsilon_M \ll 1$, then our approximation $\epsilon \ll 1$ is justified *a posteriori*. Using this optimal value of $\epsilon$ (and therefore $t_0$) we finally obtain our key result

$$\frac{\kappa_{\text{eff}}}{\kappa} \approx 12\epsilon_M. \tag{51}$$

Thus the error correction 'gain' (the factor by which the lifetime is extended by the 3-bit encoding relative to the single bit lifetime $\tau = 1/\kappa$) is (again assuming $\epsilon_M \ll 1$)

$$G_{\text{ECC}} \approx \frac{1}{12\epsilon_M}. \tag{52}$$

Note that because of the approximations we have made, this is actually a lower bound on the gain. The gain takes a significant hit from the relatively large prefactor of 12 in eqn (51). In order to achieve a gain of 10, we would need to be able to achieve $\epsilon_M \approx 0.0083$. Another interesting point is that, because the circuit is fault tolerant (to first order in $\epsilon, \epsilon_M$) the error correction gain approaches infinity as $\epsilon_M \to 0$, but keeping the optimal $\epsilon = \epsilon_M$ in this limit requires the time delay $t_0$ to scale towards zero as

$$t_0 = \frac{\epsilon_M}{\kappa}. \tag{53}$$

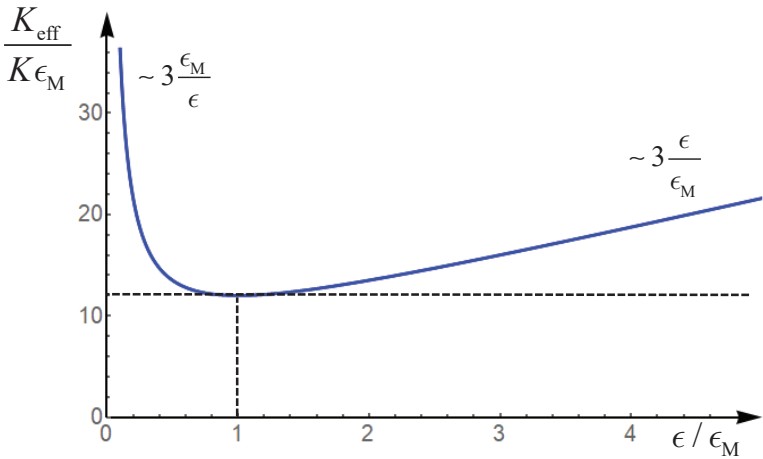

Figure 12: Effective error rate for an error-corrected quantum memory based on the 3-qubit repetition code. $K$ is the physical qubit bitflip rate, $\epsilon_M$ is the majority voting error probability and $\epsilon$ is the memory error that grows with waiting time $t_0$ as given by eqn (40). To the left of the minimum, the fidelity is limited by the errors in the majority voting. To the right of the minimum, the fidelity is limited by the memory errors associated with the wait time.

Hence even with perfect majority voters and correction operations, we ultimately will be limited by the speed of these perfect operations.

Finally it is important to note that while the fidelity of the logical memory will decay more slowly than that of a single bit of physical memory, there is a prefactor in front of the exponential that is smaller than unity because of the cost of encoding (state preparation) and decoding which is more difficult for 3 bits than for 1 bit. Hence even though for long times the FT memory may have much higher fidelity, for very short times, it will be slightly worse. This is a feature common to all error correction circuits.

All of the above results are illustrated with particular examples in Fig. 13, which is worth studying closely. In the right panel, one clearly sees that the $R_M = 1.0$ curve deviates from unity only quadratically at short times, indicating that the system can perfectly correct single errors. The $R_M = 0.925$ curve has a small linear slope caused by the errors in the majority voting circuit. These errors also cause the offset from unit reliability at zero time. However the downward slope is less than for a single physical qubit and the two curves cross at $\kappa t_0 \approx 0.03$. The break-even point is reached at $\kappa t_0 \approx 0.6$ where the physical error probability $\epsilon$ is now large enough that the repetition code fails to improve it.

---

### Exercise 2

Assuming the desired state of a single bit can be prepared without error with probability $1 - \epsilon_1$, find the probability that a 3-bit repetition code logical state can be prepared within the controllable space using the FTMAJ circuit. Now think about how at the very end of the memory time, decoding from the 3-bit repetition code into a single bit must be performed. How much do the infidelities associated with the combined encoding and decoding operations reduce the fidelity at short times relative to the use of a single physical bit for memory?

---

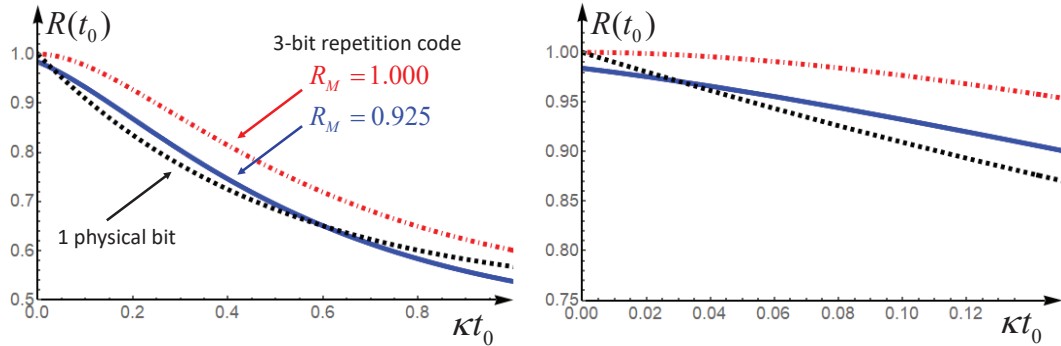

Figure 13: Reliability $R$ of FT memory operation for one cycle of error correction followed by memory idle for time $t_0$. The x-axis is dimensionless time $\kappa t_0$, where $\kappa$ is the single physical flip rate. Dot-dash curve is for perfect majority voting units with reliability $R_M = 1.0$. Solid curve is for the case $R_M = 0.925$. The dashed curve is the unencoded single physical bit memory reliability. Right-hand panel shows a close-up of the curves for small times. Note: these plots do not include any errors from the final readout of the memory after many correction cycles.

---

### Exercise 3

Assume the existence of a MAJ circuit that can handle 5 inputs and has error probability $\epsilon_M$. Repeat the above analysis to obtain the optimal value of the error correction gain $G_{ECC}$. How does it scale with $\epsilon_M$ and why?

---

## 2.7 Fault-Tolerant Classical Gate Operations

So far we have discussed only memory operations. The desired memory operation is simply the identity–we want to get back what we put in. To do computation we of course need completely general types of gates, not just the identity. Let us consider the NAND gate (NOT-AND). This gate has two inputs and one output and it is universal for classical computation–any desired Boolean function can be represented with a circuit constructed solely of NAND gates. Fig. 14 gives the standard schematic representation for the NAND gate and its truth table is shown in Table 1.

Table 1: Truth table for the NAND gate.

| Input A | Input B | Output Q |
|:---:|:---:|:---:|
| 0 | 0 | 1 |
| 0 | 1 | 1 |
| 1 | 0 | 1 |
| 1 | 1 | 0 |

Let us assume that each physical NAND gate has a probability $\epsilon$ of failing. Hence the reliability is

$$R_0 = 1 - \epsilon. \tag{54}$$

This is the same as eqn (44) except that here, unlike the case of the memory, we take $\epsilon$ to

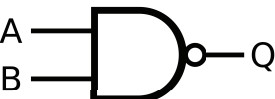

Figure 14: Standard circuit diagram symbol for the NAND gate which has two input bits, A and B, and one output bit Q.

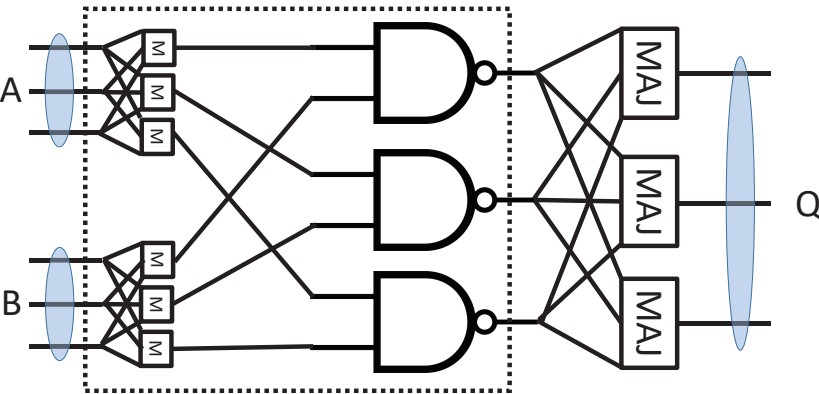

Figure 15: Fault-tolerant logical NAND gate for the 3-bit repetition code. Shaded ellipses indicate wire bundles that represent logical bits in the repetition code. The dashed rectangle encloses the components whose reliability determines overall reliability of the logical NAND.

be a constant that does not change with operating time. We take the conservative point of view that if the NAND gate fails to give the correct answer for any of its 4 possible inputs, we consider that it always fails (even though the output may be correct some of the time). Let us again consider the 3-bit repetition code. The corresponding fault-tolerant logical NAND is shown in Fig. 15.

We see that the design is similar to the memory repetition code. The wire bundle representing logical input bit A(B) feeds each of the A(B) inputs of the three NAND gates. The three Q output lines are sent through the same majority voting error-correction circuit that was used for the fault-tolerant memory circuit. We use the components enclosed within the dashed rectangle to compute the circuit reliability. As with the memory, we define the circuit reliability to be the probability that if the logically encoded input(s) all lie in the controllable space, then so does the output bundle. Unlike the memory operation, the NAND operation has two inputs which gives it more ways to fail. The analog of eqn (45) is thus

$$R_{M0} = R_M^2 R_0 \, . \tag{55}$$

The difference arises because the output of the physical NAND is reliable only if the physical NAND is reliable and both of the two majority voters that feed it are also reliable. With this difference in the definition of $R_{M0}$, eqn (46) for the overall reliability of the logical circuit operation remains valid. Inserting eqn (55) into eqn (46) and expanding to second order in the failure probabilities yields

$$R \approx 1 - 3[2\epsilon_M + \epsilon]^2 \, . \tag{56}$$

The fault-tolerance gain is therefore

$$G_{\text{NAND}} = \frac{\epsilon}{3[2\epsilon_M + \epsilon]^2} \, . \tag{57}$$

Unlike the case of the fault-tolerant memory in which $\epsilon$ was a variational parameter, here the parameters are fixed. However, for the particular case $\epsilon = \epsilon_M$, we obtain $G_{\text{NAND}} = \frac{1}{27\epsilon_M}$. We see that the gain is less than for the fault-tolerant memory because of the additional hardware parts count for the NAND gate associated with the fact that it has two inputs.

---

**Exercise 4**

Draw schematic circuits using only NAND gates and fan in/out that realize the following gates

    a) NOT

    b) AND

    c) XOR (exclusive OR)

    d) NOR (NOT OR)

    e) MUX (direct one of many inputs to one output conditioned on the state of control bits)

    f) DEMUX (direct one input to one of many outputs conditioned on the state of control bits)

    g) MAJ gate for the 3-bit repetition code

---

## 2.8 Recursion and Fault-Tolerance Thresholds

We have now seen two related examples for the construction of fault-tolerant circuits, one for memory and one for the universal NAND gate. These circuits are fault-tolerant in the sense that, if the physical error probability is below the break-even value, the logically encoded circuit has higher reliability than the minimal physical circuit. This is true despite that the fact that the encoded circuit contains more elements and the error-correction component of the circuit may itself be faulty. The key question that now arises is, 'Can we make the error rate arbitrarily low?' The answer is yes, provided that the physical error rate is sufficiently small and the design is fully fault-tolerant. One way, not necessarily the optimal way, to achieve this goal is to recursively iterate the encoding. The zeroth level of recursion is the physical memory bits and NAND gates. The first level error correction is a $n$-bit repetition code scheme as described above (with $n = 2m + 1$). The second level of recursion is to make a similar $n$-bit repetition encoding using the level-1 memory bits and NAND gates (that are themselves encoded). In theory, this recursion can be repeated as often as needed until the error probability is below any specified level. Table 2 shows the example of a recursively constructed memory circuit based on an $n$-bit repetition code. We see that the hardware count grows exponentially with the recursion level. However the logical error probability falls doubly exponentially. The reason for this is that the error correction gain gets larger and larger as the error rate falls with each iteration. Thus the logical error probability falls exponentially with the hardware cost.

    We see that fault tolerance is a collective phenomenon that appears only in the 'thermodynamic limit' of infinite hardware resources. While this can never be reached, the key point is that, in principle, we can approach exponentially close to perfect operations. In practice however, it seems reasonable to argue on physical grounds that all fault tolerance is only approximate. If we define the effective error probability at level $j$ in the recursion to be $\epsilon_j$, we

Table 2: Recursive fault tolerance for a memory circuit that uses $n$ physical bits to encode one logical bit that can correct a single error. The hardware cost grows exponentially with recursion level, but the error probability falls double exponentially. The error correction gain at the first recursion level is defined to be $G = 1/(c_n \epsilon)$. The coefficient $c_n$ depends on the particular code. If the code can correct more than one error, the logical error rate falls even faster with recursion level.

| Recursion Level | Hardware Overhead | Leading Error Probability | Error Correction Gain |
|---|---|---|---|
| 0 | 1 | $\epsilon$ | 1 |
| 1 | $n^1$ | $\sim c_n \epsilon^2$ | $G$ |
| 2 | $n^2$ | $\sim c_n (c_n \epsilon^2)^2 = c_n^3 \epsilon^4$ | $G^3$ |
| 3 | $n^3$ | $\sim c_n^7 \epsilon^8$ | $G^7$ |
| L | $n^L$ | $\sim (1/c_n)(c_n \epsilon)^{2^L}$ | $G^{(2^L-1)}$ |

have for a code that corrects errors to first order the recursion relation

$$\epsilon_{j+1} = c_n (\epsilon_j)^2, \tag{58}$$

where $c_n$ is a constant that depends on the particular code. Under recursion, the error probability falls to zero, provided that the initial error, $\epsilon_0 \equiv \epsilon$ is below the fault-tolerance threshold $\epsilon^* = 1/c_n$.

Since this is physics and not mathematics, we might more generally expect the recursion relation to look something like this

$$\epsilon_{j+1} = \lambda \epsilon_j + c_n (\epsilon_j)^2, \tag{59}$$

where $\lambda \ll 1$ is a constant. The $\lambda$ term in this purely phenomenological expression represents the fact that once we have pushed the errors down to some extremely small level, we will discover that our error model is not exactly correct and something is going wrong. By (crude) analogy with a Landau-Ginsburg free energy expansion, we might ask what symmetry would guarantee that $\lambda$ vanishes. In this case the 'symmetry' is the assumption that all qubits have uncorrelated errors. These correlated errors could occur for some physical reason (e.g. someone unplugged the power supply to the computer) or because the procedure designed to correct single-bit errors is not perfectly fault tolerant and has a small probability of causing two-bit errors.

This symmetry may be a very good approximation but is unlikely to be exactly true. For small but non-zero $\lambda$, the error rate will initially fall doubly exponentially but once $\epsilon_j \ll \lambda \epsilon^*$ the recursion relation becomes well approximated by

$$\epsilon_{j+1} = \lambda \epsilon_j. \tag{60}$$

Hence the flow slows from double exponential to single exponential decay. One might think it is good news that the error rate is falling exponentially, but don't forget that the hardware cost is rising exponentially. Hence, the error rate is converging only as the inverse of a polynomial (rather than exponentially) in the hardware cost. This is what it means to be non-fault tolerant. You can keep making the error smaller but it becomes exponentially expensive in hardware to achieve an exponentially small error probability.

# 3  Quantum Error Correction For Two-Level Systems (Qubits)

> Q: Is quantum information carried by waves or particles?
> A: Yes!
>
> Q: Is quantum information analog or digital?
> A: Yes!

We are now ready to enter the remarkable and magic world of quantum error correction. In the last 20 years there has been tremendous experimental progress and the coherence times of superconducting qubits have risen more than five orders of magnitude. Even if this progress can be continued, it is good to recall the fundamental law of quantum information science:

> **'There is no such thing as too much coherence.'**

No matter how good the hardware is (and presently it is much worse than standard classical hardware), or how good it becomes in the future, there will always be demand for quantum computers that can run longer (execute algorithms with greater 'circuit depth'). The current grand challenge in the field is to develop (nearly) fully fault-tolerant systems with dramatically enhanced reliability, fidelity of operations and memory coherence times. Without robust quantum error correction, large-scale quantum computation will most likely be impossible.

In the author's view, the fact that quantum error correction is even possible in principle is much more amazing and counter-intuitive than the possibility of quantum computation itself. Naively, it would seem that quantum error correction is completely impossible. Recall that classical error correction codes work by copying information about the data into ancillary bits. We saw above that the simplest example of such a classical error correcting code is the repetition code which works by simply making redundant copies of the data and then making measurements to enforce majority voting to predict the correct state of the data and recovery from errors. There are a number of seemingly insurmountable difficulties in applying this approach to the quantum case. First, the *no-cloning theorem* [13, 14] (described below) does not allow us to copy an unknown quantum state of a qubit onto ancilla qubits. Recall that the classical 3-bit EEC described above used 'fanout' by 3x to send the same bit state to 3 different MAJ units.

> **The no-cloning theorem tells us**
> **'There is no such thing as quantum fanout.'**

Second, in order to determine if an error has occurred, we would have to make a measurement, and the back action (state collapse) from that measurement would itself produce random unrecoverable errors.

There is a third fundamental difference between quantum and classical gates. Let us recall the classical NAND gate discussed above. We saw that it has two inputs but only one output. From the one bit of output it is (in general) impossible to deduce which of the 4 input states produced it. That is, the NAND gate is irreversible–information is lost in going from the input to the output. Such information loss is impossible in a quantum system because every gate is represented by a unitary operation that maps the Hilbert space onto itself. Every unitary $U$ has by definition an inverse $U^{-1} = U^{\dagger}$. Hence quantum gates are reversible and information cannot be destroyed by a gate. The only way that a quantum gate can cause loss of information is if it entangles a system with an unobservable bath which quickly scrambles the information and

makes it effectively unrecoverable. This is precisely what decoherence is–the bath acquiring information about the system and causing 'measurement-induced dephasing' [15, 16]

Charles Bennett [17] invented classical logic that is reversible. In such a setup every gate must of course have as many wires (bits) at the output as at the input. This is relevant to thermodynamically reversible classical computation engines for which the only energy cost comes from erasing information to reset the computer. This changes the entropy by $\Delta S = k_B \ln 2$ per bit. Irreversibly pushing this entropy into a heat bath costs energy $T \Delta S$. The computation itself can in principle cost no energy at all. Bennett's analysis is interesting in the study of the trade-off between Shannon entropy of information and thermodynamic entropy. It is also a kind of classical precursor to reversible quantum computation.

It is important to understand another feature of the quantum case which makes it more subtle than the classical case. In the classical case, a bit of information $b$ can be encoded in two different voltage levels of a circuit, say 0 and +5 volts. For example, voltage 0 can represent $b = 0$ and +5 volts can represent $b = 1$. The only other possibility is the reverse: voltage zero represents $b = 1$ and +5 volts represents $b = 0$. There are only two possible encodings and they differ simply by the NOT operation. If Alice sends Bob a message without telling him which encoding she is using, it is an easy task for him to try to read the message and if it makes no sense, he can take the complement of the message (i.e., apply the NOT operation to all the bits) and successfully read the message.

Things are very different in the quantum case. The states of a qubit are defined by the co-latitude $\theta$ and longitude $\varphi$ on the Bloch sphere using the standard parametrization

$$|+\hat{n}\rangle = \cos\left(\frac{\theta}{2}\right)|\uparrow\rangle + \sin\left(\frac{\theta}{2}\right)e^{i\varphi}|\downarrow\rangle, \tag{61}$$

$$|-\hat{n}\rangle = +\sin\left(\frac{\theta}{2}\right)|\uparrow\rangle - \cos\left(\frac{\theta}{2}\right)e^{i\varphi}|\downarrow\rangle. \tag{62}$$

These basis states are the eigenstates of $\hat{n} \cdot \vec{\sigma}$, with eigenvalues $\pm 1$ which can represent bit values $|b = 0\rangle$ and $|b = 1\rangle$ respectively. Note however, that the quantization axis $\hat{n}$ is the unit vector

$$\hat{n} = (\sin(\theta)\cos(\varphi), \sin(\theta)\sin(\varphi), \cos(\theta)), \tag{63}$$

defined by two real numbers and hence requires an infinite number of bits to specify. When Alice measures $\hat{n} \cdot \vec{\sigma}$ for a qubit prepared in an unknown state, she obtains either $+1$ or $-1$ and thus acquires precisely one classical bit of information, just as she would for a randomly chosen classical bit. This upper bound on the amount of classical information (1 bit) that can be accessed via measurement is called the *Holevo bound* [18, 19].

Each choice of quantization axis effectively corresponds to a different encoding of the information. Since Alice can choose an arbitrary quantization axis, the number of possible encodings is infinite. Let us define Bob's basis choice to be the standard basis $|\uparrow\rangle, |\downarrow\rangle$, corresponding to quantization axis $\hat{z} = (0, 0, 1)$. What happens if Alice chooses to use the quantization axis $\hat{n}$ to encode her message? If Alice, sends Bob the state $|+\hat{n}\rangle$, Bob will obtain the measurement result $+1$ with probability

$$p_+ = \langle\uparrow | +\hat{n}\rangle\langle+\hat{n}| \uparrow\rangle = \cos^2\left(\frac{\theta}{2}\right), \tag{64}$$

and measurement result -1 with probability

$$p_- = \langle\downarrow | +\hat{n}\rangle\langle+\hat{n}| \downarrow\rangle = \sin^2\left(\frac{\theta}{2}\right). \tag{65}$$

If Alice chooses $\hat{n} = +\hat{z}$ ($\theta = 0$) or $\hat{n} = -\hat{z}$ ($\theta = \pi$), then we essentially have the classical result. There is no randomness and the message Bob receives will either exactly match the

message Alice sent or will be its exact complement (i.e., differ by a NOT operation). However as the quantization axis moves away from $\pm\hat{z}$ the measurement result will become more and more random and for the case that $\hat{n}$ lies along the equator of the Bloch sphere, Bob's result will be completely uncorrelated with the message Alice intended.

**Box 2. No Cloning Theorem:** *The no-cloning theorem states that it is impossible to make a copy of an unknown quantum state. The essential idea of the no-cloning theorem is that in order to make a copy of an unknown quantum state, you would have to measure it to see what the state is and then use that knowledge to make the copy. However measurement of the state produces random back action (state collapse) and it is not possible to fully determine the state. This is a reflection of the fact that measurement of a qubit yields one classical bit of information which is not enough in general to fully specify the state via its latitude and longitude on the Bloch sphere.*

*Of course if you have prior knowledge, such as the fact that the state is an eigenstate of $\sigma^x$, then a measurement of $\sigma^x$ tells you the eigenvalue and hence the state. The measurement gives you one additional classical bit of information which is all you need to have complete knowledge of the state.*

*A more formal statement of the no-cloning theorem is the following. Give an unknown state $|\psi\rangle = \alpha|\uparrow\rangle + \beta|\downarrow\rangle$ and an ancilla qubit initially prepared in a definite state (e.g. $|\downarrow\rangle$), there does not exist a unitary operation U that will take the initial state*

$$|\Phi\rangle = [\alpha|\uparrow\rangle + \beta|\downarrow\rangle] \otimes |\downarrow\rangle, \tag{66}$$

*to the final state*

$$\begin{aligned} U|\Phi\rangle &= [\alpha|\uparrow\rangle + \beta|\downarrow\rangle] \otimes [\alpha|\uparrow\rangle + \beta|\downarrow\rangle] \\ &= \alpha^2|\uparrow\uparrow\rangle + \alpha\beta[|\uparrow\downarrow\rangle + |\downarrow\uparrow\rangle] + \beta^2|\downarrow\downarrow\rangle. \end{aligned} \tag{67}$$

*unless U depends on $\alpha$ and $\beta$.*

*The proof is straightforward. The RHS of eqn (66) is linear in $\alpha$ and $\beta$, whereas the RHS of eqn (67) is quadratic. This is impossible unless U depends on $\alpha$ and $\beta$. For an unknown state, we do not know $\alpha$ and $\beta$ and therefore cannot construct U.*

**Box 3. Quantization Axis as Encoding** *Viewing the choice of quantization axis as an encoding choice naturally leads us to the concept of intrinsic randomness in the results of quantum measurements. How else can we reconcile the fact that all quantum measurement results of $\hat{n} \cdot \vec{\sigma}$ are discrete ($\pm 1$) with the fact that surely the physics is continuous as $\hat{n}$ is continuously varied between the two 'classical' encodings $+\hat{z}$ and $-\hat{z}$? It is only randomness in the discrete measurement results that allows something to be continuous–namely the probability distribution which evolves continuously as $\hat{n}$ is varied.*

A peculiar feature of quantum measurements that is important to note is that, because the state collapses when Bob measures it, he always obtains one of the eigenvectors of the measurement operator and has no idea if the act of measurement caused any back action which makes the state now different from what Alice sent. That is, if Bob makes a measurement of $\hat{m} \cdot \vec{\sigma}$ for some arbitrary axis $\hat{m}$, he is asking the question, 'Does the spin lie along the $\pm\hat{m}$

direction?' Remarkably, the answer is always, yes! It does not matter what the initial state actually was. (This only affects the probabilities of the different outcomes.) This foundational notion in quantum information has been succinctly summarized by Sasha Korotkov in the following statement:

> **'In quantum mechanics, you don't see what you get.**
> **You get what you see!**

Of course if Alice sends Bob many copies of the same state, Bob can obtain an estimate of $\langle \vec{\sigma} \rangle$ by making measurements of each spin component. He can then align his decoding frame to Alice's encoding frame.

---

### Exercise 5

*No-cloning theorem prevents superluminal communication.*

Bob and Alice are far apart from each other but share a single Bell pair

$$|B_0\rangle = \frac{1}{\sqrt{2}} \left( |\uparrow\rangle|\downarrow\rangle - |\downarrow\rangle|\uparrow\rangle \right).$$

Show that if cloning an unknown state is possible, the relativistic prohibition on superluminal communication can be violated. Hint: Let Alice choose to measure her half of the Bell pair in either the $X$ basis or the $Z$ basis. This choice determines a one-bit message that she wishes to send.

(a) Describe in detail a protocol that Bob can employ to immediately and reliably read this message if, and only if, cloning is possible.

(b) Explain how, by choosing an arbitrary measurement axis $\hat{n}$, Alice could send an arbitrarily large number of bits to Bob using only a single Bell pair. Estimate how many copies of his qubit Bob would have to clone, to reliably receive $M$ bits from Alice.

---

### Exercise 6

**Adaptive frame estimation** Suppose that Alice and Bob want to establish a common quantization axis for their joint experiments. Alice agrees to send Bob $N \gg 1$ copies of the same state $|+\hat{n}\rangle$. Bob is free to make each measurement with an arbitrary choice of quantization direction $\hat{n}'$. Describe qualitatively how Bob should adaptively choose his measurement directions based on past measurement results so as to converge as rapidly as possible on Alice's choice of quantization axis?

Suppose that Bob has managed to find an axis $\hat{n}'$ that is close to Alice's: $|\hat{n}' - \hat{n}| = \epsilon \ll 1$. Estimate how many additional measurements Bob will need to make to reduce the error to $\epsilon/2$. That is, describe the asymptotic convergence of the optimal adaptive measurement algorithm.

---

It turns out to be surprisingly difficult to isolate the key features that give a quantum computer its extraordinary power. Part of the answer can be found in its analog character–quantum superposition states have continuous real (or complex) amplitudes. This analog character is worrisome however because it might mean that small errors will destroy the extra power of the

computer just as they do for classical analog computers.[4] Remarkably, the situation is saved by the fact that the quantum computer also has characteristics that are digital. Because any measurement of the state of a qubit always yields a binary result, *measured* quantum errors are discrete even though the errors themselves are continuous. This is a novel situation in which state collapse is our friend.[5] This dual analog/digital character makes it possible to perform quantum error correction and (in principle) obtain nearly ideal behavior from imperfect and noisy hardware. I personally feel that this discovery by Peter Shor in 1995 [20,21] and by Andrew Steane in 1996 [22,23] is even more amazing than the concept of quantum computation (on ideal hardware) itself. The reader may wish to consult the reviews by Raussendorf [24] and by Gottesman [25–27].

But now an important question emerges: can we really take advantage of state collapse and use it as a resource to help us correct errors? Isn't quantum error correction impossible because the act of measurement to check if there is an error would collapse the state, destroying any possible quantum superposition information? Also we have to be able to perform quantum error correction on unknown states–and even in situations where a qubit does not have its own definite state because it is entangled with other qubits during the course of a calculation. Remarkably however, one can encode the information in such a way that the presence of an error can be detected by measurement, and if the code is sufficiently sophisticated, the error can be corrected, just as in classical computation. The key will be collapsing the state onto a definite error state without learning anything about the logical information being stored.

We discussed above the fact that classically there are only two possible encodings for 1 bit of information in one physical bit. Therefore, the only classical physical error that exists is the bit flip.[6] Quantum mechanically there are other types of errors (e.g., phase flip, energy decay, erasure channels, etc.). Some of these errors correspond to random unitaries applied by the environment (or accidentally by the imperfect control circuits) to the qubit; e.g., $\sigma^x$ produces bit flip and $\sigma^z$ produces phase flip. We can even have 'coherent errors' corresponding to unitary rotations around an arbitrary axis $\hat{m}$

$$U = e^{-i\frac{\theta}{2}\hat{m}\cdot\vec{\sigma}} = \cos\frac{\theta}{2}\hat{I} - i\sin\frac{\theta}{2}\hat{m}\cdot\vec{\sigma}\,. \tag{68}$$

The second equality shows that this operation produces a coherent superposition of four possibilities: no error (identity), bit flip ($\sigma^x$), phase flip ($\sigma^z$), and both ($\sigma^y = i\sigma^x\sigma^z$). Any error correction scheme thus has to deal with situations where we have quantum uncertainty about whether there is an error and what type it might be! Furthermore, some errors correspond to irreversible, non-unitary quantum operations, e.g., energy relaxation described by

$$\sigma^- = \frac{1}{2}\left[\sigma^x - i\sigma^y\right]\,. \tag{69}$$

For a more complete discussion of quantum operations see Chap. 8 in [19].

Despite these significant additional complications and subtleties, quantum codes have been developed [19–23, 25, 27, 28] (using a minimum of 5 qubits) which will correct all possible quantum errors. By concatenating these codes to higher levels of redundancy, even small imperfections in the error correction process itself can be corrected. Thus quantum superpositions can in principle be made to last arbitrarily long even in an imperfect noisy system provided the noise is sufficiently weak and is uncorrelated on different qubits. It is this remarkable insight

---

[4]There is a no-go theorem for error correction in classical analog computers. The continuous states of analog computers are all equally valid, and thus one cannot tell if an error has shifted one of them.

[5]We will not discuss the case of autonomous error correction in which information about errors is gradually acquired by continuous quantum non-demolition monitoring. The fact that this is possible is also related to the ability of QND measurements to gradually project the system onto a discrete error state.

[6]Excluding the possiblity of erasure errors in which the bit is lost or destroyed.

that makes quantum computation possible. Many other ideas have been developed to reduce error rates. Kitaev in particular has developed novel theoretical ideas for topologically protected qubits (toric and surface codes) which are impervious to local perturbations [29–31]. The surface code model for superconducting qubits is being actively pursued, but technological advances will likely be required to reach the break-even point [32].

Certain topological field theories contain excitations that have non-abelian braiding statistics. That is to say, moving the defects around each other in space moves the system from one degenerate ground state to another. In certain cases these braiding operations can be used to perform quantum computations. Because local perturbations in the Hamiltonian cannot mix different topologically protected states, the hope is that such a system might realize 'topologically protected' qubits that avoid the necessity of quantum error correction. This dream has not yet been realized, but much effort is being devoted to creating quantum materials that are (hopefully) appropriate to the task [33].

Let us begin our study of quantum error correction by considering the quantum version of the repetition code. Recall that the minimal classical repetition encoding step involves making two copies of the first data bit and then using majority voting to correct for (single) bit-flip errors. Unfortunately, the no-cloning theorem prevents replication of an unknown qubit state because there is no unitary transformation $U$ which takes

$$[\alpha|0\rangle + \beta|1\rangle] \otimes |00\rangle \longrightarrow [\alpha|0\rangle + \beta|1\rangle]^{\otimes 3}. \tag{70}$$

Of course such a unitary *can* be used to prepare this state if we know the parameters $\alpha$ and $\beta$. However, as discussed in Box 2, the no-cloning theorem prevents us from doing this for an unknown state because, from the linearity of quantum mechanics and the fact that the RHS is a non-linear function of the amplitudes $\alpha, \beta$, it follows that in general $U$ must be a function of those same amplitudes. The key point for quantum error correction is that we have to be able to correct errors in unknown states such as occur in the middle of large computations where the bits are highly entangled with each other. Using the quantum circuit illustrated in Fig. 16, one can however perform the repetition code transformation:

$$[\alpha|0\rangle + \beta|1\rangle] \otimes |00\rangle \longrightarrow [\alpha|000\rangle + \beta|111\rangle], \tag{71}$$

since this is in fact a unitary transformation and the RHS is still a linear function of $\alpha, \beta$. The qubits are entangled but the original state has not be cloned. The encoding circuit requires no knowledge of the amplitudes $\alpha, \beta$ and is carried out using two controlled-NOT (CNOT) gates described in Box 4.

Just as in the classical case, these three physical qubits form a single logical qubit. The two logical basis states are

$$|0_{\mathrm{L}}\rangle = |000\rangle, \tag{72}$$
$$|1_{\mathrm{L}}\rangle = |111\rangle. \tag{73}$$

The analog of the single-qubit Pauli operators for this logical qubit are readily seen to be

$$\begin{aligned} X_{\mathrm{L}} &= X_1 X_2 X_3, \\ Y_{\mathrm{L}} &= i X_{\mathrm{L}} Z_{\mathrm{L}} = -Y_1 Y_2 Y_3, \\ Z_{\mathrm{L}} &= Z_1 Z_2 Z_3, \end{aligned} \tag{74}$$

and they obey the usual commutation relations of the Pauli matrices.

We see that this logical encoding complicates things considerably because now to do even a simple single logical qubit rotation we have to perform some rather non-trivial three-qubit joint operations. It is not always easy to achieve an effective Hamiltonian that can produce

such joint operations, but this is an essential price we must pay in order to carry out quantum error correction. In fact, it is the very 'unnaturalness' of these logical operations that makes it difficult for the environment to carry them out and thereby destroy the stored information. We have to find a way to beat the environment at this game. Further below we will expand on the idea that quantum error correction is an adversarial game and discuss in more detail what capabilities we need relative to those of the environment if we are to win the game.

---

**Box 4. The CNOT Gate:** *CNOT is a two-qubit gate that flips the target qubit, if and only if, the control qubit is in the $|1\rangle$ state:*

$$\text{CNOT} = \left(\frac{I_1 + Z_1}{2}\right)X_2 + \left(\frac{I_1 - Z_1}{2}\right)I_2. \tag{75}$$

*The first parentheses enclose the projector onto the up state for the control qubit (qubit 1). The second parentheses enclose the projector onto the down state for the control qubit. Thus if qubit 1 is in $|\uparrow\rangle = |1\rangle$, then $X_2$ flips the second qubit while the remaining term vanishes. Conversely when qubit 1 is in $|\downarrow\rangle = |0\rangle$, the coefficient of $X_2$ vanishes and only the identity in the second term acts.*

*In the classical context one can imagine measuring the state of the control bit and then using that information to control the flipping of the target bit. However in the quantum context, it is crucially important to emphasize that measuring the control qubit would collapse its state. We must therefore avoid any measurements and seek a unitary gate which works correctly when the control qubit is in $|0\rangle$ and $|1\rangle$ and even when it is in a superposition of both possibilities $\alpha|0\rangle + \beta|1\rangle$. It is this latter situation which will allow us to generate entanglement. When the control qubit is in a superposition state, the CNOT gate causes the target qubit to be both flipped and not flipped in a manner that is correlated with the state of the control qubit. These are not ordinary classical statistical correlations (e.g. clouds are correlated with rain), but rather special (and powerful) quantum correlations resulting from entanglement.*

---

### Exercise 7

Prove that the CNOT gate operation is
a) Hermitian
b) Squares to the Identity
c) From (a) and (b), prove that CNOT is unitary.

---

It turns out that this simple code cannot correct all possible quantum errors, but only a single type of error on at most one qubit. This fact is consistent with the counting argument we used in the classical case. With three classical bits we have 8 states which gives us enough 'room' to describe one data bit and four possible error states (no error or one bit flip at three possible locations). Similarly the 3-bit quantum repetition code can correct one bit flip or one phase flip but not both. For specificity, let us take the error operating on our system to be a single bit flip, either $X_1, X_2$, or $X_3$. These three together with the identity operator, $I$, constitute the set of operators that produce the four possible error states of the system we will be able to correctly deal with. Following the formalism developed by Daniel Gottesman [24, 25, 27], let

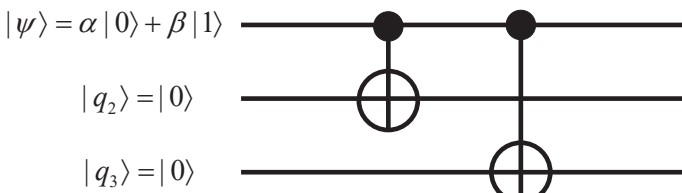

Figure 16: Quantum circuit to encode a single physical qubit into a logical qubit based on the 3-qubit repetition code. The entangled state of the logically encoded bit is created using two CNOT operations to entangle the two ancillae with the first qubit. The circuit is 'read' from left to right, but each operation corresponds to a unitary gate applied right to left. The solid circle denotes the control qubit and the open circle denotes the target qubit of the CNOT gate. The target qubit is flipped if and only if the control qubit is in the 1 state. Since the control qubit is in a superposition state, the initial product state turns into an entangled state.

us define two *stabilizer* operators

$$S_1 = Z_1 Z_2 \,, \tag{76}$$
$$S_2 = Z_2 Z_3 \,. \tag{77}$$

These stabilizers have two nice properties: they commute with each other (i.e., $[S_1, S_2] = S_1 S_2 - S_2 S_1 = 0$) and they commute with all three of the logical qubit operators listed in eqn (74). The first property ensures that they can both be measured simultaneously and the latter property means that the act of measurement does *not* destroy the quantum information stored in any superposition of the two logical qubit states. Said another way, the logical qubit basis states have been chosen to be +1 eigenstates of $S_1$ and $S_2$. Thus every superposition of the two logical basis states is also a +1 eigenstate of the two stabilizers, and hence measurement of the stabilizer does not collapse states in the logical space in any way.

Furthermore the stabilizers each commute or anticommute with the four error operators in such a way that we can uniquely identify what error (if any) has occurred. Each of the four possible error states (including no error) is an eigenstate of both stabilizers with the eigenvalues listed in Table 3.

---

**Exercise 8**

Prove that when the important minus sign is included in the definition of the logical operators introduced above, they obey the correct Lie algebra for Pauli operators

$$[X_L, Y_L] = 2i Z_L \,, \quad [Y_L, Z_L] = 2i X_L \,, \quad [Z_L, X_L] = 2i Y_L \,. \tag{78}$$

---

Measurement of a stabilizer yields one bit of classical information and is referred to as the error syndrome.[7] Thus measurement of the two stabilizers yields two bits of classical information which uniquely identify which of the four possible error states the system is in and allows the experimenter to correct the situation by applying the appropriate error operator, $I, X_1, X_2, X_3$ to the system to cancel the original error.

---

[7]'Error syndrome' can also refer to the full collection of stabilizer measurement results.

Table 3: Stabilizer eigenvalues for the 4 different correctable error states of the 3-bit quantum repetition code.

| error | $S_1$ | $S_2$ |
|:---:|:---:|:---:|
| $I$ | $+1$ | $+1$ |
| $X_1$ | $-1$ | $+1$ |
| $X_2$ | $-1$ | $-1$ |
| $X_3$ | $+1$ | $-1$ |

It is instructive to compare the stabilizer measurements to the majority voting algorithm used in the classical 3-bit repetition code. In the classical case, the simplest way to do majority voting is to measure each bit and see if one is different from the others. If we were to do this in the quantum case, it would correspond to measuring $Z_1, Z_2, Z_3$ and then multiplying the measurement results to form $S_1$ and $S_2$. This leads to disaster because now, in addition to learning the values of $S_1$ and $S_2$, we have learned the values of each of the qubits. Hence the quantum superposition is destroyed. The stabilizers are two-qubit joint parity operators. They tell us whether two qubits are in the same state without telling us what that state is!

$$Z_1 Z_2 |00\rangle = +|00\rangle, \tag{79}$$

$$Z_1 Z_2 |11\rangle = +|11\rangle, \tag{80}$$

$$Z_1 Z_2 |01\rangle = -|01\rangle, \tag{81}$$

$$Z_1 Z_2 |10\rangle = -|10\rangle. \tag{82}$$

It is this remarkable ability to identify errors without learning the state in which the error has occurred that permits quantum error correction for unknown states. Once again, we see the importance of learning only the information we need to fix the error and nothing more. Any extra information we learn causes dangerous back action on the fragile quantum state we are trying to protect.

**Box 5. Measuring Joint Parity** *Note that coupling a measurement apparatus to compound operators like $Z_1 Z_2$ is somewhat unnatural because it requires a 3-body interaction (two-qubits and one object in the measurement apparatus). This operator cannot distinguish between two very different states $|\uparrow\uparrow\rangle$ and $|\downarrow\downarrow\rangle$ (and it is essential that we not be able to distinguish between them). A coupling to $Z_1 + Z_2$ would be more natural (since it only requires two-body interactions). While $Z_1 + Z_2$ cannot distinguish between $|\uparrow\downarrow\rangle$ and $|\downarrow\uparrow\rangle$, it does distinguish the two states with parallel spins from each other.*

Similarly it is critical that we not measure $Z_1$ and $Z_2$ separately and then multiply the results. This would cause state collapse in a way that would destroy the information we are trying to recover since these two operators do not individually commute with the logical operators $X_L$ and $Y_L$. From a 'computer science' point of view, one can measure $Z_1 Z_2$ by performing two CNOT operations on an ancilla qubit, the first controlled by $Z_1$ and the second by $Z_2$, followed by reading out the state of the ancilla. From the physics point of view, a measurement of $Z_1 Z_2$ can be achieved through the universal controllability of a qubit/cavity system in the strong-dispersive regime. A protocol for multi-qubit syndrome measurements in this regime was described theoretically by [34] and carried out experimentally by [35]. A somewhat different scheme was used in the first superconducting qubit quantum error correction experiment [36].

As in the classical repetition code, we are relying on the errors being rare so that simple majority voting helps rather than hurts. If there are two bit-flip errors, this procedure fails. Finally we note that there is a third stabilizer

$$S_3 = Z_1 Z_3 = S_1 S_2 , \tag{83}$$

which we have not utilized because it is simply the product of the first two stabilizers and hence does not provide any additional information. However in building a fully fault-tolerant quantum computer one has to take into account the possibility that the measurements of the error syndromes (stabilizer eigenvalues) could be faulty. If one measures all three stabilizers, one can use the redundancy of the resulting 3 classical bits of information (a 'measurement code') to help ameliorate the effects of stabilizer measurement errors [37].

---

**Exercise 9**

If the single bit-flip error probability is $\epsilon \ll 1$ and the single syndrome measurement error probability is $\delta \ll 1$, find the optimal (i.e., maximum likelihood) error recovery scheme for the 3-qubit quantum repetition code.

---

## 3.1 Coherent Errors

We now have our first taste of the fantastic power of quantum error correction. We have however glossed over some important details by assuming that either an error has occurred or it hasn't (that is, we have been assuming we are in a definite error state). At the next level of sophistication we have to recognize that we need to be able to handle the possibility of a quantum superposition of an error and no error. After all, in a system described by smoothly evolving superposition amplitudes, errors can develop continuously. Suppose for example that the correct state of the three physical qubits is

$$|\Psi_0\rangle = \alpha|000\rangle + \beta|111\rangle , \tag{84}$$

and that there is some perturbation to the Hamiltonian such that after some time there is a small rotation of the qubits. Since our code only corrects bit flips, let us assume that this rotation is around the $x$ axis. In general, all three qubits will develop different such errors simultaneously. For simplicity, let us take the special case where only the second qubit is perturbed. Then the state of the system is

$$|\Psi\rangle = e^{-i\theta_2 X_2}|\Psi_0\rangle = \cos(\theta_2)|\Psi_0\rangle - i\sin(\theta_2)X_2|\Psi_0\rangle . \tag{85}$$

What happens if we apply our error correction scheme to this state? The measurement of each stabilizer will always yield a binary result, thus illustrating the dual digital/analog nature of quantum information processing. With probability $P_0 = \cos^2(\theta_2)$, the measurement result will be $S_1 = S_2 = +1$. In this case the state collapses back to the original ideal one and the error is removed! Indeed, the experimenter has no idea whether $\theta_2$ had ever even developed a non-zero value. All she knows is that if there was an error, it is now gone. This is the essence of the Zeno effect in quantum mechanics: repeated observation can stop (sufficiently smooth) dynamical evolution. (It is also, once again, a clear illustration of Korotkov's maxim that in quantum mechanics 'You get what you see.') Rarely however (with probability $P_2 = \sin^2 \theta_2$) the measurement result will be $S_1 = S_2 = -1$ heralding the presence of an $X_2$ error. The correction protocol then proceeds as originally described above. Thus error correction still works for superpositions of no error and one error. A simple extension of this argument shows that it works for an arbitrary superposition of all four correctable error states (but of course fails for cases with 2 or 3 errors).

## 3.2   Errors due to entanglement with a bath

There remains however one more level of subtlety we have been ignoring. The above discussion assumed a classical noise source modulating the Hamiltonian parameters. However in reality, a typical source of error is that one of the physical qubits becomes entangled with its environment. We generally have no access to the bath degrees of freedom and so for all intents and purposes, we can trace out the bath and work with the reduced density matrix of the logical qubit. When we ignore the bath, the operation on the system under study becomes non-unitary. Clearly the system is generically not in a pure state. How can we possibly go from an impure state (containing the entropy of entanglement with the bath) to the desired pure (zero entropy) state? Ordinary unitary operations on the logical qubit preserve the entropy so clearly will not work. Fortunately our error correction protocol involves applying one of four possible unitary operations *conditioned on the outcome of the measurement of the stabilizers*. The wave function collapse associated with the measurement gives us just the non-unitarity we need and the error correction protocol works even in this case. Effectively we have a Maxwell demon which uses Shannon information entropy (from the measurement results) to remove an equivalent amount of von Neumann entropy from the logical qubit!

To see that the protocol still works, we generalize eqn (85) to include the bath

$$|\Psi\rangle = \sqrt{1-|\epsilon|^2}|\Psi_0\rangle|\text{Bath}_0\rangle + \epsilon X_2|\Psi_0\rangle|\text{Bath}_2\rangle. \tag{86}$$

For example, the error could be caused by the second qubit having a coupling of strength $g$ to a bath operator $\mathcal{O}_2$ of the form

$$V_2 = g X_2 \mathcal{O}_2, \tag{87}$$

acting for a short time $\epsilon\hbar/g$ so that

$$|\text{Bath}_2\rangle \approx (-i)\mathcal{O}_2|\text{Bath}_0\rangle. \tag{88}$$

Notice that once the stabilizers have been measured, then either the experimenter obtained the result $S_1 = S_2 = +1$ and the state of the system plus bath collapses to

$$|\Psi\rangle = |\Psi_0\rangle|\text{Bath}_0\rangle, \tag{89}$$

or the experimenter obtained the result $S_1 = S_2 = -1$ and the state collapses to

$$|\Psi\rangle = X_2|\Psi_0\rangle|\text{Bath}_2\rangle. \tag{90}$$

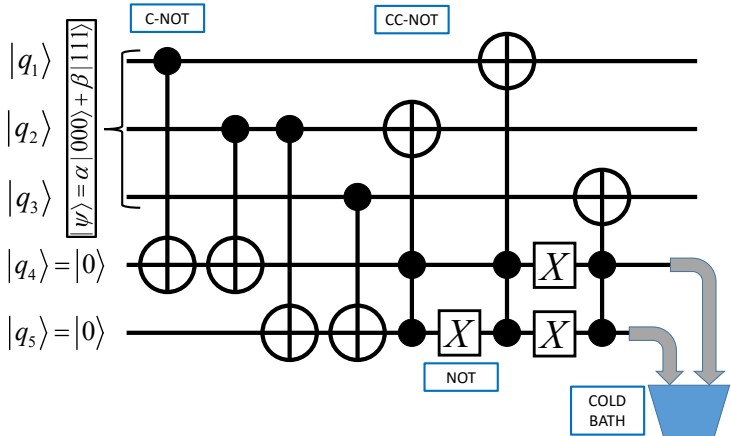

Figure 17: Example protocol for quantum error correction without measurements. The first part of the circuit (not shown) encodes the information in the topmost qubit into a 3-qubit repetition code to protect it against bit-flip errors. The second part of the circuit uses C-NOT gates to map the two error syndromes (joint parities) $S_1 = Z_1 Z_2$ and $S_2 = Z_2 Z_3$ onto the $q_4$ and $q_5$ ancillae qubits respectively. The CC-NOT (Toffoli) and the NOT gates then use the syndrome information to carry out the appropriate bit-flip operations required for error correction. The 4th and 5th ancillae qubits must then be reset prior to reuse. Any entropy associated with errors in qubits 1-3 is removed to the cold bath via this irreversible reset process.

Both results yield a product state in which the logical qubit is unentangled with the bath. Hence the algorithm can simply proceed as before and will work.

Finally, there is one more twist in this plot. We have so far described a measurement-based protocol for removing the entropy associated with errors. It is our ability to carry out unitary operations conditioned on the result of measurements that allows us to remove errors and hence entropy from the system. One downside of this technique is that measurements typically require sending signals between the room-temperature control electronics and (at least in the case of superconducting qubits) the computer sitting at the bottom of the dilution refrigerator. There are a number of technical challenges associated with this, but they can be overcome with high-speed real-time FPGA control systems [38]. There exists another route to the same goal in which purely unitary multi-qubit operations are used to move the entropy from the logical qubit to some ancillae, and then the ancillae are reset to the ground state to remove the entropy. The reset operation could consist, for example, of putting the ancillae in contact with a cold bath and allowing the qubits to spontaneously and irreversibly decay into the bath.

Fig. 17 illustrates application of this idea to the 3-qubit repetition code. The top three qubits are the repetition code. The first four CNOT gates transfer the values of the two joint parities, $Z_1 Z_2$ and $Z_2 Z_3$ into the bottom two ancillae qubits. The remaining Toffoli (Controlled-Controlled-NOT) gates effect the error correction. In the last step the ancillae must be reset to the ground state before the QEC process can repeat. The first experiment carrying out QEC in superconducting qubits used a simplified version of this protocol that required only three qubits [36]. An important recent experiment [39] successfully used an autonomous scheme to correct errors in a bosonic qubit (that stores information in superpositions of small numbers of microwave photons in a resonator).

Because the ancillae are in a mixed state with some probability to be in the excited state and some to be in the ground state, the bath ends up in a mixed state containing (or not

containing) photons resulting from the decay. Thus the entropy ends up in the bath. It is important for this process to work that the bath be cold so that the qubits always relax to the ground state and are never driven to the excited state. We could if we wished, measure the state of the bath and determine which error (if any) occurred, but in this protocol, no actions conditioned on the outcome of such a measurement are required.

From a 'computer science' point of view, one can measure $Z_1 Z_2$ by performing two CNOT operations on an ancilla qubit, the first controlled by $Z_1$ and the second by $Z_2$, followed by reading out the state of the ancilla. From the physics point of view, a measurement of $Z_1 Z_2$ can be achieved through the universal controllability of a qubit/cavity system in the strong-dispersive regime. A protocol for multi-qubit syndrome measurements in this regime was described theoretically by [34] and carried out experimentally by [35]. A somewhat different scheme was used in the first superconducting qubit quantum error correction experiment [36].

Because the unitary Hadamard gate

$$H = \frac{1}{\sqrt{2}}[X + Z], \tag{91}$$

interchanges $X$ and $Z$ operations ($HXH = Z, HZH = X$), it is straightforward to make an analogous three-qubit quantum repetition code for phase-flip errors (which generally occur more frequently than bit-flip errors). The 9-qubit Shor code [19, 20] concatenates phase-flip and bit-flip repetition codes to protect against both types of errors. It consists of a 3-qubit repetition code to protect against phase-flip errors. Each element of this repetition code is itself a logical qubit consisting of a 3-qubit repetition code to protect against bit-flip errors.

---

**Exercise 10**

The 9-qubit Shor code [19,20] has beautiful symmetry and simplicity and enjoys certain technical advantages in performing logical operations. However it is not the smallest possible code that can correct any single qubit error. Using the fact that there are four types of errors $(I, X, Y, Z)$ for each qubit, show that the smallest code that can store 1 data bit and the information on the location and error type for up to one error requires the use of $N = 5$ qubits [40–42].

---

Quantum error correction protocols based on logical qubits constructed from physical qubits were first executed experimentally in 2001 in NMR [42–44], then in 2004 in trapped ions [45], in 2005 in quantum optics [46, 47], in 2012 in superconducting qubits [36, 48] and in 2014 in diamond nitrogen-vacancy (NV) centers [49–51]. Despite considerable progress [52–57], logical qubit lifetimes are still below the break-even point (where carrying out the protocol extends rather than shortens the lifetime of the quantum information) and are in the range of $10 - 90\%$ of the lifetime of the best constituent physical qubit components.

Quantum error correction is extremely challenging to carry out in practice. The first error correction protocol to reach the break-even point was achieved by the Yale group [38]. This was done not using two-level systems as qubits but rather by storing the quantum information in the states of a harmonic oscillator (a superconducting microwave resonator containing superpositions of $0, 1, 2, \ldots$ photons). As of the time of this writing, the record for extending the lifetime of a logical qubit is held by the Devoret group at Yale using the GKP bosonic code [58] to reach a factor of $\sim 2.3$ beyond the break-even point [59]. Bosonic codes and error correction offer a number of practical advantages and will be the focus of the second half of these notes.

## 3.3 Quantum Error Correction as an Adversarial Game

It is interesting to view quantum error correction as an adversarial game between one player who is trying to keep a quantum computation running as long as possible and a noise demon who is trying to destroy the fidelity of the computation. As we saw in our discussion of classical error correction, it is never possible to strictly beat the noise demon and have the computation to be perfectly reliable for all time. It is however possible to extend the lifetime (circuit depth) exponentially in the hardware overhead (double exponentially in the concatenation level of the code). The faster and more accurate is the error correction circuit, the more rapidly you gain from concatenation.

The quantum game is more subtle. The noise demon has considerable powers–in fact has universal control allowing arbitrary quantum operations (unitary and non-unitary) on each of the individual qubits. This combined with a two-qubit entangling gate (e.g. CNOT) would be enough for the demon to carry out universal computation. We thwart this by constraining (i.e., assuming) the noise processes on individual qubits to be uncorrelated. On the other hand, we are running an algorithm on the quantum computer and carrying multi-qubit gate operations which will propagate the demon's noise throughout the system unless our QEC codes are properly designed.

Remarkably, we have found that QEC can be carried out using only CNOT and Hadamard gates plus Pauli $Z$ and $X$ gates conditioned on measurements of $Z$ Pauli operators (since we can measure $X$ Pauli operators by applying $H$ and then measuring $Z$). For the case of non-measurement based autonomous protocols, we also require qubit reset.

The limited set of operations required for QEC is a subset of the Clifford group (see Box 6). The Gottesman-Knill theorem (see Box 6) [60, 61] tells us that Clifford group operations are inadequate for universal control of a computer (despite the two-qubit entangling gate CNOT being a member of the group). Despite this fact, Clifford operations can be used to defeat a noise demon that can make random arbitrary continuous rotations of qubits. This is because, as noted previously, measurement of stabilizers (that are themselves Clifford operators) discretizes the continuous errors that develop and turns them into discrete Pauli errors.

If the noise demon has the power to make correlated errors on the qubits, then it has full computational power and defeating it is much more challenging, but not impossible if the correlations are short-ranged and sufficiently weak [62–64] that this additional power granted to the demon does not let it propagate errors too rapidly. In general, one has to use carefully designed QEC codes that can handle multiple qubit errors and beat them back before they spread throughout the system.

When we come to bosonic codes for continuous variable systems (harmonic oscillators), we will find that Gaussian operators (operations that can be carried out with time-evolution under quadratic Hamiltonians) are the analog of the Clifford group operations, in the sense that they are usually easier to implement, and they are non-universal and can be efficiently simulated classically. However, unlike the qubit case where Clifford group operations are sufficient to carry out QEC, there is a no-go theorem [65] that says that Gaussian noise channels cannot be corrected with Gaussian operations. We will therefore require more complex resources to carry out continuous-variable quantum error correction. As we will see, recent dramatic experimental progress has made those resources available within circuit QED.

**Box 6. Clifford Group and the Gottesman-Knill Theorem** *The unitary Pauli matrices $\hat{I}, X, Y, Z$ multiplied by phase factors $i^m, m \in \{0, 1, 2, 3\}$ form a group under multiplication; e.g. $XY = iZ$. This has a natural extension to strings of Pauli operators over $N$ qubits. The $N$-qubit Clifford group was defined by Gottesman to be the normalizer of the $N$-qubit Pauli group, i.e., the set of unitaries that map (via conjugation) Pauli operators into other Pauli operators. Thus $V$ is an element of the Clifford group if for every element $P$ of the Pauli group, $P' = VPV^\dagger$ is also an element of the Pauli group. Clearly $e^{i\varphi} V$ is Clifford if $V$ is Clifford, but such global phase factors can generally be ignored. Clearly the Clifford group contains the Pauli group but is larger. The entire Clifford group (including the Pauli subgroup) can be generated from only three unitary gates: controlled-NOT, Hadamard (H) and S where*

$$\text{CNOT}_{12} = \frac{\hat{I}_1 + Z_1}{2}\hat{I}_2 + \frac{\hat{I}_1 - Z_1}{2}X_2, \tag{92}$$

$$H = \frac{1}{\sqrt{2}}[X + Z] = \frac{1}{\sqrt{2}}\begin{pmatrix} +1 & +1 \\ +1, & -1 \end{pmatrix}, \tag{93}$$

$$S = \sqrt{Z} = \begin{pmatrix} +1 & 0 \\ 0, & +i \end{pmatrix}. \tag{94}$$

*This small gate set is very powerful and can generate highly entangled $N$-qubit states. Hadamard creates superpositions*

$$H|0\rangle = |+\rangle \equiv \frac{1}{\sqrt{2}}[|0\rangle + |1\rangle]), \tag{95}$$

*and, when combined with a CNOT, can create entangled Bell states*

$$\text{CNOT}_{12}H_1|00\rangle = \frac{1}{\sqrt{2}}[|00\rangle + |11\rangle]. \tag{96}$$

*The Gottesman-Knill theorem tells us that surprisingly, despite the fact that the Clifford gates can generate huge entangled superposition states of $N$ qubits starting from a single state $|0\rangle^{\otimes N}$, such circuits are easy to simulate classically [61]. To see why this is so, notice that the starting state can be uniquely defined by its stabilizers (the list of operators for which it is a +1 eigenstate). For example the stabilizers of $|0\rangle^{\otimes N}$ are simply $\{Z_1, Z_2, \dots, Z_N\}$. Applying H to qubit $m$ produces a superposition state, but we do not need to keep track of the superposition amplitudes, we need only update the list of stabilizers by making the replacement $Z_m \to X_m$. Similarly, if we use H and CNOT to create a Bell pair from qubits $\ell, m$, we replace the pair of stabilizers $\{Z_\ell, Z_m\} \to \{X_\ell X_m, Z_\ell Z_m\}$. Single-qubit stabilizers are replaced by two-qubit stabilizers, but the total number of stabilizers remains the same. We can efficiently simulate Clifford circuits by simply updating the list of Cliffords. It can also be shown that we can efficiently compute the expectation values of any physical observables by examining the updated list of stabilizers.*

*Because they can be efficiently simulated classically, Clifford circuits plus computational basis measurements do not capture the full power of universal quantum computers. One hint as to why this is so, is that you cannot use such circuits to detect violations of the Bell inequalities. The latter requires being able to rotate detectors (or equivalently states) by $\pi/4$, not simply $\pi/2$. For this we need a non-Clifford gate such as the T gate*

$$T = \sqrt{S} \equiv \begin{pmatrix} 1 & 0 \\ 0, & e^{i\pi/4} \end{pmatrix}, \tag{97}$$

*which is needed to create what is colloquially known in the field as 'magic.' Once we include the T gate, the group size becomes infinite and we can create an arbitrarily good approximation to any state in the $N$-qubit Hilbert space [19].*

# 4 Quantum Operations and Kraus Representations of Error Channels

As mentioned previously, errors that entangle portions of the quantum computer with its environment appear to be non-unitary quantum operations if we do not have access to the environment. This means that the computer is no longer in a so-called 'pure' state but must be described as a statistical mixture of different pure states. In general, the state of a quantum system (or at least our knowledge of the quantum state) must be described not by a ray in Hilbert space but rather by a Hermitian density matrix

$$\hat{\rho} = \sum_j p_j |\psi_j\rangle\langle\psi_j|, \tag{98}$$

representing a statistical mixture of different states $|\psi_j\rangle$ that occur with probabilities $p_j$ obeying

$$\sum_j p_j = 1. \tag{99}$$

It is important to note that the $|\psi_j\rangle$ are normalized, but *not* necessarily orthogonal.

---

**Exercise 11**

Prove that

$$\mathrm{Tr}\,\hat{\rho} = \sum_j p_j, \tag{100}$$

even when the states $|\psi_j\rangle$ are not orthogonal.

**Exercise 12**

Prove that $\hat{\rho}$ is positive semi-definite. That is, show that

$$\langle\psi|\hat{\rho}|\psi\rangle \geq 0, \forall\,|\psi\rangle. \tag{101}$$

---

The density matrix gives us all the information needed to compute the expectation value of any observable. Quantum errors associated with unitary coupling to unobserved degrees of freedom (i.e., a bath) are non-unitary 'quantum operations' [19]. As noted in Sec. 3.2, quantum error correction can still work in this case. Here we pursue this topic in more detail and generality. Non-unitary quantum operations create impure states which must be described by a density matrix. Before we further explore the formalism to correct such quantum errors we must first learn how they cause the density matrix to evolve. Because quantum operations are the most general transformations (unitary or non-unitary) that can be applied to a quantum system, we will find that both errors and error recovery are quantum operations.

The density matrix $\hat{\rho}$ is a semi-positive-definite Hermitian matrix with real non-negative eigenvalues and unit trace. Unitary evolution under the Hamiltonian and non-unitary dissipative or error processes occurring during some time interval $\Delta t$ must preserve these properties. Such transformations of the density matrix are called completely-positive trace-preserving (CPTP) maps. In the so-called Kraus representation, a CPTP map is given by

$$\hat{\rho}' = \hat{\rho}(t + \Delta t) = \sum_\ell \hat{K}_\ell \hat{\rho}(t)\hat{K}_\ell^\dagger. \tag{102}$$

As we show later in eqn (227), knowledge of the Kraus representation of this CPTP map for small time intervals gives us the Lindblad master equation.

Working in the interaction picture of the nominal Hamiltonian, the set of Kraus operators $\mathcal{E} = \{\hat{K}_0, \hat{K}_1, \hat{K}_2, \ldots\}$ describe the transformation of the state under various errors resulting from Hamiltonian parameter deviations from their ideal values or from coupling to a bath. Even if some (or all) of the Kraus operators are non-unitary, in order for the trace to be preserved, it must still be true that

$$\sum_\ell \hat{K}_\ell^\dagger \hat{K}_\ell = \hat{I}, \tag{103}$$

where $\hat{I}$ is the identity. This follows from the invariance of the trace under cyclic permutations of the operators.

Now consider an arbitrary vector $|\Phi\rangle$ in the Hilbert space and define a set of 'daughter' states via $|\Phi_\ell\rangle = K_\ell^\dagger |\Phi\rangle$. Then we see that for all $|\Phi\rangle$

$$\sum_\ell \langle\Phi|\hat{K}_\ell \hat{\rho}(t) \hat{K}_\ell^\dagger|\Phi\rangle = \sum_\ell \langle\Phi_\ell|\hat{\rho}|\Phi_\ell\rangle \geq 0. \tag{104}$$

Since this is true for all possible vectors $|\Phi\rangle$, the Kraus map is thus completely positive as well as trace preserving, as required.

As an example, consider a two-level qubit. Suppose that during the time interval $\Delta t$ it has probability $p_z$ to suffer a phase flip, and probability $p_x$ to suffer a bit flip. (Notice that these are both unitary errors.) The set of error operators is[8]

$$\hat{K}_x = \sqrt{p_x}\,\sigma_x, \tag{105}$$

$$\hat{K}_z = \sqrt{p_z}\,\sigma_z, \tag{106}$$

$$\hat{K}_0 = \sqrt{1 - p_x - p_z}\,\hat{I}, \tag{107}$$

where $\hat{K}_0$ describes the case of no error and is necessary to include so that eqn (103) is satisfied. Notice the important fact that

$$\langle\hat{K}_k^\dagger \hat{K}_k\rangle = \mathrm{Tr}\left[\hat{K}_k \hat{\rho}(t) \hat{K}_k^\dagger\right] = p_k \tag{108}$$

gives the probability that the $k$th error will occur. Equation (103) guarantees that the probabilities add up to unity.

As a second example, suppose that during the time interval $dt$ a qubit has probability

$$p^- = \gamma\,dt, \tag{109}$$

for the excited state $|e\rangle$ to decay to the ground state $|g\rangle$. This irreversible decay is an example of a non-unitary error, and is referred to as the qubit amplitude damping channel [66]. The decay could for example be the result of spontaneous emission of a photon into a cold bath. Once the photon leaves, there are no thermal photons available to be absorbed and reverse the decay. The set of error operators is

$$\hat{K}_- = \sqrt{p^-}\,\sigma^- = \sqrt{p^-}|g\rangle\langle e|, \tag{110}$$

$$\hat{K}_0 = \sqrt{1 - p^-}|e\rangle\langle e| + |g\rangle\langle g|. \tag{111}$$

This follows from eqn (103) and the fact that $\sigma^+\sigma^- = |e\rangle\langle e|$. Under the action of this error set, the density matrix $\hat{\rho}$ transforms to

$$\hat{\rho}' = \{|e\rangle[(1 - p^-)\rho_{ee}]\langle e|\} + \{|g\rangle[p^-\rho_{ee} + \rho_{gg}]\langle g|\} + \sqrt{1 - p^-}\{|e\rangle\rho_{eg}\langle g| + |g\rangle\rho_{ge}\langle e|\}. \tag{112}$$

---

[8]We will see shortly that the Kraus operators are not unique and any transformation among them corresponding to a unitary 'rotation' produces fully equivalent description of the dynamics.

Notice that we have defined $p^-$ as the probability that a downward jump will occur if the qubit is in the excited state. The actual probability that a jump will occur is given by

$$\bar{p}^- = \langle \hat{K}_-^\dagger \hat{K}_- \rangle = p^- \rho_{ee}\,. \tag{113}$$

If the time interval is short so that the jump probability is small ($p^- \ll 1$), then $\sqrt{1-p^-} \approx 1 - \frac{p^-}{2}$ and

$$\hat{K}_0 \approx \hat{I} - \frac{p^-}{2}|e\rangle\langle e|\,. \tag{114}$$

In this limit, eqn (112) yields

$$\langle g|\hat{\rho}'|e\rangle \approx \left[1 - \frac{p^-}{2}\right]\langle g|\hat{\rho}|e\rangle\,, \tag{115}$$

$$\langle e|\hat{\rho}'|e\rangle = \left[1 - p^-\right]\langle e|\hat{\rho}|e\rangle\,, \tag{116}$$

showing that the off-diagonal elements of the density matrix decay at half the rate of the diagonal elements, essentially because they represent amplitudes instead of probabilities. This is the origin of the factor of 2 in the standard NMR expression for the decay of transverse spin components

$$\frac{1}{T_2} = \frac{1}{2T_1} + \frac{1}{T_\varphi}\,, \tag{117}$$

where $1/T_\varphi$ is the intrinsic dephasing rate (associated with the jump operator $\sigma^z$ and not included in our example above). Roughly speaking, the off-diagonal elements decay like amplitudes and the diagonal elements decay like probabilities (squares of amplitudes).

One important feature of the Kraus representation to keep in mind is that it is *not* unique. Suppose that we define new Kraus operators via

$$\hat{F}_j = \sum_k U_{jk} \hat{K}_k\,, \tag{118}$$

where $U$ is a unitary transformation. Then it is straightforward to show that

$$\sum_j \hat{K}_j \hat{\rho} \hat{K}_j^\dagger = \sum_j \hat{F}_j \hat{\rho} \hat{F}_j^\dagger\,. \tag{119}$$

This fact will prove useful when we study the Knill-Laflamme conditions which quantum codes need to satisfy in order to be error correctable.

---

**Exercise 13**

Construct a pair of Kraus operators that define the qubit reset operation. Reset is a non-unitary operation that maps every vector in the Hilbert space to the ground state, $|g\rangle$. Be sure that your result is trace preserving.

**Exercise 14**

Consider the initial state described by a Bloch polarization vector $\vec{S}$ living either on the surface or in the interior of the Bloch sphere

$$\hat{\rho} = \frac{1}{2}\{\hat{I} + \vec{S} \cdot \vec{\sigma}\}\,. \tag{120}$$

Prove that $\langle \vec{\sigma} \rangle = \vec{S}$. Find the new Bloch vector after application of the error set given in eqns (105-107).

---

---

**Exercise 15**

Prove eqn (119). Hint: Be very careful evaluating $\left\{U\hat{K}\right\}^{\dagger}$.

---

## 4.1 Quantum Codes and the Knill-Laflamme Condition

The generic task of quantum error correction is to find two logical codewords (two orthogonal quantum states forming a logical qubit) $|W_\sigma\rangle$, where $\sigma =\uparrow,\downarrow$, embedded in a large Hilbert space. The codewords are required to be robust such that if any one of the errors $\hat{K}_k$ in the set of Kraus operators $\mathcal{E}$ occurs, no quantum information is lost and any quantum superposition of the logical codewords can be faithfully recovered. (Note that some of the Kraus operators may correspond to errors on multiple physical qubits within the logical qubit.)

In order for the code to be quantum error correctable, the logical codewords must satisfy the quantum error correction criteria [19, 67], known also as the Knill-Laflamme conditions [40, 68]

$$\langle W_\sigma|\hat{K}_\ell^{\dagger}\hat{K}_k|W_{\sigma'}\rangle = \alpha_{\ell k}\delta_{\sigma\sigma'}, \tag{121}$$

for all $\hat{K}_\ell, \hat{K}_k$ in $\mathcal{E}$, where $\alpha_{\ell k}$ are entries of a Hermitian matrix that is required to be independent of the logical words (i.e., not depend on $\sigma, \sigma'$). This is a general condition that applies to all possible codes, not just stabilizer codes.

The independence of the entries $\alpha_{\ell k}$ from the logical codewords and the structure of the non-diagonal entries guarantee that the different errors are distinguishable and correctable. To understand the Knill-Laflamme conditions let us first consider the Kronecker delta. This guarantees that the states

$$\hat{K}_k|W_\uparrow\rangle, \quad \hat{K}_\ell|W_\downarrow\rangle \tag{122}$$

are orthogonal. Without that we would not be able to diagnose whether the system had been in state $|W_\downarrow\rangle$ and suffered error $\hat{K}_k$ or had been in state $|W_\uparrow\rangle$ and suffered error $\hat{K}_\ell$, because the two scenarios lead to states which are not orthogonal to each other and therefore not distinguishable. This means that the *distance d* of the code must be large enough that $\lfloor(d-1)/2\rfloor$ exceeds the weight of the worst (i.e., highest-weight) error in the error set; that is, $2\,\mathrm{maxweight}(\hat{K}_k) \le (d-1)$.

The fact that the matrix $\alpha$ is not necessarily diagonal, means that we cannot tell which error occurred. However this problem can be resolved by using the fact that the Kraus representation is not unique. Applying the transformation in eqn (118) with an appropriately chosen unitary, we can diagonalize the Hermitian matrix $\alpha$ to obtain

$$\langle W_\sigma|\hat{F}_\ell^{\dagger}\hat{F}_k|W_{\sigma'}\rangle = \beta_k\delta_{\ell k}\delta_{\sigma\sigma'}, \tag{123}$$

where $\beta_k$ is the $k$th (real and positive) eigenvalue of $\alpha$. Note that

$$P_k^{\downarrow} = \langle W_\downarrow|\hat{F}_k^{\dagger}\hat{F}_k|W_\downarrow\rangle, \tag{124}$$

$$P_k^{\uparrow} = \langle W_\uparrow|\hat{F}_k^{\dagger}\hat{F}_k|W_\uparrow\rangle \tag{125}$$

are the probabilities that the $k$th error (in the new error basis $\hat{F}_k$) will occur for each codeword. The fact that these probabilities are the same for both codewords

$$P_k = P_k^{\uparrow} = P_k^{\downarrow} = \beta_k, \tag{126}$$

that is, that $\alpha$ and hence $\beta$ is independent of $\sigma, \sigma'$, is essential. Otherwise there would be a quantum back action that would distort any linear superposition of the two codewords.

We can think of this distortion as resulting from the Bayesian update of our knowledge of the system state when we see that a particular error has occurred. If that error is more likely

in one codeword than the other, Bayes rule tells us we have to update our prior probability distribution to put more weight on that codeword. To see this how this classical language translates into quantum mechanical back action, consider a general superposition state within the code space

$$|\psi\rangle = \lambda_\downarrow |W_\downarrow\rangle + \lambda_\uparrow |W_\uparrow\rangle \,. \tag{127}$$

If error $\hat{F}_k$ occurs, the resulting normalized state is

$$|\psi_k\rangle = \frac{\hat{F}_k |\psi\rangle}{\langle\psi|\hat{F}_k^\dagger \hat{F}_k|\psi\rangle^{\frac{1}{2}}} \,. \tag{128}$$

Before proceeding further, we note as a side remark that this means that the density matrix conditioned on our knowledge (resulting say from some measurement of the system or the environment) that error $k$ has occurred is

$$\hat{\rho}_k = |\psi_k\rangle\langle\psi_k| = \frac{\hat{F}_k |\psi\rangle\langle\psi|\hat{F}_k^\dagger}{\langle\psi|\hat{F}_k^\dagger \hat{F}_k|\psi\rangle} \,. \tag{129}$$

Notice that the normalization factor in the denominator is precisely the probability $P_k$ that error $k$ will occur. Hence if we ensemble average over all possible errors, weighted by their probability of occurrence, we obtain the unconditional density matrix

$$\hat{\rho}' = \sum_k P_k \hat{\rho}_k = \sum_k \hat{F}_k \hat{\rho} \hat{F}_k^\dagger = \sum_k \hat{K}_k \hat{\rho} \hat{K}_k^\dagger \,, \tag{130}$$

in agreement with eqn (102).

Returning to the task at hand and using eqns (123-125), we find

$$|\psi_k\rangle = \sqrt{\frac{P_k^\downarrow}{Z}} \lambda_\downarrow \left\{ \frac{1}{\sqrt{P_k^\downarrow}} \hat{F}_k |W_\downarrow\rangle \right\} + \sqrt{\frac{P_k^\uparrow}{Z}} \lambda_\uparrow \left\{ \frac{1}{\sqrt{P_k^\uparrow}} \hat{F}_k |W_\uparrow\rangle \right\} \,. \tag{131}$$

Each state vector in the curly brackets is properly normalized and the two are orthogonal. The normalization constant is given by

$$Z = |\lambda_\downarrow|^2 P_k^\downarrow + |\lambda_\uparrow|^2 P_k^\uparrow \,. \tag{132}$$

If and only if $P_k^\downarrow = P_k^\uparrow = P_k$, so that the error probability is the same for the two codewords, do we obtain

$$\sqrt{\frac{P_k^\downarrow}{Z}} = \sqrt{\frac{P_k^\uparrow}{Z}} = 1 \,, \tag{133}$$

so that

$$|\psi_k\rangle = \lambda_\downarrow \left\{ \frac{1}{\sqrt{P_k}} \hat{F}_k |W_\downarrow\rangle \right\} + \lambda_\uparrow \left\{ \frac{1}{\sqrt{P_k}} \hat{F}_k |W_\uparrow\rangle \right\} \,. \tag{134}$$

If the probability $P_k$ had depended on the codeword, the relative coefficients in this expression would have changed, thereby distorting the quantum information stored in the (unknown) coefficients $\lambda_{\uparrow,\downarrow}$. In this case it is impossible to find a unitary error-recovery operation that is independent of the values of the unknown coefficients $\lambda_{\uparrow,\downarrow}$.

We can think of the distortion as being created by a partial measurement of the quantum state by the environment that partially collapses the state towards the codeword that is more likely to have that error. We can gain intuition on this by thinking of the extreme case where

error $\hat{F}_k$ can occur only for one of the codewords, say $|W_\downarrow\rangle$. (That is, $\langle W_\downarrow|\hat{F}_k^\dagger F_k|W_\downarrow\rangle = P_k^\downarrow$, $\langle W_\uparrow|\hat{F}_k^\dagger F_k|W_\uparrow\rangle = 0$.) Then if that error occurs, the state must collapse completely onto the only codeword that is allowed to have that error

$$|\psi_k\rangle = \frac{1}{\sqrt{P_k^\downarrow}}\hat{F}_k|W_\downarrow\rangle, \tag{135}$$

thereby destroying any superposition that may have existed.

If the Knill-Laflamme conditions are satisfied, and if we know that error $\hat{F}_k$ has occurred, we can apply a particular unitary operation based on our knowledge of which error occurred to recover from the error. The unitary must perform the state transfers

$$U^{(k)}\left\{\frac{1}{\sqrt{P_k}}\hat{F}_k|W_\downarrow\rangle\right\} = |W_\downarrow\rangle, \tag{136}$$

$$U^{(k)}\left\{\frac{1}{\sqrt{P_k}}\hat{F}_k|W_\uparrow\rangle\right\} = |W_\uparrow\rangle. \tag{137}$$

Because the initial error states (in the curly brackets) are orthogonal to each other and the restored codewords are orthogonal to each other, the angle between the state vectors is preserved and there must exist a unitary state transfer operation that works. Such a unitary has the form

$$U^{(k)} = \frac{1}{\sqrt{P_k}}\left\{|W_\downarrow\rangle\langle W_\downarrow|\hat{F}_k^\dagger + |W_\uparrow\rangle\langle W_\uparrow|\hat{F}_k^\dagger\right\} + U^{(k)}_{\text{completion}}, \tag{138}$$

where the last term is a matrix which determines how states other than the two error states are mapped into each other. The details of this (non-unique) 'unitary completion' are largely irrelevant because we only apply this unitary to the two particular error states that result from $F_k$ acting on the two codewords.

We explore the error recovery operation more formally in the next subsection.

---

### Exercise 16

Prove that the 3-qubit repetition code satisfies the Knill-Laflamme conditions for single bit-flip errors $X$, whose Kraus operators are given by eqns (105-107) (with $p_z = 0$), but do not satisfy them for amplitude damping ($T_1$ decay) errors, whose Kraus operators are given in eqns (110-111).

---

## 4.2 Error Recovery Operation

In the previous section, we found that if we know which error has occurred, we can recover from that error by applying a unitary recovery operation that is particular to that error. In a *non-degenerate*[9] stabilizer code one can determine which error occurred simply by measuring the full set of stabilizers. In this section we show that it is possible to know which error occurred even for non-stabilizer codes. To formally construct the full error recovery operation based on the Knill-Laflamme condition, let us define the projector onto the code space

$$\hat{P}_c = \sum_\sigma |W_\sigma\rangle\langle W_\sigma|. \tag{139}$$

---

[9]In a *degenerate* code, more than one error leads to the same syndrome. However these errors are equivalent up to multiplication by one of the stabilizers. Hence one can choose to correct any of the errors in the equivalence class and fully recover the original state up to a possible overall minus sign due to the stabilizer multiplication.

Because the codewords are orthogonal, this is a proper projector obeying $P_c^2 = P_c$. It is convenient to express the Knill-Laflamme conditions in terms of this projector

$$\hat{P}_c \hat{F}_\ell^\dagger \hat{F}_k \hat{P}_c = \beta_k \delta_{\ell k} \hat{P}_c. \tag{140}$$

Error recovery is also a quantum operation given by

$$\mathcal{R}\hat{\rho}' = \sum_\ell \hat{R}_\ell \hat{\rho}' \hat{R}_\ell^\dagger, \tag{141}$$

with

$$\hat{R}_\ell = \frac{1}{\sqrt{\beta_\ell}} \hat{P}_c \hat{F}_\ell^\dagger. \tag{142}$$

Substitution of this into eqn (130) yields

$$\mathcal{R}\hat{\rho}' = \sum_{\ell,k} \frac{1}{\beta_\ell} \hat{P}_c \hat{F}_\ell^\dagger \hat{F}_k \hat{P}_c \hat{\rho} \hat{P}_c \hat{F}_k^\dagger \hat{F}_\ell \hat{P}_c, \tag{143}$$

where we have made the assumption that there are no errors initially so that

$$\hat{\rho} = \hat{P}_c \hat{\rho} \hat{P}_c. \tag{144}$$

Applying the Knill-Laflamme condition from eqn (140) yields

$$\mathcal{R}\hat{\rho}' = \left(\sum_\ell \beta_\ell\right) \hat{P}_c \hat{\rho} \hat{P}_c = \hat{\rho}. \tag{145}$$

The above result proves that the Knill-Laflamme conditions are sufficient for a recovery operation to exist. There remains however the question of how we realize it in practice. For this purpose it is useful to define projectors onto the error spaces associated with each Kraus operator

$$\hat{P}_m = \frac{1}{\beta_m} \hat{F}_m \hat{P}_c \hat{F}_m^\dagger. \tag{146}$$

---

**Exercise 17**

Use eqn (123) to prove that the projectors onto the error subspaces obey $\hat{P}_m \hat{P}_n = \hat{P}_m \delta_{mn}$.

---

Because the projection operators $\hat{P}_m$ are all mutually commuting and are hermitian, they can be simultaneously measured. Because they are projectors, their eigenvalues are either 0 or 1. Because they project onto mutually orthogonal subspaces (see Ex. 17), no more than one can have a measured value of 1 (in order to guarantee $\hat{P}_m \hat{P}_n = 0$ for $m \neq n$). These projectors span the space of all possible error states,

$$\sum_m \hat{P}_m = \hat{I}_\mathcal{E}, \tag{147}$$

where $\hat{I}_\mathcal{E}$ is (effectively) the identity. That is, it is the identity within the space of all the errors $\hat{F}_k$ acting on the codewords. Since $\hat{I}_\mathcal{E}$ has eigenvalue 1, then every measurement of the set $\{\hat{P}_m; m = 0, 1, 2, \dots\}$ will yield exactly one non-zero value. This non-zero value uniquely flags

which error state has occurred. One could even imagine (ignoring experimental difficulties) measuring the operator

$$C = \sum_m m \hat{P}_m \,. \tag{148}$$

This returns integer eigenvalues which index the error number.

Once the error label has been determined to be (say) $k$, one applies the conditional unitary $U^{(k)}$ in eqns (136-137) to the error state in eqn (134) to recover from the error and obtain the original state in eqn (127)

$$|\psi\rangle = U^{(k)}|\psi_k\rangle \,. \tag{149}$$

Here however we want to more formally discuss this using the polar decomposition described in [19]. The polar decomposition of a complex number into its magnitude and phase

$$z = re^{i\theta} = \sqrt{z^*z}\, e^{i\theta} \,. \tag{150}$$

For an operator, it turns out there is an analogous result

$$\hat{R}_\ell = \sqrt{R_\ell R_\ell^\dagger}\, \hat{U}_\ell \,, \tag{151}$$

where $\hat{U}_\ell$ is a (not necessarily unique) unitary operator. It is straightforward to verify that this expression correctly reproduces $\hat{R}_\ell \hat{R}_\ell^\dagger$. From eqn (142) we have that

$$R_\ell R_\ell^\dagger = \frac{1}{\beta_\ell} \hat{P}_\text{c} \hat{F}_\ell \hat{F}_\ell^\dagger \hat{P}_\text{c} = \hat{P}_\text{c} \,. \tag{152}$$

Thus

$$R_\ell = \sqrt{\hat{P}_\text{c} \hat{P}_\text{c}}\, \hat{U}_\ell = \hat{P}_\text{c} \hat{U}_\ell \,. \tag{153}$$

Now notice that if we reverse the order of the operators we have

$$R_\ell^\dagger R_\ell = \hat{U}_\ell^\dagger \hat{P}_\text{c} \hat{U}_\ell \,, \tag{154}$$

and we also have that the same quantity is the projector onto the $\ell$th error subspace

$$R_\ell^\dagger R_\ell = \frac{1}{\beta_\ell} \hat{F}_\ell \hat{P}_\text{c} \hat{F}_\ell^\dagger = \hat{P}_\ell \,. \tag{155}$$

Combining these results we have

$$\hat{U}_\ell \hat{P}_\ell \hat{U}_\ell^\dagger = \hat{P}_\text{c} \,, \tag{156}$$

proving that $U_\ell$ is indeed the desired recovery unitary that maps the error subspace back to the code space.

The above results give a formal constructive proof that quantum error correction is possible if the code satisfies the Knill-Laflamme conditions. Nielsen and Chuang [19] give a proof that in addition to being sufficient, the Knill-Laflamme conditions are also necessary. The recovery operation constructed here is not necessarily the most convenient from an experimental point of view. This is particularly true for the case of stabilizer codes where one only needs to measure $\log_2 N_\text{error}$ stabilizer values (error syndromes) to identify which of $N_\text{error}$ errors has occurred. In the next subsection we will illustrate this approach for the case of the quantum repetition code.

## 4.3 Recovery Operation for the Quantum Repetition Code

Recall from our previous discussion that the 3-qubit repetition code can protect against (single) physical bit flip errors by using a 'majority rule' to determine which bit (if any) flipped. Our error model will be that each qubit can independently flip with probability $p_x$. If $p_x$ is sufficiently small, the possibility that two physical qubits flip is small, and the possibility that all three flip is even smaller. As we saw in the study of the classical repetition code, these rare events do eventually cause the code to fail. For simplicity here, we will neglect these low probability possibilities and take the error set to be the four leading-order Kraus operators describing no more than 1 bit flip

$$\hat{K}_0 = \sqrt{1-3p_x}\,\hat{I}\,, \tag{157}$$

$$\hat{K}_j = \sqrt{p_x}X_j\,. \tag{158}$$

Here $j = 1,2,3$ labels which qubit flipped, and $j = 0$ refers to the case where no spins have flipped. To lowest order in $p_x$ the probability of an error is $3p_x$ which is three times worse than for a single qubit. As noted earlier, our error correction scheme has to overcome this factor of three just to break even!

Let us check the Knill-Laflamme condition. Using the logical codewords from eqns (72-73) one can readily verify that for this particular example, the matrix $\alpha$ in eqn (121) is diagonal and independent of the codewords.

$$\alpha = \begin{pmatrix} 1-3p_x & 0 & 0 & 0 \\ 0 & p_x & 0 & 0 \\ 0 & 0 & p_x & 0 \\ 0 & 0 & 0 & p_x \end{pmatrix}. \tag{159}$$

Hence the Knill-Laflamme conditions are satisfied. This result is true only to lowest order in $p_x$ because we neglected the possibility of two or three bit flips.

Instead of defining four error subspace projectors as we did above, we will define projectors onto the +1 eigenstates of the stabilizers that were defined in eqns (76-77)

$$\Pi_1 = \frac{1}{2}\left[1 + Z_1 Z_2\right], \tag{160}$$

$$\Pi_2 = \frac{1}{2}\left[1 + Z_2 Z_3\right]. \tag{161}$$

The recovery operation then consists of four Kraus operators

$$R_0 = \Pi_1 \Pi_2\,, \tag{162}$$
$$R_1 = X_1[1-\Pi_1]\Pi_2\,, \tag{163}$$
$$R_2 = X_2[1-\Pi_1][1-\Pi_2]\,, \tag{164}$$
$$R_3 = X_3\Pi_1[1-\Pi_2]\,. \tag{165}$$

---

### Exercise 18

Find the matrix $\alpha$ for the 3-qubit repetition code to all orders in $p_x$ and show that beyond linear order it does not satisfy the Knill-Laflamme conditions.

---

**Exercise 19**

Verify that the four Kraus operators in the above recovery define a trace-preserving operation.

This section has given a brief introduction to quantum operations and the Kraus representation of CPTP maps for qubit systems. The Knill-Laflamme conditions are defined for an 'exact' QEC code: they assume a fixed error set and give the necessary and sufficient conditions under which this error set can be exactly corrected. This however is not a physically realistic error model. In general for any physical system (whether bosonic or qubit based), errors occur continuously in time and so the number of errors that can occur in a finite time interval is effectively unbounded. However, the probability of more than a small number of errors occurring in a short time interval (which physically cannot be less than the time that it takes to run the QEC protocol) is typically small. The probability that an error can occur that is outside the correctable error set (e.g., two bit flips in the 3-bit repetition code) is thus always finite. To summarize, an 'exact' QEC code does not correct all errors, but rather all errors in a fixed set, typically defined to include errors up to a certain specified order in a small parameter related to the time delay between error-correction cycles.

With this fact in mind, it turns out that one can slightly relax the Knill-Laflamme conditions by only requiring that they be satisfied up to a specified order in the small parameter related to the time delay between error-correction cycles. This gives us new freedom to find so-called *approximate QEC codes* which are still useful, despite not being 'exact' [66]. As an illustration of this concept, we will consider in the next section an approximate error correction code for qubit amplitude damping. This will be helpful background for our later study of bosonic codes since the dominant error for a bosonic system realized with photons (e.g. optical photons in fibers or microwave photons stored in a superconducting resonator) is amplitude damping–excitation loss. We extend our study of Kraus operators to the case of continuous variable systems when we study bosonic quantum error correction codes.

## 4.4 Four-qubit Amplitude-Damping Code

The qubit amplitude-damping channel defined above in eqns (110-111) can be perfectly corrected against single-qubit errors using the minimal $N = 5$ qubit code [40–42], but also admits an interesting 4-qubit code [66] that approximately satisfies the Knill-Laflamme conditions in eqn (121). The two logical codewords are

$$|0_{\mathrm{L}}\rangle = \frac{1}{\sqrt{2}}[|0000\rangle + |1111\rangle], \tag{166}$$

$$|1_{\mathrm{L}}\rangle = \frac{1}{\sqrt{2}}[|1100\rangle + |0011\rangle]. \tag{167}$$

Note that both codewords have mean excitation number equal to two and so seem equally likely to suffer an excitation loss. We will encounter this fact again when we study the simplest 'binomial code' for bosonic error correction. In point of fact, for the qubit case, the excitation loss probabilities are only approximately equal. To see this, consider the case where the 3rd qubit (counting from left to right) decays. Using eqns (110-111) the corresponding 4-qubit Kraus operator is

$$\hat{E}_{-}^{(3)} = \hat{K}_0^{(1)} \hat{K}_0^{(2)} \hat{K}_-^{(3)} \hat{K}_0^{(4)}. \tag{168}$$

Applying this to the codewords gives

$$\hat{E}_-^{(3)}|0_{\mathrm{L}}\rangle = \frac{1}{\sqrt{2}}\sqrt{(1-p^-)}^3\sqrt{p^-}|1101\rangle, \tag{169}$$

$$\hat{E}_-^{(3)}|1_{\mathrm{L}}\rangle = \frac{1}{\sqrt{2}}\sqrt{(1-p^-)}\sqrt{p^-}|0001\rangle. \tag{170}$$

The norms of these states are the probability of (only) the 3rd qubit decaying. We see that the norms are equal only to first order in $p^-$. We also see that the two error words are orthogonal, so that part of the Knill-Laflamme condition is exactly satisfied. However, the $\alpha$ matrix in eqn (121) does depend on the codeword if we take into account terms beyond lowest order in $p^-$.

We have so far analyzed the case where it is the 3rd qubit that decays. The remaining 3 possibilities for single-qubit decay have the same form. All of the error words are mutually orthogonal and also orthogonal to the original codewords, so to lowest order in $p^-$ the Knill Laflamme conditions are satisfied.

Consider now the case that none of the qubits decay. Again using eqns (110-111), this is described by the 4-qubit Kraus operator

$$\hat{E}_0 = \hat{K}_0^{(1)}\hat{K}_0^{(2)}\hat{K}_0^{(3)}\hat{K}_0^{(4)}, \tag{171}$$

whose action on the codewords is

$$\hat{E}_0|0_{\mathrm{L}}\rangle = \frac{1}{\sqrt{2}}\left[|0000\rangle + (1-p^-)^2|1111\rangle\right], \tag{172}$$

$$\hat{E}_0|1_{\mathrm{L}}\rangle = (1-p^-)\frac{1}{\sqrt{2}}\left[|1100\rangle + |0011\rangle\right]. \tag{173}$$

The norms of these two states are

$$\langle 0_{\mathrm{L}}|\hat{E}_0^\dagger\hat{E}_0|0_{\mathrm{L}}\rangle = \frac{1}{2}\left[1 + (1-p^-)^4\right], \tag{174}$$

$$\langle 1_{\mathrm{L}}|\hat{E}_0^\dagger\hat{E}_0|1_{\mathrm{L}}\rangle = (1-p^-)^2. \tag{175}$$

Again, these are equal only to lowest order in $p^-$.

To understand the recovery unitaries, it is useful to rewrite eqn (172) as

$$\hat{E}_0|0_{\mathrm{L}}\rangle = \frac{1}{2}[1+(1-p^-)^2]|0_{\mathrm{L}}\rangle + \frac{1}{2}[1-(1-p^-)^2]|0_{\mathrm{L}}'\rangle, \tag{176}$$

where

$$|0_{\mathrm{L}}'\rangle = \frac{1}{\sqrt{2}}\left[|0000\rangle - |1111\rangle\right]. \tag{177}$$

Taking out a factor $(1-p^-)$ to match the one in eqn (173) and expanding everything else to lowest order in $p^-$ yields

$$\hat{E}_0|0_{\mathrm{L}}\rangle \approx (1-p^-)\left\{|0_{\mathrm{L}}\rangle + p^-|0_{\mathrm{L}}'\rangle\right\} \tag{178}$$

$$\approx (1-p^-)\left\{\cos\frac{\theta}{2}|0_{\mathrm{L}}\rangle + \sin\frac{\theta}{2}|0_{\mathrm{L}}'\rangle\right\}, \tag{179}$$

where $\theta \approx 2p^-$. Thus to lowest order in $p^-$, the recovery operation for the non-unitary evolution associated with the 'no-jump' operation $\hat{E}_0$ is a unitary rotation that maps

$$\cos\frac{\theta}{2}|0_{\mathrm{L}}\rangle + \sin\frac{\theta}{2}|0_{\mathrm{L}}'\rangle \longrightarrow |0_{\mathrm{L}}\rangle, \tag{180}$$

$$|1_{\mathrm{L}}\rangle \longrightarrow |1_{\mathrm{L}}\rangle. \tag{181}$$

It is interesting that the code has three stabilizers

$$S_1 \quad = Z_1 Z_2\,, \tag{182}$$

$$S_2 \quad = Z_3 Z_4\,, \tag{183}$$

$$S_3 \quad = X_1 X_2 X_3 X_4\,. \tag{184}$$

Measurement of the three error syndromes thus yields 3 classical bits of information which should be more than enough to handle the 5 most probable error states (no decay and four different single-qubit decays). It turns out however that this code does not exactly satisfy the Knill-Laflamme conditions–it does so only to lowest order in $p^- = \gamma dt$. It is the first example of an *approximate quantum code* in which the weaker condition of satisfying the Knill-Laflamme conditions only to first order widens the space of available codes, while still allowing error correction (to first order) [66]. Because the Knill-Laflamme conditions are not satisfied exactly, we cannot use simple stabilizer measurements as before. Similarly, one cannot use the fallback procedure of measuring the projectors onto each of the error states. Leung et al. found a different error decoding scheme which, somewhat surprisingly, involves measurements of the states of two individual qubits rather than any of the stabilizers. The recovery procedure leaves small errors in the state and also fails completely if a certain intermediate measurement result has the wrong value. However both of these errors are second order in $p^-$ and hence the code corrects first-order errors. The reader is directed to [66] for the details.

# 5 Topological Quantum Error Correction: Kitaev's Toric Code and the Surface Code

The so-called 'surface code' [32,69] and its modern variant the XZZX code [70,71] for quantum error correction has received wide theoretical scrutiny and many experimental efforts are underway to realize it to create topologically protected logical quantum bits. The surface code concept began with a concrete and exactly solvable many-body model for a so-called $\mathbb{Z}_2$ spin liquid due to Kitaev [29] and it is this that we will discuss in this section, closely following the discussion in [31]. The presentation relies on topological concepts that are extensively discussed in [31] and readers unfamiliar with these concepts may wish to skip this optional section on first reading.

Because the different topological sectors of the model are robust against local perturbations, the states of this model have potential application as an error-correctable quantum memory for quantum information processing. In this context, the model and its associated states are known as the **Kitaev 'toric code.'** We will first treat this as a Hamiltonian problem with an interesting topological structure in its ground state manifold. In actual applications as a topological memory, one does not need to realize the Hamiltonian. It is sufficient to have a set of measurement operations and single-bit Pauli operations conditioned on the measurement results to drive the system back into the ground state manifold when errors cause it to escape. It may also be possible to create a special dissipative bath to autonomously cool the system back into its ground state without active intervention.

Consider a square lattice, and put a spin one-half (i.e., a qubit) in the center of each bond of the lattice (instead of at each vertex or lattice site!). For an $L_x \times L_y$ rectangle, there are $2L_x L_y$ spins. We impose periodic boundary conditions, thus forming a torus. We use $s$ to label the 'star' of four bonds emanating from a lattice site (or vertex), $j$ to label an individual bond and its associated spin, and $p$ to label a square (or plaquette). Both the star and plaquette hold a total of four spins; see Fig.18. The toric code Hamiltonian takes the form

$$H_{\text{tc}} = -\sum_s A_s - \sum_p B_p\,, \tag{185}$$

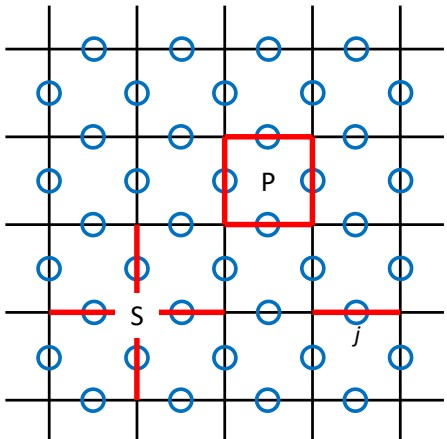

Figure 18: Illustration of the toric code model on a square lattice. Circles indicate the positions of spin-1/2 particles on the bonds of the lattice; $j$ labels the spin located at the center of the $j$th lattice bond; P labels a plaquette containing a total of four spins; S labels a 'star' also containing a total of four spins.

where

$$A_s = \prod_{j \in s} \sigma_j^z; \quad B_p = \prod_{j \in p} \sigma_j^x. \tag{186}$$

In the above $j \in s$ means the bond $j$ has $s$ as an end point, and $j \in p$ means $j$ is one edge of the square $p$. As we will see shortly, each of the $A_s$ and $B_p$ can be viewed as a stabilizer of an error correcting code defined by the ground state manifold of this Hamiltonian.

The Hamiltonian (185) looks quite strange and unphysical, as it involves four- instead of two-spin couplings, and the spin-rotation symmetry is badly broken. What is important however is that it is made of (albeit complicated) *local* couplings. As we will see, it captures all the universal topological properties of $\mathbb{Z}_2$ spin liquids. To reveal its solvability (which is by no means obvious), we note that all terms in (185) commute with each other:

$$[A_s, A_{s'}] = [B_p, B_{p'}] = [A_s, B_p] = 0. \tag{187}$$

As a result, all eigenstates, including ground states, can be chosen to be eigenstates of all of the $A_s$ and $B_p$. Since $A_s^2 = B_p^2 = 1$, they all have eigenvalues $\pm 1$ and we expect (because of the minus signs in the Hamiltonian) a ground state to have

$$A_s = B_p = 1, \tag{188}$$

for every $s$ and $p$. Since we have $L_x L_y$ sites and the same number of squares, it appears that there are $2L_x L_y$ constraints in eqn (188), which is the same as the number of spins in the system. If this were the case we would have a unique ground state. This is, however, not the case. In fact, the $A$ and $B$ operators are not all independent; it is easy to show that

$$\prod_s A_s = \prod_p B_p = 1. \tag{189}$$

As a consequence eqn (188) only represents $2L_x L_y - 2$ independent constraints, two fewer than the number of spins. This results in a $D$-fold ground state degeneracy with $D = 2^2 = 4$. To see that this topological degeneracy is robust against disorder and does not rely on the translation symmetry of the Hamiltonian, let us introduce disorder to the Hamiltonian (185) by considering

$$H'_{\text{tc}} = -\sum_s J_s A_s - \sum_p K_p B_p, \tag{190}$$

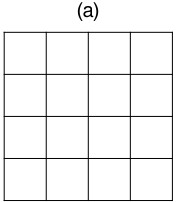 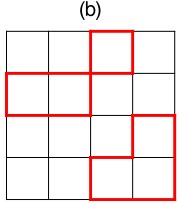 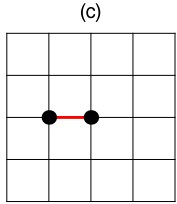 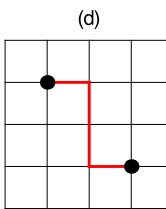

Figure 19: A square lattice in which the bonds with up spins are denoted by thin gray lines and those with down spins are denoted by thick black lines. (a) The configuration with all spins up. (b) For configurations satisfying $A_s = 1$ for all sites $s$, the thick bonds form closed loops. The no-loop configuration of (a) is a special case here. (c) Hitting one bond with $\sigma^x$ in (a) creates a pair of charges (black dots) on neighboring sites connected by this bond. (d) Repeated applications of $\sigma^x$ can move the two charges apart, with a string connecting them. In this model the string costs no energy and is invisible to local observables. The charges (which do cost energy to create) are thus deconfined and constitute well-defined quasiparticle excitations of the system. Since the string is *not* a closed loop, charges are topological defects of the ground state configurations depicted in (a).

where $J_s$ and $K_p$ are positive but otherwise random coupling constants. It is easy to see that the ground states still obey eqn (189) and are thus unchanged, and so is the degeneracy.[10]

The discussion thus far might leave the reader with the impression that the Hamiltonian (185) is rather trivial. This is, of course, not the case. The highly non-trivial nature of the toric code can be appreciated by considering its ground state wave functions. Let us start with the constraint $A_s = 1$. To satisfy this constraint it is most natural to work in the $\sigma^z$ basis for all the spins. In the ground state we must have an even number of up spins among the four in each group. One way to understand this particular constraint is to view the up spins as background, and focus on the down-spin bonds; this constraint is equivalent to requiring that all such bonds form closed loops [see Fig. 19(b)].

There are, of course, many such loop configurations. To be in one of the ground states however, we must also satisfy the other constraint $B_p = 1$. One way to ensure this is to start with any loop configuration $|\text{loop}\rangle$ (which automatically satisfies $A_s = 1$ for any $s$) or their arbitrary linear combinations, and then project it to the Hilbert subspace with $B_p = 1$:

$$|0\rangle = \prod_p P_p |\text{loop}\rangle, \tag{191}$$

where

$$P_p = (1 + B_p)/2 \tag{192}$$

is the projection operator that projects a state to one with $B_p = 1$, as guaranteed by the fact that the possible eigenvalues of $B_p$ are $\pm 1$. It is important to note that the operation of $P_p$ does not change the eigenvalues of $A_s$ because $[A_s, P_p] = 0$.

Let us consider in some detail what $P_p$ does when it hits a particular loop configuration (or state). The constant term $(1/2)$ of course does nothing. The other term $(B_p/2)$ is much more interesting. If none of the four bonds forming the square $p$ is involved in any existing loops then it creates a small loop around $p$, because $B_p$ flips all four spins in the plaquette. For similar reasons, if such a small loop already exists, then $B_p$ annihilates it. Most interestingly, if a subset of these four bonds are parts of other loop(s), then $B_p$ changes their configurations

---

[10]More generic local perturbations will lift this degeneracy and split the ground states, but the splitting vanishes exponentially with $L_x$ or $L_y$, similar to the situation for fractional quantum Hall liquids [31].

(by inserting a 'detour' into the path). We thus find that the operation $\prod_p P_p$ will generate a very complicated superposition of a huge number of loop configurations with arbitrarily large loops, even if the initial configuration in |loop⟩ contains no loops at all [which is the case if we start with the state in which all spins are up, as in Fig. 19(a)]. Such a massive superposition creates a long-range entangled and topologically-ordered state, with properties identical to the $\mathbb{Z}_2$ resonating valence bond (RVB) spin liquid state [31].

What are the possible outcomes of the projection operation in eqn (191)? Naively, one might think this is sensitive to the initial loop configuration, which would result in many possible final states. However, our earlier analysis found there are only four different ground states, indicating many different initial loop configurations will result in the *same* ground state through the operation in (191). Then the natural question is how to distinguish the different ground states as well as the initial loop configurations that lead to them. The distinction lies in topologically non-trivial loops. As discussed before, a torus is *not* a singly-connected manifold. As a result, there exist loops that wrap around the torus along either the x- or y-directions, that cannot shrink to zero (or no loop). Pairs of such loops, on the other hand, can join through the operation of $B_p$ and become a topologically trivial loop. However, this operation does not change the parity of such non-trivial loops. We thus find that we can assign two parity quantum numbers to each loop configuration, based on (the parity of) the number of non-trivial loops wrapping around the x- and y-directions. Since the operation of $B_p$ does not change these topological quantum numbers, we find eqn (191) generates four possible ground states, depending on the four possible combinations of these quantum numbers of the initial loop configuration. It should be clear by now that the four-fold ground state degeneracy of the toric code has the same topological origin as that of the RVB spin liquid discussed in [31], and such degeneracy takes the form $D^g$ in a genus $g$ lattice with $D = 4$.

What about excited states of the Hamiltonian (185)? Due to the property (187), they are simply states with some of the $A$s or $B$s equal to $-1$ instead of $+1$. The most elementary excitation is thus a lattice site with $A_s = -1$, or a square with $B_p = -1$. These are the two types of quasiparticles of the system. The first lives on a lattice site, and we will call this excitation 'charge,' while the second lives in a plaquette (which is also the smallest closed loop), and we will call it 'flux,' in obvious analogy with electromagnetism. The big difference here is that the charge and flux can only take vales 0 or 1, and form (separate) $\mathbb{Z}_2$ sets under addition. (That is charge and flux are only conserved modulo 2.)

At this point it may seem that these elementary charge or flux excitations are very similar to a single (local) spin flip in a ferromagnet. Again this is not the case. In fact a *single* excitation cannot be created by any *local* operations. Let us consider a charge excitation first. To create one at site $s$, it appears that all one needs to do is flip the spin on one of the bonds connected to $s$ using $\sigma^x$. It is easy to convince oneself that this operation actually creates a *pair* of charges on neighboring sites $s$ and $s'$, connected by the bond [see Fig. 19(c)]. Additional operations may separate these two charges, but never eliminate one of them [see Fig. 19(d)]. We thus find charges can only be created in pairs. Exactly the same is true for flux excitations. Of course, mathematically this property is guaranteed by the constraints (189).

Physically, the somewhat strange properties discussed above reflect the fact that these are topological excitations [or defects in the topological order; see Fig. 19(d)] that are intrinsically *non-local*. Such non-locality is reflected by the fact that these excitations can feel the existence of each other even when they are very far apart. Naively there is no interaction among them as the energy of the system is *independent* of their separations. However as we demonstrate below if one drags a charge around a flux, the system picks up a non-trivial Berry phase very much like that in the Aharonov-Bohm effect (hence the names charge and flux!). There is thus a 'statistical interaction' between a charge and a flux, which is of topological nature.[11]

---

[11]Such statistical interaction is identical to that between 'anyons' [31], except here charge and flux are *distinct*

Consider a charge on site $s$, and a loop $\mathcal{L}$ that starts and ends at $s$. We can drag the charge around this loop using the operator $\prod_{j\in\mathcal{L}} \sigma_j^x$ (where $j \in \mathcal{L}$ means the bond $j$ is part of the loop $\mathcal{L}$), resulting in the final state

$$|\text{final}\rangle = \prod_{j\in\mathcal{L}} \sigma_j^x |\text{initial}\rangle\,. \tag{193}$$

Using the fact

$$\prod_{j\in\mathcal{L}} \sigma_j^x = \prod_{p\in\mathcal{A}} B_p\,, \tag{194}$$

where $\mathcal{A}$ is the region enclosed by $\mathcal{L}$, we find

$$|\text{final}\rangle = |\text{initial}\rangle\,, \tag{195}$$

when there is an even number of flux excitations (which are equivalent to zero due to the $\mathbb{Z}_2$ nature of the flux) inside the loop, while

$$|\text{final}\rangle = -|\text{initial}\rangle\,, \tag{196}$$

when there is an odd number of flux excitations inside the loop, meaning the charge picks up a -1 Berry phase when circling around the flux (or vice versa).

Making contact with the RVB spin liquid [31], we can identify the charge excitation of the toric code as the spinon of the RVB liquid. There also exist flux-like excitations in the RVB spin liquid, which are more complicated to visualize in terms of the spin degrees of freedom, and we will not discuss them further.

The ground state(s) of the Kitaev toric code model have all the defining universal properties of a $\mathbb{Z}_2$ spin liquid. However, one could argue that the original RVB physics is not easily recognizable in the toric code Hamiltonian (185) or its ground state (191): There are no fluctuating singlet bonds, because the toric code Hamiltonian is far from being rotationally invariant. Recall that originally, the RVB spin liquid was proposed to describe antiferromagnets with (approximate) rotational invariance in spin space, and can be pictured as an equal amplitude superposition of all nearest-neighbor valence bond configurations. Indeed, while Anderson's variational RVB wave function was historically the first quantum many-body wave function with the kind of long-range entanglement that is also present in the Kitaev toric code, it long remained an open question whether it can actually appear as the ground state of a local rotationally invariant Hamiltonian. By now, a number of such local model Hamiltonians are known, which tend to be more complex than the Kitaev toric code. Of particular relevance here, is a local parent Hamiltonian for the RVB state on the Kagomé lattice [72], which is expected to belong to the same $\mathbb{Z}_2$ spin liquid phase as the Kitaev toric code, and many of its defining properties have been demonstrated explicitly.[12]

## 5.1 Toric Code as Quantum Memory

The four degenerate ground states of the Kitaev model on the torus mean that the toric code can be used to store two **quantum bits** of information. That is, the set of quantum superpositions of these four ground states span the same Hilbert space as superposition states of two spin one-half objects (**qubits**), $\vec{\sigma}_1, \vec{\sigma}_2$ which we will refer to as the **'logical qubits'** created out of the

---

instead of identical particles. Distinct types of particles with statistical interactions are often referred to as **mutual anyons**.

[12]It turns out having a frustrated lattice is crucial for the (nearest-neighbor) RVB state to represent a gapped topological phase. The RVB state on non-frustrated bipartite lattices, on the other hand, has been shown to represent a critical point instead of a stable phase.

collection of $N$ **'physical qubits.'** If we have a square array of $N$ physical qubits and manage to realize the toric code Hamiltonian and its periodic boundary conditions (a non-trivial task[13]) we can use the toric-code logical qubits as a quantum memory.

The basic idea behind this is that the two logical qubits are 'topologically protected.' That is, small deviations from the ideal Hamiltonian will lead to virtual production of pairs of 'charges' or 'fluxes' which will quickly recombine because of the energy gap. These will have little effect since moving the state of the system from one logical quantum ground state to another would require one of the topological defects to travel on a non-contractible loop winding around the torus before returning to annihilate with its partner.[14] This requires tunneling a distance $\sim \sqrt{N}$ and is exponentially small. That is, if we do perturbation theory, the small parameter $\lambda$ is the strength of the deviations from the ideal Hamiltonian divided by the excitation gap. Processes in which a (virtual) topological defect winds around the torus appear at order $\lambda^{\sqrt{N}}$, i.e., the code distance is $\sqrt{N}$.

The good news then is that the qubit states are topologically protected. That is, the (non-local) logical states and the distinction among them is invisible to local perturbations that flip or dephase individual physical qubits. The bad news is that the logical states are sufficiently well protected that it is difficult for us to change them when we want to carry out a single-qubit operation (e.g. rotation on the Bloch sphere). It is relatively easy to carry out a Pauli group operation (see Box 6) on one of the logical qubits (i.e., bit flip $\sigma_1^x$, phase flip $\sigma_1^z$ or both which yields $i\sigma_1^y$). We do this by repeated Pauli operations on a line of physical spins along a non-contractible loop of the torus as described above. But what if we want to rotate the qubit through an angle $\theta$ around the $x$ axis on the Bloch sphere? This requires

$$U_\theta = e^{i\frac{\theta}{2}\sigma_1^x} = \cos\frac{\theta}{2}\sigma_1^0 + i\sin\frac{\theta}{2}\sigma_1^x, \tag{197}$$

where $\sigma_1^0$ is the identity operation. Thus we have to carry out a quantum superposition of doing nothing and flipping a large number ($\sqrt{N}$) of physical spins. It appears to be difficult to achieve this superposition of macroscopically distinct operations though proposals to get around this difficulty using magic state distillation and gate teleportation exist and are reviewed in Ref. [69].

A second problem is created by the fact that the energy gap for excitations is finite.[15] Even if the physical qubits are in good thermal equilibrium (something difficult to achieve inside an operating quantum computer) at low temperature, there will be a finite density of topological defects. The density will be exponentially small in the gap divided by the temperature, but it will be finite. Small deviations from the ideal Hamiltonian will allow these topological defects to travel (either coherently or diffusively) around the sample producing logical errors.

The only way to beat these thermally-induced errors is to introduce a 'Maxwell demon' to 'algorithmically cool' the system by rapidly and repeatedly measuring all the constraint operators ('error syndromes') $A_s$ and $B_p$ to determine when a thermal fluctuation has produced a local pair of defects. The demon must then quickly compute what local operations on the physical qubits it needs to carry out to cause the topological defects to recombine before they

---

[13]The four-qubit interaction terms of the Hamiltonian are difficult to achieve directly (since interactions are typically two-body) and one must resort to an *effective* Hamiltonian obtained by having each of the four qubits in a star or plaquette virtually excite an ancilla qubit that has a strongly different transition frequency. It is possible to avoid the difficulty of realizing periodic boundary conditions by using the closely related 'surface code' [32,69] which has open boundary conditions.

[14]Recall that to go from one (topologically degenerate) ground state to another in a fractional quantum Hall system requires moving a Laughlin quasiparticle (a topological defect) around the torus, which is a non-contractable loop [31].

[15]There exist models in four spatial dimensions in which the topological defects are confined by a finite string tension (much like quark confinement in 3+1 space-time dimensions). These could be realized in three spatial dimensions via non-local Hamiltonians but this would be extremely challenging to realize experimentally.

can change one of the topological winding numbers. This version of quantum error correction will be extremely challenging to achieve, especially when one considers the fact no Maxwell demon will be perfect–it will introduce errors of its own when it reads out the error syndromes and tries to correct them. A very important concept in quantum error correction is that of the 'break even' point–that is the point at which the error rate of the physical qubits is low enough and the Maxwell demon works well enough and quickly enough that a code using $N$ physical qubits would yield collective logical qubits with coherence times exceeding that of the best of the individual physical qubits [38]. At the time of this writing, it appears to be very challenging [32,69] for the surface code to achieve breakeven without considerable technical advances, but substantial efforts are being made and recent theoretical progress on the XZZX code [70,71] is encouraging.

### Exercise 20

Prove eqns (187) and (189).

### Exercise 21

Show that the state (191) satisfies (188).

### Exercise 22

Prove eqn (194).

### Exercise 23

The toric code has a finite excitation gap. Find a lower bound for this gap for the Hamiltonian given in eqn (190).

### Exercise 24

Consider how the constraints in eqn (189) need to be modified when the random couplings $J_s$ and $K_p$ in eqn (190) are allowed to take negative values, and show that the ground-state degeneracy remains exactly four.

### Exercise 25

Consider the perturbing effect on the ground state(s) of the Hamiltonian given in eqn (190) of a transverse magnetic field applied to a single site $j = J$ via $V = \lambda \sigma_J^x$. Find the energy denominator that controls the strength of the perturbation and show that the effects of the perturbation do not propagate throughout the lattice but instead remain local. Use this fact to find the *exact* ground-state energy of the perturbed Hamiltonian.

### Exercise 26

The locality of the response to a local perturbation provides a hint that helps explain why generic perturbations on the exactly soluble model have exponentially small effects in lifting the degeneracy of the ground state manifold. Suppose that the perturbation Hamiltonian $V = \lambda \sum_j \sigma_j^x$ acts on all sites of an $L_x \times L_y$ lattice. At what order in perturbation theory is the ground state degeneracy first lifted?

**Exercise 27**

Show that as a flux is dragged around a charge, the system picks up a -1 Berry phase.

**Exercise 28**

Generalize the toric code to the 3D cubic lattice and discuss how the charge and flux excitations differ from the 2D case. What is the ground state degeneracy when periodic boundary conditions are imposed in all three directions?

# 6 Overview of Bosonic Modes

## 6.1 Introduction

In this section we will begin our study of continuous variable quantum information [47,73–78] in which we use the large Hilbert space of the harmonic oscillator to store and process information and carry out quantum error correction. To set the stage, let us remind ourselves of what we have learned so far in our study of two-level qubit systems. We saw previously that quantum error correction for qubits required creation of 'logical qubits' consisting of many 'physical' qubits. The quantum information encoded in the logical qubit lives in a specially designed subspace of the larger Hilbert space of the many physical qubits. The states in this subspace are highly entangled states of the physical qubits in order to hide the information from the environment and thereby protect it. The states outside the logical subspace are error states and one must design a measurement protocol whose measurement results ('error syndromes') tell us, not about which of the specific error states outside the logical subspace the system is in, but only what errors have occurred without revealing any information about the logical state in which the error has occurred. We saw that if the Knill-Laflamme conditions are satisfied, we have a guarantee that this is possible.

Having a guarantee that this is possible in principle is not the same thing as achieving it in practice.[16] A particular challenge is that the number of single-physical-qubit errors that can occur (and the rate at which they occur) scales linearly with $N$, the number of physical qubits comprising the logical qubit. If the chosen code is capable of correcting (say) all single-physical-qubit errors, the measurement scheme must determine a large amount of information, namely which of the $N$ physical qubits suffered the error and what particular type of error occurred (e.g. $X, Y, Z, X - iY, X + iY, ...$). This requires a lot of additional wiring and measurement ancillae all of which are imperfect. Furthermore, the code stabilizers are typically 'unnatural' high-weight operators. For the 3-qubit repetition code, we saw that the stabilizers are only weight two ($Z_1 Z_2$ and $Z_2 Z_3$). However this code only protects against one type of error (bit flips). The 9-qubit Shor code stabilizers are weight six [19] and therefore (at present at least) impractical to measure.

With these thoughts in mind, we turn now to the harmonic oscillator. The physical realization of the oscillator might be a mechanical mode (e.g. the ion motion in a trapped ion system) or a microwave mode in a superconducting resonator. In principle we could also use optical modes but, to date, full quantum control of the state of optical modes has not been achieved. In the circuit QED microwave regime, ultra-strong coupling to transmon artificial atoms with huge electric dipole moments (several Cooper pairs moving a millimeter) yields non-linear op-

---

[16]In theory, theory and practice are the same. However in practice, they are different.

tical effects at the level of single photons which can be exploited to achieve complete quantum control over both the atom and the photon states.

## 6.2  Pros and Cons of Microwave Bosonic Modes

One immediate advantage of microwave oscillators is that they have extremely high $Q$ and it is relatively easy to make 3D microwave resonators with lifetimes of 1-10 milliseconds–1 to 2 orders of magnitude longer than typical transmon qubit $T_1$ lifetimes. Much longer times are possible [79,80] but these generally involve using larger cavities which 'dilute' the electric field of a single photon and therefore have weaker coupling to the transmon ancillae used to control them. Because all of the internal loss of such resonators is associated with the electromagnetic surface impedance of the walls, we can think of these resonators as analogous to a Fabry-Pérot cavity formed between two optical mirrors with reflectivity very slightly less than unity leading to a large but finite 'finesse' that is a measure of the number of round trips a photon can make between the mirrors before it leaks out. As the cavity volume is increased by moving the mirrors further apart, the finesse remains fixed, but it takes a longer time for the photon to travel between bounces and hence the $Q$ increases, while the dispersive interaction with the qubit decreases by the same factor.

A nice feature of cryogenic microwave resonators is that their frequency is determined by their rigid geometry and is extremely stable. Hence their intrinsic dephasing rate is extremely low. If we think of the 0 and 1 photon states as forming a qubit, its phase coherence time for superpositions of 0 and 1

$$\frac{1}{T_2} = \frac{1}{2T_1} + \frac{1}{T_\varphi} \tag{198}$$

has negligible contribution [81] from $\frac{1}{T_\varphi}$.

A corresponding disadvantage of a harmonic oscillator is that its energy levels are all evenly spaced which prevents universal control using classical drives. The only effect of a classical drive is to displace the oscillator in momentum and position forming a coherent state. As will be discussed in Sec. 6.3, without the presence of a non-linear oscillator such as a transmon ancilla, universal control of the oscillator states is not possible. Thus while microwave cavities have much longer natural coherence times than transmons, we still need the transmon as a control ancilla [82, 83] and must deal with its faults. These can be partially ameliorated by using control pulses which only virtually excite the transmon so that it mostly remains in its ground state. However, even when the transmon controller is nominally idle, it can accidentally change states and the resulting dispersive shift in cavity frequency (occurring at a random moment in time) will dephase the cavity. Fault-tolerant protocols to eliminate the dephasing from such events have however been demonstrated by dressing the controller with off-resonant drives which eliminate the cavity dispersive shift difference between the 0 and 1 states of the controller [81].

It is a significant advantage of microwave cavities that their error rate is low and the error model is *highly biased*. The dominant error is photon loss (due to a combination of unintended internal dissipation and intended coupling to external signal access ports). Furthermore, unlike the case of an encoding in $N$ separate qubits, we do not have to figure out which of the $N$ qubits incurred an error. We have only one physical device, the oscillator, that incurs errors. We saw earlier that the smallest qubit error correcting code has $N = 5$ qubits and has a total of 16 possible error states (including zero errors). Even if we assume the noise is highly biased and consists only of amplitude damping ($\sigma_-$ jump operator) the smallest qubit code that can correct a single damping error still requires $N = 4$ qubits and has a total of 5 possible error states [66]. Remarkably, there exists an analogous oscillator bosonic code [84, 85] to correct

amplitude damping that requires only 5 states in the oscillator Hilbert space compared the $2^N = 16$ states of the corresponding $N = 4$ qubit code.

The mean rate of photon loss for a damped oscillator is proportional to the photon number

$$\frac{d\bar{n}}{dt} = -\kappa\bar{n}.\tag{199}$$

As noted above, $\kappa$ is much smaller than the typical energy relaxation rate $1/T_1$ for a transmon and the intrinsic dephasing rate of the cavity is (so far) completely negligible. We will be able to take advantage of this error asymmetry in the design of bosonic error correction codes. The topic is also addressed in the lectures of Mazyar Mirrahimi and the seminar by Shruti Puri at this school. [Videos available at https://physinfo.fr/houches2019/program.html.]

One natural advantage of the harmonic oscillator is the large size of its Hilbert space. It would be very convenient and hardware-efficient to replace the states of many physical qubits with the states of one oscillator. Let us see what happens when we do this. $N$ qubits have a Hilbert space dimension $D_N = 2^N$. We can fit these into levels $0, 1, \ldots, D_N - 1$ of the oscillator. One simple way to do this would be to use the binary encoding

$$|b_{N-1}\ldots b_2 b_1 b_0\rangle_{\text{qubits}} \Longleftrightarrow |M\rangle_{\text{oscillator}},\tag{200}$$

where $|M\rangle_{\text{oscillator}}$ is the Fock state containing $M$ bosons and $b_{N-1}\ldots b_2 b_1 b_0$ is the set of 0's and 1's that define the qubit state in the standard ($Z$) basis and whose values form the binary representation of the number $M$. Thus for N=3 the bit and oscillator state correspondences would be

$$
\begin{aligned}
|000\rangle_{\text{qubits}} &\Longleftrightarrow |0\rangle_{\text{oscillator}},\\
|001\rangle_{\text{qubits}} &\Longleftrightarrow |1\rangle_{\text{oscillator}},\\
|010\rangle_{\text{qubits}} &\Longleftrightarrow |2\rangle_{\text{oscillator}},\\
|011\rangle_{\text{qubits}} &\Longleftrightarrow |3\rangle_{\text{oscillator}},\\
|100\rangle_{\text{qubits}} &\Longleftrightarrow |4\rangle_{\text{oscillator}},\\
|101\rangle_{\text{qubits}} &\Longleftrightarrow |5\rangle_{\text{oscillator}},\\
|110\rangle_{\text{qubits}} &\Longleftrightarrow |6\rangle_{\text{oscillator}},\\
|111\rangle_{\text{qubits}} &\Longleftrightarrow |7\rangle_{\text{oscillator}}.
\end{aligned}\tag{201}
$$

This seems wonderfully efficient, but there are some problems we must point out. First, notice that the highest photon number needed in the encoding for $N$ qubits grows exponentially $D_N - 1 = 2^N - 1$. In the presence of single-photon energy relaxation rate $\kappa$, the rate of decay from photon Fock state $M$ to $M - 1$ is given by $M\kappa$. Hence the worst-case bosonic state decay rate becomes exponentially large. This is true for all possible encodings, not just the binary encoding shown here. Thus, as discussed in 'Climbing Mount Scalable' [86], we cannot use the infinite Hilbert space of the harmonic oscillator to replace an arbitrarily large number of two-level qubits. Nevertheless, bosonic codes utilizing the lowest few levels of an oscillator will prove to be quite powerful.

In addition, there is a particular problem with the binary encoding which is illustrated by the following. The error model for the bosons is to a very good approximation simply photon loss whose 'jump operator' is the photon destruction operator $a$ that takes $|M\rangle$ to $\sqrt{M-1}|M-1\rangle$. Notice however in the qubit representation, single photon loss produces transitions for example like this (in a 5-qubit system)

$$|10000\rangle_{\text{qubits}} \to |01111\rangle_{\text{qubits}}.\tag{202}$$

Thus, in terms of the qubits being represented, the errors produce multiple, correlated bit flips. Such an error model would seem to make QEC impossible. One might be able to get

around this by replacing the binary encoding with the Gray code [19, 87] whose neighboring codewords all have Hamming distance one. However the $\sqrt{M-1}$ matrix elements of the boson destruction operator become very complex in this representation, so it is not obvious if this is much better.[17]

Another approach to encoding that has certain distinct advantages is to use qubit encoding to represent the bosonic oscillator (discretized) coordinate rather than the boson number [88–90].

With all these considerations in mind, it seems clear that we should not try to represent too many physical qubits in a single oscillator and may not want to choose to represent them by a simple binary or Gray coding or the like. Furthermore it may be useful to take a 'continuous-variable' point of view and recognize that unlike qubits, harmonic oscillators have continuous coordinates and momenta. In this regard, one natural source of inspiration are classical communication codes based on modulating the phase, amplitude and polarization of light and/or its time-bin structure. Ignoring the polarization, we can represent the amplitudes of the two quadratures of a carrier wave by a pair of numbers $(I, Q)$ defined by how the voltage (say) varies in time

$$V(t) = I(t)\cos(\omega t) + Q(t)\sin(\omega t), \tag{203}$$

where $\omega$ is the carrier frequency, $I(t)$ is the 'in-phase' amplitude and $Q(t)$ is the 'quadrature-phase' amplitude. Confusingly these are both referred to as quadrature amplitudes and often combined into a 2D vector or single complex number labeling a point $z = I + iQ$. The simplest classical example is Phase Shift Keying (PSK) in which the bits $b = 0, 1$ are encoded in the phase of the wave via

$$I = 2b - 1, \quad Q = 0, \tag{204}$$

as illustrated in the left panel of Fig. 20. The quantum analog of this code is the two-legged Schrödinger cat code [91]. The middle panel illustrates the two-qubit quantum amplitude keying code. Here the four states all have the same amplitude but differing phases. This bears some resemblance to the quantum four-legged cat code [38, 92, 93] discussed in the lectures by Mazyar Mirrahimi at this school [https://physinfo.fr/houches2019/program.html]. The right panel shows the four-bit 16 state quadrature amplitude modulation code in which the phase space points are distributed on a lattice. This bears some resemblance to the quantum GKP code [58, 94–99] discussed in the lectures by Barbara Terhal at this school [https://physinfo.fr/houches2019/program.html] and in Sec. 8 of the present notes.

The highly successful Schrödinger cat code [38, 92, 93] takes particular advantage of the fact that coherent states are eigenstates of the dissipation jump operator $a$. A coherent state is the closest quantum relative of the classical quadrature states discussed above and can be created by a classical drive applied to the oscillator. However a quantum superposition of two or more coherent states is a 'non-Gaussian' state and requires a non-classical control to create. Before diving into all of this, we need to understand two topics: universal control of oscillator states and the quantum description of amplitude damping for a harmonic oscillator. The former will be described in Sec. 6.3 For the latter, we will derive the Kraus map describing the dissipation in Sec. 6.4.

## 6.3 Gaussian and Non-Gaussian States of Oscillators

The ground state wave function of a harmonic oscillator is of course a Gaussian function of the coordinate

$$\Psi(q) = \frac{1}{\sqrt{2\pi q_{\text{ZPF}}^2}} e^{-\frac{q^2}{4q_{\text{ZPF}}^2}}, \tag{205}$$

---

[17]Isaac Chuang, private communication.

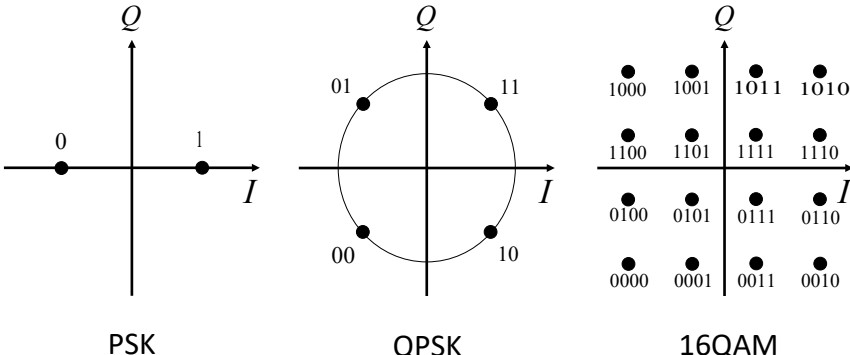

Figure 20: Phase space representation of different standard classical encodings for optical communication. Left panel: PSK is phase shift keying. The two codewords representing one bit differ by a $\pi$ phase shift. Middle panel: QPSK is quadrature phase shift keying. Every codeword has the same amplitude, but the four codewords can store two bits of information. Right panel: QAM is quadrature amplitude modulation. The sixteen codewords carry four bits of information at the expense of having to resolve both amplitude and phase of the signal. All three of these codes assign bit values to the particular points in phase space using the Gray code. In this manner every point has Hamming distance 1 from each of its neighbors. This minimizes the number of bit-flip errors if noise causes the receiver to misinterpret the signal associated with one point with that of one of the neighboring points.

where $q_{\mathrm{ZPF}}$ is the RMS position zero-point fluctuation (uncertainty) in the ground state. In quantum optics, the coordinate of an electromagnetic oscillator mode is generally taken to be the electric field of the mode at some particular position, and the ground state fluctuations in the electric field are referred to as *vacuum noise* [100]. As usual, the wave function is also a Gaussian in the momentum representation.

It turns out that under the influence of any single- or multi-mode Hamiltonian that is quadratic in position and momentum (or equivalently quadratic in the creation and annhilation operators), Gaussian states evolve into other Gaussians. This means that it is easy to classically compute and represent the time-evolution in terms of a few parameters representing the mean values and covariances of the creation and annhilation operators, $\langle a \rangle$, $\langle a^{\dagger}a \rangle$, $\langle aa \rangle$, $\langle a^{\dagger}b^{\dagger} \rangle$, etc. We do not need to keep track of higher-order correlators, because for Gaussian states, Wick's theorem guarantees that they are simple products of lower order correlators. For example (assuming for simplicity that $\langle a \rangle = \langle b \rangle = 0$), Wick's theorem tells us that

$$\langle a^{\dagger}ab^{\dagger}b \rangle = \langle a^{\dagger}a \rangle \langle b^{\dagger}b \rangle + \langle a^{\dagger}b^{\dagger} \rangle \langle ab^{\dagger} \rangle + \langle a^{\dagger}b \rangle \langle ab^{\dagger} \rangle. \tag{206}$$

In condensed matter physics, terms like $\langle aa \rangle$ and $\langle ab \rangle$ are known as anomalous correlators. In quantum optics they are described as representing one- and two-mode squeezing.

We can draw an analogy [101] between the simplicity of computing time evolution under quadratic bosonic Hamiltonians and the Gottesman-Knill theorem [60, 61], that a quantum computer based on qubits and using

   i) Preparation of qubits in computational basis states,

   ii) Clifford gates (only), and

   iii) Measurements in the computational basis,

is easy to simulate classically. The discussion of this analogy given below is summarized in Table 4.

As a reminder, the Pauli group is generated by products of qubit $X$ and $Z$ operators, and the Clifford group is the set of operations in the *normalizer* of the Pauli group–that is the set of unitary operations $C$ that leave the Pauli group invariant (see Box 6). For example the Hadamard operation is a member of the Clifford group because

$$HZH^\dagger = HZH = X, \quad HXH = Z. \tag{207}$$

The full Clifford group for an arbitrary number of qubits can be generated using only three types of gates $S = \sqrt{Z}$, $H$, and CNOT. Despite the fact that Clifford circuits can generate maximally entangled Bell states (see Ex. 29), they are easy to simulate classically [61]. Essentially one can define an $N$-qubit starting (product) state by the list of $N$ one-qubit Pauli operators which stabilize the state (that is, the state is an eigenvector with eigenvalue $+1$ for each Pauli operator in the list). One can keep track of each Clifford operation applied to the state by simply updating the list of $N$ stabilizers (e.g., conjugating with $H$ takes a $Z$ stabilizer to an $X$ stabilizer, $Z$ takes $X$ to $-X$, $S$ takes $X$ to $Y$, CNOT takes $XI$ to $XX$, etc.). Even though the $H$ gate creates superpositions of single qubits and CNOT $H$ can be used to create entangled Bell pairs, one does not have to keep track of the large number of superpositions generated by Clifford circuits, but rather simply needs to track the changes in the list of stabilizers [61]. This ability to efficiently simulate Clifford circuits classically indicates that they do not represent the full power of quantum computation. This is presumably due to the fact that the full non-classical correlations inside Bell states cannot be revealed without making 45 degree rotations on the Bloch sphere (or rotating the measurement axis by 45 degrees). If we add a $T$ gate which produces such rotations, we can violate the Bell inequalities and achieve universal quantum computation. At the same time, the circuits become exponentially hard to simulate classically (because now you have to keep track of superpositions (i.e., multiple branching) lists of stabilizers).

---

**Exercise 29**

Find Clifford circuits that will take a pair of qubits initially in $|\downarrow\downarrow\rangle$ into each of the 4 maximally entangled Bell states.

$$|B_0\rangle = \frac{1}{\sqrt{2}}\left[|\uparrow\downarrow\rangle - |\downarrow\uparrow\rangle\right], \tag{208}$$

$$|B_1\rangle = \frac{1}{\sqrt{2}}\left[|\uparrow\downarrow\rangle + |\downarrow\uparrow\rangle\right], \tag{209}$$

$$|B_2\rangle = \frac{1}{\sqrt{2}}\left[|\uparrow\uparrow\rangle - |\downarrow\downarrow\rangle\right], \tag{210}$$

$$|B_3\rangle = \frac{1}{\sqrt{2}}\left[|\uparrow\uparrow\rangle + |\downarrow\downarrow\rangle\right]. \tag{211}$$

---

Because Gaussian operations for bosonic systems are easy to simulate classically, we do not expect such circuits to yield universal computation. The analogy goes further because two-mode squeezing produces maximally entangled states (see Ex. 32) just as Clifford circuits produce Bell states. There is one point however on which continuous variables are not analogous. As emphasized in Sec. 3.3 for the case of qubits, even though the noise demon has the power to do non-Clifford operations, we can (under certain assumptions for the noise model) defeat it and carry out QEC using only Clifford circuits and measurements. The analogous

Table 4: Qubit Clifford Circuits vs. Gaussian Bosonic Circuits. TMS stands for two-mode squeezing. The Weyl-Heisenberg group is the group of (non-commuting) translations in phase space effected via $X(s)$ which translates by $s$ in position and $Z(t)$ which translations by $t$ in momentum. Under non-Gaussian measurements, $|1\rangle\langle 1|$ is the projector onto the Fock state with one photon and $a^\dagger a$ stands for photon number resolving detection.

| Aspect | Qubits | Bosonic Mode |
|---|---|---|
| Standard Basis | $Z$, $\{0,1\}$ | $|q\rangle$ |
| Conjugate Basis | $X$, $|0\rangle \pm |1\rangle$ | $|p\rangle = \int dq\, e^{iqp}|q\rangle$ |
| Basis Transformation | Hadamard | Fourier Transform |
| Group | Pauli | Weyl-Heisenberg |
| | $\{Z,X\}$ | $\{X(s) = e^{is\hat{p}},\ Z(t) = e^{it\hat{q}}\}$ |
| | Clifford | Gaussian Unitaries |
| | $\{Z,H,\text{CNOT}\}$ | $\{e^{i\theta\hat{n}}, e^{i\lambda\hat{q}^2}, e^{ig\hat{q}_1\hat{q}_2}\}$ |
| Entangled State | Bell: $|00\rangle + |11\rangle$ | EPR (TMS): $\int dq\, |q\rangle_1 |q\rangle_2$ |
| Universal Computation | $T$ gate | Non-Gaussian unitary: $U_3(\chi) = e^{i\chi\hat{q}^3/3}$ |
| Clifford Measurements | $Z, X$ | homodyne $\hat{q}, \hat{p}$ |
| Non-Clifford Measurements | $\vec{b}\cdot\vec{\sigma}$ | $a^\dagger a, aa, |1\rangle\langle 1|, \hat{q}^2$ |

statement for continuous variables is not true. There is a no-go theorem stating that Gaussian resources (i.e., Gaussian operations plus homodyne measurement) are insufficient to do continuous variable QEC [65]. Table 4 summarizes the overall analogy we have discussed here.

---

**Exercise 30**

Consider a driven undamped harmonic oscillator with interaction picture Hamiltonian

$$\hat{V}(t) = +i[\epsilon(t)\hat{a}^\dagger(t) - \epsilon^*(t)\hat{a}(t)]. \tag{212}$$

Solve the equation of motion for $\hat{a}(t)$. Show from this that the vacuum state evolves into a coherent state $|\alpha(t)\rangle$ and derive an expression for $\alpha(t)$.

---

---

### Exercise 31

Consider an undamped harmonic oscillator with the following interaction picture Hamiltonian describing two-photon driving (single-mode squeezing)

$$\hat{V}(t) = +\frac{i}{2}[\epsilon(t)\hat{a}^\dagger(t)\hat{a}^\dagger(t) - \epsilon^*(t)\hat{a}(t)\hat{a}(t)]. \tag{213}$$

Solve the equation of motion for $\hat{a}(t)$. Hint try a solution of the form

$$\begin{pmatrix} \hat{a}(t) \\ \hat{a}^\dagger(t) \end{pmatrix} = \begin{pmatrix} \cosh\theta(t) & \sinh\theta(t) \\ \sinh\theta(t) & \cosh\theta(t) \end{pmatrix} \begin{pmatrix} \hat{a}(0) \\ \hat{a}^\dagger(0) \end{pmatrix}. \tag{214}$$

### Exercise 32

Consider the two-mode squeezed state

$$|\Psi_r\rangle = \frac{1}{\cosh(r)} e^{\tanh(r)\, a^\dagger b^\dagger} |00\rangle. \tag{215}$$

Write the state in the Fock basis and describe qualitatively how it is entangled.

Form the reduced density matrix for the $a$ mode (or equivalently the $b$ mode) and show it is precisely a thermal state with a Bose-Einstein distribution of photon numbers. This is the essence of Hawking radiation from a black hole. If the $b$ mode falls into the black hole and becomes inaccessible to us, we see a thermal distribution for the $a$ mode.

---

## 6.4 Kraus Map for the Damped Harmonic Oscillator

In Sec. 4 we studied density matrices for qubits and the Kraus representation of the CPTP maps for general quantum operations on states (density matrices). Here we extend this notion to the weakly damped harmonic oscillator. For simplicity, we will assume that a cold bath provides the damping so that the oscillator can lose excitations to the bath but never gain them. The coupling to the bath is of the form

$$V = \hbar\lambda a \mathcal{O}_{\text{bath}}, \tag{216}$$

and Fermi's Golden Rule for the decay rate of an oscillator of frequency $\omega$ from Fock state $|n\rangle$ to Fock state $|n-1\rangle$ gives

$$\Gamma_n = \lambda^2 2\pi \langle n|a^\dagger a|n\rangle \sum_f \langle i|\mathcal{O}^\dagger_{\text{bath}}|f\rangle \langle f|\mathcal{O}_{\text{bath}}|i\rangle \delta(\omega - \Omega_{if}) \tag{217}$$

$$= \kappa n, \tag{218}$$

where $i$ and $f$ label the initial and final states of the bath, $\Omega_{if}$ is the energy difference between the two bath states,

$$\kappa = \lambda^2 S(\omega), \tag{219}$$

and

$$S(\omega) = 2\pi \sum_f \langle i|\mathcal{O}^\dagger_{\text{bath}}|f\rangle \langle f|\mathcal{O}_{\text{bath}}|i\rangle \delta(\omega - \Omega_{if}) \tag{220}$$

is the spectral density of the bath operator at the frequency of the oscillator. (In some sense the defining characteristic of a bath is that this spectral density does not depend on the microscopic state $i$ of the bath.)

With this result in hand we see that the Kraus operator corresponding to loss of a photon during a small time interval $dt$ must be

$$\hat{K}_1(dt) = \sqrt{\kappa dt} a, \tag{221}$$

because this correctly yields the probability of a single photon loss

$$P_1(dt) = \text{Tr}\left\{\hat{K}_1^\dagger \hat{K}_1 \hat{\rho}\right\} = \langle \hat{n} \rangle \kappa \, dt. \tag{222}$$

Loss of more than one photon in a very short time interval is unlikely, so we are left only with a single remaining Kraus operator that describes the zero loss case and from eqn (103) must obey

$$\hat{K}_0^\dagger(dt)\hat{K}_0(dt) = 1 - \hat{K}_1^\dagger(dt)\hat{K}_1(dt) = 1 - \kappa \, dt \, a^\dagger a. \tag{223}$$

To lowest order in $dt$, the unique solution for $\hat{K}_0(dt)$ that approaches the identity for $dt \to 0$ is

$$\hat{K}_0(dt) = \sqrt{1 - \kappa a^\dagger a \, dt} \approx 1 - \frac{\kappa}{2} a^\dagger a \, dt. \tag{224}$$

We can now write the evolution equation for the density matrix over the small time interval $dt$

$$\hat{\rho}(t + dt) = \hat{K}_0(dt)\hat{\rho}\hat{K}_0^\dagger(dt) + \hat{K}_1(dt)\hat{\rho}\hat{K}_1^\dagger(dt) \tag{225}$$

$$= \hat{\rho}(t) - \frac{\kappa}{2}\left[a^\dagger a \hat{\rho} + \hat{\rho} a^\dagger a - 2a\hat{\rho}a^\dagger\right]. \tag{226}$$

We can now convert this into a differential equation

$$\frac{d\hat{\rho}}{dt} = \kappa\left[a\hat{\rho}a^\dagger - \frac{1}{2}a^\dagger a \hat{\rho} - \frac{1}{2}\hat{\rho} a^\dagger a\right] \equiv \kappa \mathcal{D}[a])\hat{\rho}, \tag{227}$$

where $\mathcal{D}[a]$ is a 'superoperator' defined by this equation.

This is the standard Lindblad master equation for the time evolution of the density matrix. It is clear from the cyclic property of the trace that the trace of the density matrix is automatically conserved in this dynamics. The last two terms in the square brackets correspond to time evolution under a non-Hermitian Hamiltonian

$$H = -i\frac{\kappa}{2}a^\dagger a, \tag{228}$$

describing the damping. Under the influence of this term the probability to be in a particular photon number state decays away exponentially and the trace of the density matrix falls continuously. This is repaired by the first term in square brackets, known as the 'quantum jump' term because it has no interpretation in terms of a Hamiltonian. This term describes the fact that when the system leaves Fock state $n + 1$ it must land in Fock state $n$. This term compensates the other terms and restores conservation of probability. One can verify this by using eqn (227) to write out the equation of motion for

$$\frac{d}{dt}\hat{\rho}_{n,n} = \frac{d}{dt}\langle n|\hat{\rho}|n\rangle = \kappa\left[(n+1)\hat{\rho}_{n+1,n+1} - n\hat{\rho}_{n,n}\right]. \tag{229}$$

In Boltzmann equation language, the first term on the RHS is the 'scattering in' term and the second is the 'scattering out' term.

Strictly speaking, Fermi's Golden Rule is not valid for time intervals $dt < W^{-1}$, where $W$ is the inverse frequency bandwidth of the bath spectral density. Hence the master equation is valid for small $dt$ but not too small. The time intervals have to be slightly coarse-grained and we have to be a bit cautious when we interpret eqn (227) as a differential equation. We have

made the Markov approximation for the bath spectral density by assuming that the bandwidth $W$ is so large that we can make $dt$ so small that the unitary evolution of the system under the action of $V(t)$ over that interval is small. Furthermore, we have implicitly assumed that $S(\omega)$ is constant over the large bandwidth so that its Hilbert transform vanishes. In the language of many-body physics, this means that the bath only contributes a frequency-independent and purely imaginary 'self-energy' (i.e., damping) to the oscillator and does not renormalize the oscillator frequency (which would occur if the self-energy had a real part given by the Kramers-Kronig (Hilbert) transform of the imaginary part via

$$\Sigma(\omega + i\delta) = \int_{-\infty}^{+\infty} \frac{d\omega'}{2\pi} \frac{S(\omega')}{\omega - \omega' + i\delta}. \tag{230}$$

The real part of the self-energy $\Sigma$ essentially describes level repulsion between the states of the system and the bath. If the spectral density is larger for bath states higher than $\omega$ and lower for bath states below $\omega$, then the oscillator frequency will be pushed downwards.

Now that we have found the Kraus operators describing the effect of damping on an oscillator for a small time interval $dt$, a useful next step is to find Kraus operators corresponding to evolution over a finite time interval. We will organize them according to the number of photons $\ell$ that are emitted into bath in the time interval $[0, t]$

$$\hat{\rho}(t) = \sum_{\ell=0}^{\infty} \hat{K}_\ell(t)\hat{\rho}(0)\hat{K}_\ell^\dagger(t). \tag{231}$$

Let us start with the case $\ell = 0$. Zero photons will be emitted in the time interval $[0, t]$ if and only if zero photons are emitted in every small interval $dt$. Hence

$$\hat{K}_0(t) = \left[\hat{K}_0(dt)\right]^{t/dt} = e^{-\frac{\kappa}{2}\hat{n}t}, \tag{232}$$

where $\hat{n} = a^\dagger a$ is the photon number operator.

With this established we can now find the Kraus map for the case where precisely one photon is emitted in a particular intermediate time window $[\tau, \tau + d\tau]$

$$\hat{K}_0(t-\tau)\hat{K}_1(d\tau)\hat{K}_0(\tau)\hat{\rho}(0)\hat{K}_0^\dagger(\tau)\hat{K}_1^\dagger(d\tau)\hat{K}_0^\dagger(t-\tau)$$
$$= \kappa\, d\tau\, e^{-\frac{\kappa}{2}\hat{n}(t-\tau)} a e^{-\frac{\kappa}{2}\hat{n}\tau}\hat{\rho}(0)e^{-\frac{\kappa}{2}\hat{n}\tau} a^\dagger e^{-\frac{\kappa}{2}\hat{n}(t-\tau)}$$
$$= \kappa\, d\tau\, e^{-\kappa\tau} e^{-\frac{\kappa}{2}\hat{n}t} a\hat{\rho}(0)a^\dagger e^{-\frac{\kappa}{2}\hat{n}t}, \tag{233}$$

where in deriving the last equality we have used the simple commutation relation

$$a e^{-\frac{\kappa}{2}\hat{n}\tau} = e^{-\frac{\kappa}{2}(\hat{n}+1)\tau} a. \tag{234}$$

Notice from eqn (233) that the time of the photon loss must be the same for the operators acting on the left of $\hat{\rho}(0)$ and for adjoint operators acting on the right of $\hat{\rho}(0)$. If we sum over all possible times at which the photon loss can occur, we arrive at a rather simple expression for the action of $\hat{K}_1(t)$

$$\hat{K}_1(t)\hat{\rho}(0)\hat{K}_1^\dagger(t) = \kappa \int_0^t d\tau\, e^{-\kappa\tau} e^{-\frac{\kappa}{2}\hat{n}t} a\hat{\rho}(0)a^\dagger e^{-\frac{\kappa}{2}\hat{n}t} \tag{235}$$

$$= \left[1 - e^{-\kappa t}\right] e^{-\frac{\kappa}{2}\hat{n}t} a\hat{\rho}(0)a^\dagger e^{-\frac{\kappa}{2}\hat{n}t}, \tag{236}$$

from which we obtain the exact expression for $\hat{K}_1(t)$

$$\hat{K}_1(t) = \sqrt{[1 - e^{-\kappa t}]} e^{-\frac{\kappa}{2}\hat{n}t} a. \tag{237}$$

Remarkably, this expression tells us that it does not really matter at what point during the time interval the photon was lost. We can commute the destruction operator through to the right so that the loss appears to be at the very beginning of the interval and is followed by an interval $t$ with zero jumps. This is a special property of the harmonic oscillator. In the more general case, because the frequency of an anharmonic oscillator depends on the number of quanta in the oscillator, the phase accumulated during the time interval $[0, t]$ would depend on exactly when the photon was lost. If we do not have the information on when the event occurred, we have to sum over (trace out) all those different times and the oscillator will be dephased.

It is now straightforward to extend the above derivation to the general case of losing $\ell$ photons and obtain [84, 102]

$$\hat{K}_\ell(t) = \sqrt{\frac{[1 - e^{-\kappa t}]^\ell}{\ell!}} e^{-\frac{\kappa}{2}\hat{n}t} a^\ell . \tag{238}$$

With eqns (238) and (231), we now have the complete solution of the Lindblad master equation for the undriven damped harmonic oscillator.

---

**Exercise 33**

Derive eqn (238) for general $\ell$ via recursion from the result for $\ell - 1$.

**Exercise 34**

Explicitly verify that the Kraus operators given in eqn (238) satisfy the trace-preserving condition given in eqn (103). Hint: evaluate eqn (103) in the Fock state basis.

---

We remind the reader that the Kraus representation in eqn (231) is not unique. We have chosen to organize it according to the number of excitations $\ell$ that are lost, but any unitary rotation amongst the $\hat{K}_\ell(t)$ would also be a correct solution of the master equation. The particular representation we have chosen can be thought of as representing an experiment in which we measure the number of photons that have leaked into the environment, and then ignore the measurement result. A unitary rotation among the corresponding Kraus operators could be made that corresponds to (say) monitoring the signals leaking out of the cavity using a homodyne or heterodyne detector (and then again ignoring the result). Since we don't know the measurement result, we end up with the same Lindblad master equation for the oscillator because the master equation is derived by tracing out (i.e. averaging over) the environment and the partial trace over the environment is independent of the basis in which it is performed.

If we actually are monitoring the environment with a particular type of measurement (photon counting, homodyne, etc.), then we can follow the so-called 'quantum trajectory' of the system conditioned on the sequence of measurement results (by applying only the Kraus operators corresponding to the particular measurement results, e.g. the sequence of photons emitted into the environment at specific times). This gives us the state of the system conditioned on our knowledge of the measurement results. If we have perfect measurements that fully determine the state of the environment (in some basis), then it can no longer be entangled with the oscillator and both the oscillator and environment are left in known pure states.

In deriving the Lindblad equation, we have implicitly assumed that we are working in the interaction picture where the only term in the Hamiltonian is the weak coupling to the bath. If there are external drives or other Hamiltonian terms acting on the oscillator, then in the interaction picture we simply need to add the usual commutator term

$$\frac{d\hat{\rho}}{dt} = -i\left[\hat{V}(t), \hat{\rho}(t)\right] + \kappa \mathcal{D}[a]\hat{\rho} . \tag{239}$$

Notice the very important sign difference for the commutator term relative to the Heisenberg (or interaction picture) equation of motion of an ordinary operator. This is because, even though the density matrix is technically an operator, it is representing the state of the system, not the operators that act upon that state.

The case of a driven damped oscillator is very important and useful to understand. We will treat it using the method of displaced frames following the discussion presented in Ref. [103]. Within the rotating wave approximation, the interaction picture representation of an external drive at the resonator frequency whose (envelope amplitude) is $\epsilon(t)$ is given by

$$\hat{V}(t) = \epsilon(t)a^\dagger + \epsilon^*(t)a. \tag{240}$$

The effect of such an external drive is to displace the oscillator in phase space, changing its position and momentum. It is therefore useful to perform a (time-dependent) frame change (displacement transformation) to try to eliminate the drive. The transformed density matrix is

$$\tilde{\rho}(t) = D^\dagger(\alpha)\hat{\rho}(t)D(\alpha), \tag{241}$$

where

$$D(\alpha) = e^{\alpha a^\dagger - \alpha^* a} = e^{\alpha a^\dagger}e^{-\alpha^* a}e^{-\frac{1}{2}|\alpha|^2}, \tag{242}$$

is the phase-space displacement operator (discussed further in Sec. 8.1). Using

$$D^\dagger(\alpha)aD(\alpha) = a + \alpha, \tag{243}$$

it follows that the time derivative of the transformed density matrix obeys

$$\frac{d\tilde{\rho}}{dt} = \left[\dot{\alpha}^*a - \dot{\alpha}a^\dagger, \tilde{\rho}\right] - i\left[\epsilon(t)a^\dagger + \epsilon^*a, \tilde{\rho}\right] + \kappa\mathcal{D}[a + \alpha]\tilde{\rho}. \tag{244}$$

We can write the dissipation superoperator term as

$$\kappa\mathcal{D}[a + \alpha]\tilde{\rho} = \kappa\mathcal{D}[a]\tilde{\rho} - i\left[i\frac{\kappa}{2}(\alpha^*a - \alpha a^\dagger), \tilde{\rho}\right], \tag{245}$$

which shows that the displacement $\alpha(t)$ produces a Hermitian driving term that can be chosen to cancel the external driving term by requiring that $\alpha(t)$ obey

$$-i\dot{\alpha} + \epsilon - i\frac{\kappa}{2}\alpha = 0. \tag{246}$$

From this it follows that in the presence of the drive $\epsilon(t)$, the displaced density matrix $\tilde{\rho}$ obeys the undriven master equation given in Eq. (227). Ref. [103] provides further details on the driven damped oscillator and shows that if it suffers from dephasing, this is worsened in the presence of the drive.

# 7 Quantum Error Correction for Bosonic Modes

## 7.1 Introduction

Now that we have the Kraus representation for the amplitude damping channel of a harmonic oscillator well in hand, we are ready to begin our study of bosonic error correcting codes. Mazyar Mirrahimi's lectures cover Schrödinger cat code [38,91–93] and Barbara Terhal's lectures address the Gottesman-Kitaev-Preskill (GKP) code [58,94–99]. In this section I will address a very simple family of codes, the binomial codes [84], also known as 'kitten' codes because

Table 5: Comparison of qubit and bosonic codes for amplitude damping. For the bosonic binomial ('kitten') code, the no-jump Kraus operator is $\hat{K}_0 = \sqrt{\hat{I} - \kappa \, dt \, \hat{n}}$, where $\hat{n} = a^\dagger a$ is the oscillator excitation (boson) number. For the 4-qubit code, $\hat{K}_0 = \sqrt{\hat{I} - \gamma \, dt \, \hat{n}}$, where $\hat{n} = \sum_{j=1}^{4} \sigma_j^+ \sigma_j^-$, is the total qubit excitation number. Photon number parity is not the sole stabilizer needed to define the binomial code space, but it is the only one that needs to be measured to do quantum error correction of amplitude damping.

| | 4-qubit Code | Bosonic Kitten Code |
|---|:---:|:---:|
| codeword $\lvert 1_L \rangle$ | $\frac{1}{\sqrt{2}}(\lvert 0000 \rangle + \lvert 1111 \rangle)$ | $\frac{1}{\sqrt{2}}(\lvert 0 \rangle + \lvert 4 \rangle)$ |
| codeword $\lvert 0_L \rangle$ | $\frac{1}{\sqrt{2}}(\lvert 1100 \rangle + \lvert 0011 \rangle)$ | $\lvert 2 \rangle$ |
| Mean excitation number $\bar{n}$ | 2 | 2 |
| Hilbert space dimension $D$ | $2^4 = 16$ | $\{0,1,2,3,4\} = 5$ |
| $N_{\text{error}}$ | $\{\hat{K}_0, \sigma_1^-, \sigma_2^-, \sigma_3^-, \sigma_4^-\} = 5$ | $\{\hat{K}_0, a\} = 2$ |
| Stabilizers | $S_1 = Z_1 Z_2, \ S_2 = Z_3 Z_4, \ S_3 = X_1 X_2 X_3 X_4$ | $\hat{\Pi} = (-1)^{a^\dagger a}$ |
| Number of Stabilizers $N_{\text{Stab}}$ | 3 | 1 |
| Approximate QEC? | Yes, 1st order in $\gamma t$ | Yes, 1st order in $\kappa t$ |

the codewords are quite small but resemble those of the Schrödinger cat code. The relative performance of all these codes under various circumstances is thoroughly reviewed in [85].

Another interesting topic is multi-mode codes which employ bosonic states entangled between two different resonators (or flying modes). The simplest member of the family of single-mode kitten codes, bears an interesting resemblance to an early two-mode code designed for amplitude damping [104]. A novel two-mode code designed for superconducting circuits (the 'pair-cat' code) has been proposed in [105].

## 7.2 Binomial Codes

As discussed previously in Sec. 4.4, the smallest qubit code that can exactly correct an arbitrary single-qubit error has $N = 5$ qubits [40–42], however there exists a four-qubit code which can approximately (i.e., to lowest order in the error rate) correct amplitude damping errors. The four qubits of this code have a Hilbert space of dimension $2^4 = 16$.

In this section we describe the simplest member of the family of binomial codes for a single bosonic mode. Like the four-qubit code, it is able to correct amplitude damping to lowest order in the error rate. However it requires only 5 states in the oscillator Hilbert space and, unlike the qubit amplitude damping code, it has a simple recovery operation based on measurement of only a single stabilizer, the photon number parity. Furthermore unlike the four-qubit code, the bosonic code has already been demonstrated experimentally to (very nearly) reach [106] and (more recently) to even exceed [107] the break-even point for QEC. A CZ entangling gate has been demonstrated between two logically encoded qubits [108] as well as remote state transfer [109], a teleported CNOT gate [110] and an exponential SWAP gate [111]. The two codes are compared in Table 5 where we see that the qubit code is essentially a 'unary'[18] representation of the binomial code.

---

[18]See https://en.wikipedia.org/wiki/Unary_numeral_system.

The simplest member of the binomial code family consists of the following two code logical words expressed in the Fock basis

$$|0_{\mathrm{L}}\rangle = \frac{1}{\sqrt{2}}\left[|0\rangle + |4\rangle\right], \tag{247}$$

$$|1_{\mathrm{L}}\rangle = |2\rangle. \tag{248}$$

We see immediately that, like the four-qubit amplitude damping code, there are on average two excitations in both codewords. Both codewords are +1 eigenstates of photon number parity

$$\hat{P} = e^{i\pi a^\dagger a}, \tag{249}$$

which anticommutes with the loss operator

$$\hat{P}a = -a\hat{P}. \tag{250}$$

Thus the parity operator is the stabilizer for the code and tells us whether or not an error (photon loss) has occurred. Of course, if we wait too long between error syndrome measurements, two jumps might occur which would not flip the parity and lead to an unrecognized and unrecoverable error.

Because there is only a single mode instead of four qubits, the Kraus representation for the amplitude damping is simpler. From eqn (232) and eqn (237) we have, to lowest order in the amplitude damping rate,

$$\hat{K}_0(t) = [\hat{K}_0(dt)]^{t/dt} = e^{-\frac{\kappa}{2}\hat{n}t} \approx 1 - \frac{\kappa t}{2}\hat{n}, \tag{251}$$

$$\hat{K}_1(t) = \sqrt{[1 - e^{-\kappa t}]}e^{-\frac{\kappa}{2}\hat{n}t}a \approx \sqrt{\kappa t}\,a. \tag{252}$$

Let us begin our analysis by considering the effect of the jump evolution $\hat{K}_1$ on each codeword

$$\hat{K}_1(t)|0_{\mathrm{L}}\rangle = \sqrt{2\kappa t}|E_0\rangle, \tag{253}$$

$$\hat{K}_1(t)|1_{\mathrm{L}}\rangle = \sqrt{2\kappa t}|E_1\rangle, \tag{254}$$

where the normalized error words are

$$|E_0\rangle = |3\rangle, \tag{255}$$

$$|E_1\rangle = |1\rangle. \tag{256}$$

Similarly the effect of the no-jump evolution on each codeword is

$$\hat{K}_0(t)|0_{\mathrm{L}}\rangle = \frac{|0\rangle + (1 - 2\kappa t)|4\rangle}{\sqrt{2}} \tag{257}$$

$$\approx \sqrt{1 - 2\kappa t}\left[|0_{\mathrm{L}}\rangle + \kappa t|E_2\rangle\right], \tag{258}$$

$$\hat{K}_0(t)|1_{\mathrm{L}}\rangle = (1 - \kappa t)|1_{\mathrm{L}}\rangle \tag{259}$$

$$\approx \sqrt{1 - 2\kappa t}|1_{\mathrm{L}}\rangle, \tag{260}$$

where the third error state is

$$|E_2\rangle \equiv \frac{|0\rangle - |4\rangle}{\sqrt{2}}. \tag{261}$$

Combining these results yields

$$\langle 0_L|\hat{K}_0^\dagger(t)K_0(t)|0_L\rangle \approx \frac{1}{2}[1+(1-2\kappa t)^2] \approx 1-2\kappa t\,, \tag{262}$$

$$\langle 1_L|\hat{K}_0^\dagger(t)K_0(t)|1_L\rangle \approx (1-\kappa t)^2 \approx 1-2\kappa t\,, \tag{263}$$

$$\langle 0_L|\hat{K}_0^\dagger(t)K_0(t)|1_L\rangle = 0\,, \tag{264}$$

$$\langle 1_L|\hat{K}_0^\dagger(t)K_0(t)|0_L\rangle = 0\,, \tag{265}$$

$$\langle 0_L|\hat{K}_1^\dagger(t)K_1(t)|0_L\rangle \approx 2\kappa t\,, \tag{266}$$

$$\langle 1_L|\hat{K}_1^\dagger(t)K_1(t)|1_L\rangle \approx 2\kappa t\,, \tag{267}$$

$$\langle 0_L|\hat{K}_1^\dagger(t)K_1(t)|1_L\rangle = 0\,, \tag{268}$$

$$\langle 1_L|\hat{K}_1^\dagger(t)K_1(t)|0_L\rangle = 0\,. \tag{269}$$

In addition, parity symmetry guarantees that for any pair of logical states $\sigma_L, \sigma_L'$

$$\langle \sigma_L'|K_0^\dagger(t)K_1(t)|\sigma_L\rangle = 0\,, \tag{270}$$

$$\langle \sigma_L'|K_1^\dagger(t)K_0(t)|\sigma_L\rangle = 0\,. \tag{271}$$

Thus eqns (262-271) tell us that the Knill-Laflamme conditions in eqn (121) are fulfilled to first order in $\kappa t$. Thus an ideal recovery operation will be able to eliminate errors to first order in $\kappa t$ and thus reduce to the rate of unrecoverable errors to $\mathcal{O}((\kappa t)^2)$. The more frequently we correct the errors (using an ideal recovery operation), the greater is the QEC gain. This is the essence of so-called 'approximate quantum error correction.'

Since this is a stabilizer code, the error correction procedure consists of frequently measuring the stabilizer (photon number parity) to detect photon loss. If the parity jumps, we must apply a state transfer operation that maps

$$|E_0\rangle = |3\rangle \rightarrow |0_L\rangle\,, \tag{272}$$

$$|E_1\rangle = |1\rangle \rightarrow |1_L\rangle\,. \tag{273}$$

Since the two error states are orthogonal and the two codewords are orthogonal, this can be effected via a unitary operation (that by definition preserves the inner products of states).

If the parity does not jump, then we must apply a different recovery operation that maps

$$|1_L\rangle \rightarrow |1_L\rangle\,, \tag{274}$$

and undoes the small rotation undergone by $|0_L\rangle$

$$\cos\frac{\theta}{2}|0_L\rangle + \sin\frac{\theta}{2}|E_2\rangle \rightarrow |0_L\rangle\,, \tag{275}$$

where from eqn (258), $\sin\frac{\theta}{2} \approx \kappa t$ and to first order, $\cos\frac{\theta}{2} \approx 1$. Clearly, to first-order in $\kappa t$, this transformation is also unitary. These results establish the unitary recovery operations to be applied conditioned on the error syndrome measurement. As mentioned earlier, this QEC procedure was recently successfully carried out [106, 107], exceeding the break-even condition. Furthermore, a CZ gate between two logically encoded bosonic qubits has been demonstrated [108] using this kitten code. In addition, a teleported CNOT gate [110] and an exponential SWAP gate [111] have been demonstrated. A CNOT gate has also been realized between two cavities with one of them encoded in the kitten code [112] and a remote state transfer between cavities has been demonstrated [109, 113].

One may ask how we define break-even for a continuous variable code. Recall that for qubit codes, break-even (for memory operations) is defined as the logical qubit having a lifetime

matching that of the best single physical qubit among those comprising the logical qubit. It is convenient to define break-even for a bosonic code as the logical lifetime matching that of the longest-lived uncorrectable encoding of information in the oscillator. For amplitude damping this is the encoding with the fewest number of photons on average, namely the 0, 1 Fock state encoding

$$|0_{\mathrm{L}}\rangle = |n = 0\rangle, \tag{276}$$

$$|1_{\mathrm{L}}\rangle = |n = 1\rangle. \tag{277}$$

Clearly this encoding cannot correct photon loss since that maps all states on the logical Bloch sphere to $|0_{\mathrm{L}}\rangle$. However it has the fewest photons. Averaging over the logical Bloch sphere, the mean photon number is only $1/2$, so the average photon loss rate of the Kitten code is higher by a factor of four. This factor is the overhead incurred by the code in order to be able to correct errors and is analogous to the overhead we saw earlier for all qubit codes. Indeed, the four-qubit amplitude damping code faces exactly the same overhead factor of four to reach break-even.

It is interesting to note that the no-jump evolution does not leave the state invariant if it is not an eigenstate of the photon number operator. In states where the photon number is uncertain, the probability amplitude for the components with large photon number decays, while the probability amplitude for small photon numbers increases (when the state is normalized). This is essentially a quantum version of a Bayesian update of our estimate of the probability distribution of photon numbers based on the new information that no jump occurred.

Our simple kitten code bears a resemblance to an interesting two-mode bosonic code developed some time ago by Chuang et al. [104]

$$|0_{\mathrm{L}}\rangle = \frac{|04\rangle + |40\rangle}{\sqrt{2}}, \tag{278}$$

$$|1_{\mathrm{L}}\rangle = |22\rangle. \tag{279}$$

This code is more complex to create and manipulate because it involves entangling two separate modes. Because there are two cavities, we need to measure the parity of each to monitor for jump events in each one. Another disadvantage relative to the kitten code is that, since the mean photon number is twice as large, the rate of photon loss errors is doubled. However a major advantage of this code is the novel feature that both codewords have precisely the same total photon number, $n = 4$ (as opposed to merely the same *average* photon number as in the binomial code). Hence, provided both cavities have exactly the same decay rate, both codewords remain invariant under no-jump evolution. Since no-jump is the most probable result of any parity measurement, it is a considerable advantage not to have to apply frequent corrections for no-jump evolution. We need only apply jump corrections to the appropriate cavity when there is a parity jump in that cavity.

Experimentally, cavities are typically in the 'over-coupled' regime where their decay rates are determined by intentional coupling to external drive ports rather than by intrinsic internal losses. These intentional couplings are manually adjustable but not *in operando* unless complex piezoelectric positioning systems are installed to precisely adjust the amount by which the coupling pin from the drive port extends into the cavity. Hence it may be necessary to accept that, without special efforts, there will generically be differences in the decay rates of the cavities. However, it is straightforward to show that the no-jump dephasing back action scales quadratically with the difference in decay rates rather than linearly. Thus there can still be substantial benefits even if the decay rates differ by as much as (say) 30%.

The single-mode kitten code we have described above is able to correct a single photon loss error. This code is the simplest member of an infinite family of codes [84] which are capable

of correcting an arbitrary specified number of $\mathcal{L}$ photon loss, $G$ gain and $D$ dephasing errors from the error set

$$\left\{\hat{I}; a, a^2, \ldots, a^{\mathcal{L}}; a^\dagger, \ldots, (a^\dagger)^G; \hat{n}, \hat{n}^2, \ldots, \hat{n}^D\right\}. \tag{280}$$

The logical codewords have the form

$$\frac{1}{\sqrt{2^N}} \sum_{p=0}^{N+1} \sqrt{\left(\begin{array}{c} N+1 \\ p \end{array}\right)} |p(S+1)\rangle, \tag{281}$$

where the sum is over $p$ even for the $|0_{\mathrm{L}}\rangle$ logical state and $p$ odd for the $|1_{\mathrm{L}}\rangle$ logical state, the 'spacing' $S = \mathcal{L} + G$, and $N = \max\{\mathcal{L}, G, 2D\}$. Because the Fock state amplitudes are given by square roots of binomial coefficients, these are referred to as binomial codes. Further details can be found in [84]. To date, only the smallest binomical 'kitten' code ($\mathcal{L} = 1, G = 0, D = 0$) described above has been realized experimentally [106–109].

---

**Exercise 35**

Analyze the recovery operation for the two-mode code described above, for the case where the two cavities have different decay rates $\kappa_1 \neq \kappa_2$. Discuss how one might choose to make the no-jump corrections less often than the jump corrections for the case $\kappa_1 \approx \kappa_2$.

---

## 8 Gottesman-Kitaev-Preskill Bosonic Codes

In 2001, Gottesman, Kitaev and Preskill proposed a novel idea for quantum error correcting codes based on lattice ('grid') states in the phase space of an oscillator [58,94–96]. At the time, physical implementation appeared to be nearly impossible, but recent dramatic experimental progress has in fact realized QEC close to the break-even point in both ion traps [99] and circuit QED systems [98]. Before exploring these codes we will take a short mathematical detour to understand the geometry of phase space for a quantum oscillator.

### 8.1 Geometry of Phase Space: Non-Commuting Translations and Boosts

The density distribution $\rho(\vec{r}, \vec{p})$ of a gas in phase space plays an essential role in classical statistical mechanics. Things are trickier in quantum mechanics because the position and momentum operators do not commute. This in turn implies that the geometry of quantum phase space is such that adiabatic transport around a closed loop in phase space induces a geometric phase proportional to the area of the enclosed loop. To see this, consider a one-dimensional system with wave function $\psi(x)$. Translation of the momentum (a 'boost') is effected via

$$\mathcal{D}_p(\Delta_p)\psi(x) = e^{i\Delta_p x/\hbar}\psi(x), \tag{282}$$

while translation of the position is effected via

$$\mathcal{D}_x(\Delta_x)\psi(x) = \psi(x - \Delta_x) = e^{-i\Delta_x \hat{p}/\hbar}\psi(x), \tag{283}$$

where $\hat{p} = -i\hbar\frac{d}{dx}$ is the momentum operator (and generator of displacements).

Now consider a set of translations and boosts which moves the system around the closed loop in phase space illustrated in Fig. 21. The first translation by $+\Delta_x$ yields

$$\mathcal{D}_x(+\Delta_x)\psi(x) = \psi(x - \Delta_x). \tag{284}$$

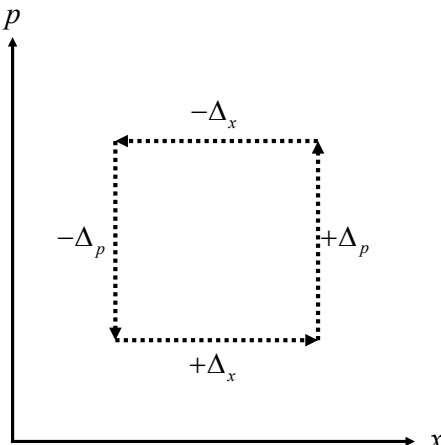

Figure 21: A closed (right-handed) loop in phase space enclosing area $A = \Delta_x \Delta_p$ which yields a geometric phase $e^{+iA/\hbar}$.

The momentum boost $+\Delta_p$ yields

$$\mathcal{D}_p(+\Delta_p)\mathcal{D}_x(+\Delta_x)\psi(x) = e^{+i\Delta_p x/\hbar}\psi(x - \Delta_x). \tag{285}$$

The second translation by $-\Delta_x$ yields

$$\mathcal{D}_x(-\Delta_x)\mathcal{D}_p(+\Delta_p)\mathcal{D}_x(+\Delta_x)\psi(x) = e^{+i\Delta_p(x+\Delta_x)/\hbar}\psi(x). \tag{286}$$

The final momentum boost by $-\Delta_p$ returns the system to its original state except for a geometric phase which depends only on the (oriented) area of the loop

$$\mathcal{D}_p(-\Delta_p)\mathcal{D}_x(-\Delta_x)\mathcal{D}_p(+\Delta_p)\mathcal{D}_x(+\Delta_x)\psi(x) = e^{+i\Delta_p \Delta_x/\hbar}\psi(x). \tag{287}$$

Finally we note that general translations along arbitrary directions defined by the vector $\mathbf{V} = (\Delta_x, \Delta_p)$ are effected via the operator

$$\hat{T}(\mathbf{V}) = e^{+i(\Delta_p \hat{x} - \Delta_x \hat{p})/\hbar} = e^{+i\mathbf{R}^{\mathrm{T}}\Omega\mathbf{V}/\hbar}, \tag{288}$$

where $\mathbf{R} \equiv (\hat{x}, \hat{p})$ is a vector whose components are the position and momentum operators and $\Omega$ is the symplectic form

$$\Omega = \begin{pmatrix} 0 & +1 \\ -1 & 0 \end{pmatrix}. \tag{289}$$

Using the Feynman disentangling theorem, this can be decomposed into

$$\hat{T}(\mathbf{V}) = \mathcal{D}_p(\Delta_p)\mathcal{D}_x(\Delta_x)e^{-\frac{i}{2}\Delta_x\Delta_p/\hbar}, \tag{290}$$

from which it follows that if the second half of the square trajectory in Fig. 21 is replaced by a single displacement, $\mathcal{D}_{-\vec{V}} = \mathcal{D}_{\vec{V}}^{\dagger}$, along the diagonal from the upper right to lower left corners of the square, the resulting triangular path has geometric phase $e^{+\frac{i}{2}\Delta_x\Delta_p/\hbar}$, consistent with the fact that it has half the area of the square trajectory.

More generally, a fact which will be useful later is that

$$\hat{T}(\mathbf{U})\hat{T}(\mathbf{V}) = \hat{T}(\mathbf{V})\hat{T}(\mathbf{U})e^{i\frac{2\pi}{\hbar}\mathbf{V}^{\mathrm{T}}\Omega\mathbf{U}} = \hat{T}(\mathbf{U}+\mathbf{V})e^{i\frac{\pi}{\hbar}\mathbf{V}^{\mathrm{T}}\Omega\mathbf{U}}, \tag{291}$$

which implies that two translations commute if and only if their symplectic norm is an integer multiple of Planck's constant, $h$

$$\frac{1}{h}\mathbf{V}^{\mathrm{T}}\Omega\mathbf{U} \in \mathbb{Z}. \tag{292}$$

Equivalently, the two translations define a parallelogram whose area is an integer multiple of $h$.

Direct experimental measurement of the geometric phase resulting from closed trajectories can be obtained by coupling a qubit to an oscillator and executing conditional displacements which can be realized in both trapped ions [97, 99, 114–119] and in circuit QED [98]. These conditional displacement operators are a generalization of eqn (288) that take advantage of the fact that the forces driving the oscillator can be made spin dependent

$$\hat{T}_c(\mathbf{V}) = \hat{T}(Z\mathbf{V}), \tag{293}$$

where $Z$ is the Pauli spin operator of the ancilla qubit. The protocol begins by placing the qubit in a superposition state and the oscillator in its ground state

$$|\psi\rangle = \frac{1}{\sqrt{2}}\{|\uparrow\rangle + |\downarrow\rangle\}|0\rangle. \tag{294}$$

Then the oscillator is moved around a clockwise (counterclockwise) loop in phase space if the spin is up (down). The spin-dependent 'phase kickback' from this unitary rotates the spin around the $z$ axis by an angle given by the difference in geometric phases of the two phase space loops

$$|\psi'\rangle = \frac{1}{\sqrt{2}}\left\{e^{i\theta_\uparrow}|\uparrow\rangle + e^{i\theta_\downarrow}|\downarrow\rangle\right\}|0\rangle. \tag{295}$$

Observing the 'Ramsey interference' fringes caused by this rotation of the ancilla qubit through angle $\theta = \theta_\downarrow - \theta_\uparrow$ is routinely used by experimentalists to precisely calibrate the relation between applied electromagnetic drive on the oscillator and the resulting displacement amplitude [97–99, 119].

## 8.2 GKP Codes: Lattices in Phase Space

Now that we understand the geometry of phase space through the commutation relations between different translations, we are ready to study the GKP codes. The GKP code space of the logical qubit is defined to be the +1 eigenspace of two translations $\hat{S}_1 = \hat{T}(\mathbf{U})$ and $\hat{S}_2 = \hat{T}(\mathbf{V})$ that obey

$$\frac{1}{h}\mathbf{V}^{\mathrm{T}}\Omega\mathbf{U} = 2, \tag{296}$$

and therefore the code space contains two states. Based on the previous discussion of codes involving $N$ physical qubits, you might think that since the oscillator Hilbert space contains an infinite number of states that you would need an infinite number of stabilizers. This is not the case however. The multi-qubit Pauli operator stabilizers for qubit codes have eigenvalues $\pm 1$ and thus provide only 1 bit of information when measured. The translations used as stabilizers for the bosonic codes have a continuum of eigenvalues lying on the unit circle in the complex plane. The statement that the code space has stabilizer eigenvalues precisely equal to +1 therefore represents far more powerful constraints.

To see how the GKP code works, let us specialize to the simplest case of a square lattice in phase space generated by the two stabilizers (displacements)

$$S_p = e^{+i2\sqrt{\pi}\hat{x}}, \tag{297}$$

$$S_x = e^{-i2\sqrt{\pi}\hat{p}}, \tag{298}$$

where from henceforth we are using dimensionless units with $\hbar = 1$. These stabilizers commute

$$S_p S_q = e^{i4\pi} S_q S_p, \tag{299}$$

and they stabilize the code space spanned by the logical codeword states

$$\Psi_0(x) = \sum_{n=-\infty}^{+\infty} \delta(x - (2n)\sqrt{\pi}), \tag{300}$$

$$\Psi_1(x) = \sum_{n=-\infty}^{+\infty} \delta(x - (2n+1)\sqrt{\pi}). \tag{301}$$

These 'picket fence' states are infinitely squeezed, unnormalizable, and have infinite energy. We will ignore these inconvenient truths for the moment but return to them later. For now, we note that they are also picket fence states in the momentum basis and their Wigner functions have support on a square lattice in phase space. Despite the fact that position and momentum do not commute, the points of the square lattice are sharply defined (infinitely squeezed in both position *and* momentum). This does not violate the Heisenberg principle because both position and momentum remain uncertain modulo the lattice constants.

Remarkably, within this logical subspace, the logical Pauli operators become simple phase space translations as illustrated in Fig. 22 where we see that the 'unit cell' has area $4\pi$ (or, restoring the units, $4\pi\hbar = 2h$) and therefore holds two quantum states. Within the logical subspace, the Pauli $X_L$ and $Z_L$ operators are displacements by half of the respective lattice constants implying that

$$X_L^2 = S_x = \hat{I}, \tag{302}$$

$$Z_L^2 = S_p = \hat{I}, \tag{303}$$

which is consistent with the properties of the Pauli matrices since within the logical space, $S_x$ and $S_y$ are each (effectively) the identity. Similarly

$$X_L = X_L^\dagger S_x = X^\dagger, \tag{304}$$

$$Z_L = Z_L^\dagger S_p = Z^\dagger, \tag{305}$$

as expected. The area of the four small squares being $\pi$ implies that

$$Z_L X_L = -X_L Z_L. \tag{306}$$

Furthermore, the shaded triangle has area $\pi/2$ implying that

$$Y_L = Y_L^\dagger = e^{i\frac{\pi}{2}} X_L Z_L = i X_L Z_L. \tag{307}$$

Thus, within the logical space, these translations form a precise representation of the Pauli group. More generally, one of the advantages of GKP codes is that Clifford group operations correspond to (relatively) simple Gaussian operations on the bosonic modes (e.g. squeezing) that can be executed using quadratic Hamiltonians. For example, the CNOT gate corresponds to the so-called SUM gate

$$\text{CNOT}_{12} = e^{-\frac{i}{\hbar}\hat{x}_1 \hat{p}_2}, \tag{308}$$

which displaces mode 2 by an amount equal to the coordinate of mode 1, $\hat{x}_2 \to \hat{x}_2 + \hat{x}_1$. (Additionally, $\hat{p}_2 \to \hat{p}_2$, $\hat{x}_1 \to \hat{x}_1$, and $\hat{p}_1 \to \hat{p}_1 - \hat{p}_2$). Here mode 1 is the control and mode 2 is the target of the CNOT.

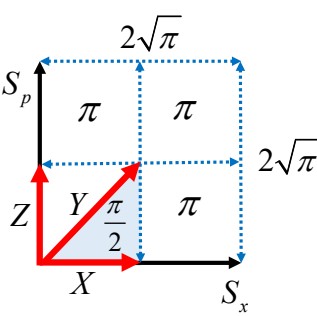

Figure 22: Unit cell of the square lattice defined by the stabilizers $S_x, S_p$. The Pauli operators $X_L$ and $Z_L$ are displacements by half a lattice constant in position and momentum respectively. The Pauli operator $Y_L = iX_L Z_L$ is displacement by half the diagonal. The factor of $i$ in this relation is consistent with the shaded area being $\pi/2$.

---

**Exercise 36**

Show that the generator of the CNOT gate, $\hat{x}_1 \hat{p}_2$, is the sum of beam splitter ($i[a_1 a_2^\dagger - a_2 a_1^\dagger]$) and two-mode squeezing ($i[a_1^\dagger a_2^\dagger - a_2 a_1]$) terms.

A more challenging exercise is to show that the CNOT gate can be written as a 50:50 beam splitter (with a particular phase) followed by single-mode squeezing (of a certain strength) in each mode, and then a second 50:50 beam splitter with the opposite phase.

---

## 8.3 Error Correction with GKP Codes

The GKP codes were originally designed to correct small (i.e., less than half a lattice constant) displacements of the oscillator in phase space. We have recently shown that they are also remarkably effective against more realistic error modes (e.g. amplitude damping) [85]. However analytical understanding is easier for the case of small displacement errors. Because both the errors and the stabilizers are displacements, they do not (generically) commute and these errors change the eigenvalues (phases) of the stabilizers. We can measure the error syndromes (changes in the stabilizer eigenvalues) by using conditional displacements. For example, suppose the erroneous cavity state is $|\Psi\rangle$ and we prepare the ancilla qubit in the superposition state $|+x\rangle$. If we perform the conditional stabilizer displacement

$$U_\lambda = \hat{I} \otimes |\uparrow\rangle\langle\uparrow| + S_\lambda \otimes |\downarrow\rangle\langle\downarrow|, \tag{309}$$

(where $\lambda = x, p$) and then measure the ancilla, the phase kickback gives us the real

$$\langle +x|\langle\Psi|U_\lambda^\dagger X U_\lambda|\Psi\rangle|+x\rangle = \text{Re}\,\langle\Psi|S_\lambda|\Psi\rangle, \tag{310}$$

and imaginary

$$\langle +x|\langle\Psi|U_\lambda^\dagger Y U_\lambda|\Psi\rangle|+x\rangle = \text{Im}\,\langle\Psi|S_\lambda|\Psi\rangle \tag{311}$$

parts of the error syndrome.

Of course with only a single ancilla, one only obtains one bit of information when measuring $X$ or $Y$ in the conditionally displaced state. Quantum phase estimation protocols that would yield more bits of resolution on the value of the stabilizers [94] are not practical at this point in the hardware development. However, the Devoret group has invented a clever scheme [98] that makes small feedback displacements on the quantum state after each repeated one-bit

measurement of each stabilizer. They were able to execute quantum error correction for both the square lattice and the triangular lattice GKP codes, coming very close to the break-even point.

For logical qubits made from multiple physical qubits, we define the break-even point as the logical qubit lifetime reaching the value of the best physical qubit in the set comprising the logical qubit. For bosonic codes we define the break-even point to be where the lifetime of the bosonic code exceeding the lifetime of the longest lived uncorrectable encoding in the oscillator. Superconducting resonators have negligible dephasing errors and are dominated by amplitude damping. In this case, the longest lived encoding uses the smallest possible photon number states, namely the $n = 0, 1$ Fock states. They lose photons at the slowest rate, but of course once a photon is lost the cavity is definitely in $|0\rangle$ and the quantum information stored in the original superpostion state is unrecoverable.

As mentioned previously, the idealized GKP lattice states are unphysical because they are infinitely squeezed and have infinite energy. Real experiments work with lattices of finite extent, typically attempting to create states with a Gaussian envelope of the form

$$|\Phi_\Lambda\rangle \sim e^{-\Lambda \hat{n}}|\Psi\rangle, \tag{312}$$

where $\Lambda$ is a regularizing constant and $|\Psi\rangle$ is a superposition of the ideal codewords in eqns (300-301). This state has a Gaussian envelope because $\hat{n} \sim \hat{x}^2 + \hat{p}^2$.

As the measurement feedback loop discussed above drives the value of the stabilizers more and more precisely towards $+1$, the state starts to approach the ideal GKP state and thus the envelope of the state has to expand. To prevent the mean photon number from running away to infinity, the Devoret group used a 'trim' step in their protocol which cuts off the runaway expansion.

Finite energy approximate GKP states are of course not fully translation invariant and the stabilizer value must suffer quantum fluctuations. Inspired by comments from Mazyar Mirrahimi, Baptiste Royer has recently developed a higher-order error correction scheme based on the exact stabilizer for the finite-energy GKP states [120]. These stabilizers are simply similarity transforms of the ideal stabilizers

$$S_x^\Lambda = e^{-\Lambda \hat{n}} S_x e^{+\Lambda \hat{n}}, \tag{313}$$

$$S_p^\Lambda = e^{-\Lambda \hat{n}} S_p e^{+\Lambda \hat{n}}. \tag{314}$$

These are neither unitary nor Hermitian and so cannot be measured directly. However it is possible to cleverly engineer an autonomous (i.e., measurement free) protocol which effectively yields an engineered dissipation that cools the system into the $+1$ eigenstate of each stabilizer. The same idea was independently discovered by the ion trap group of J. Home who put it into practice in experiment [97, 99]. This new protocol, together with technical improvements in the circuit QED experiments, has recently allowed the Devoret group to substantially exceed the break-even point for the first time [59]. It is still early days and much work remains to be done, but experimental progress on the GKP code is impressive. Ref. [96] provides a review and fault tolerance analysis of the GKP code.

# 9 Lacunae

These notes provide an introduction to the basic aspects of error correction and fault tolerance, but of necessity are not at all comprehensive. They do not address autonomous quantum error correction and error-transparent gate operations on which there has been recent theoretical [121] and experimental [39, 122] progress. These notes briefly discuss concatenation of

classical codes and circuits, but do not explore these topics for the quantum case. The reader may wish to consult [19] for an entré to the extensive literature on this topic as well as on mitigation of leakage errors via teleportation-based error correction.

There have been significant recent advances in the use of noise-biased qubits such as the dissipative-cat [123–127] and the Kerr-cat [128–131] as well as surface codes adapted to the noise-bias [70, 71], empowered by the invention of a bias-preserving topological CNOT gate [37] for the Kerr-cat. There have also been interesting theoretical analyses of GKP bosonic codes forming the first layer of error correction in a surface code [95, 132, 133]. Recent architecture developments to make the dual-rail bosonic encoding first-order fault tolerant look very promising [134].

Finally, important new ideas have been developed at the hardware level of superconducting circuits to improve fault-tolerance against ancilla qubit errors in the control and measurement of bosonic modes [135–139].

The reader is directed to all the references above to learn more about these exciting frontier developments.

## 10 Conclusion

The fact that quantum error correction is possible even in principle is quite surprising and remarkable. We are just entering the era when it is beginning to become possible in practice in the first basic experiments. We have a long way to go, but the community is now beginning to take the first steps to meet the grand challenge of moving quantum systems into the era of fault-tolerant memory and gate operations.

## Acknowledgements

The ideas described here represent the collaborative efforts of many students, postdocs and faculty colleagues who have been members of the Yale quantum information team over the past two decades. The author is especially grateful for the opportunities he has had to collaborate with colleagues Michel Devoret, Leonid Glazman, Liang Jiang, Shruti Puri, and Robert Schoelkopf as well as frequent visitor, Mazyar Mirrahimi. He is also very grateful for numerous informative conversations with Isaac Chuang and Shruti Puri on the topic of classical and quantum fault tolerance. Olivier Pfister and Rafael Alexander kindly supplied their wisdom (and copies of their recent slide decks) on continuous variable quantum information processing. The author would like to thank Christopher Wang, Kevin Smith, Baptiste Royer, Shraddha Singh, Shifan Xu, and Michel Devoret for reading and commenting on drafts of the manuscript.

**Funding information** This work was supported by the Army Research Office through grant ARO W911NF-18-1-0212, by the Yale Center for Research Computing, and by the Yale Quantum Institute.

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
