# Peer review of "Introduction to Quantum Error Correction and Fault Tolerance"

_SciPost Physics Lecture Notes, doi:SciPost Phys. Lect. Notes 70 (2023)_

## Round 3 · Referee Report · Charlotte Boettcher (Referee 1) · 2022-10-9

Strengths

These lecture notes provide a pedagogical introduction to quantum error correction and fault tolerance, using concrete examples to draw relatable analogies, that works well in combination with well-structured layout of the chapters, including instructive figures, tables, and exercises. The use of boxes to highlight certain concepts works well for an educational purpose and is well executed.

Weaknesses

Some main figures and when introducing certain concepts, can be made clearer. Suggestions are specified below under requested changes.
One main recommendation is to include links or references to notes or lectures (that were given at the same school as these notes), which so far are only referenced as ‘lecture/seminar at this school’ in these lecture notes. This does not provide the reader with a concrete way to find references. Links could for instance be provided in the beginning under section ‘Links to video lectures.’

Report

In these notes, the author introduces important concepts in quantum error correction and fault tolerance that was first presented at 2019 Summer School on Quantum Machines in Les Houches.
The notes provide an in-depth introduction to general concepts with a natural crossover from classical examples on error correction to quantum error correction.
The author hinges many of the concepts on relatable examples, thus highlighting the great educational purpose of these notes.
These lecture notes provide an introduction to the unique set of concepts (circuit depth, codewords, etc.) required to understand error correction schemes: repetition code, bosonic code, surface code, toric code, with an appropriate balance of their practicality and applications.
The notes are well structured with a comprehensive historical overview to help set the current stage and challenges that quantum information and quantum computing faces today and finally an outlook that places this work in context with the current state of the field. I therefore recommend this for publication with suggestion of minor changes outlined below.

Requested changes

I propose the following changes and additions to the Lecture notes. Recommendations will make certain concepts stand clearer.

  • On page. 7: I recommend to not define BSC in a figure text. Instead make this definition in the main text to better the flow for the reader.
  • On p. 9, codewords still needs to be specified. Not introduced yet (? Check this)
  • In Figure 8: colors would be instructive. For instance, color the lines from $P_1$, $P_2$, $P_3$ with separate colors so the mapping from each and to their respective $S_1$, $S_2$, $S_3$, ..., etc. are clearer.
  • In Fig. 8, text: clarify how the pooling is done in the figure text. ‘Unique pattern’ is vaguely used here and it’s not clear what is meant by unique pattern.
  • On p. 15; code distance
  • In Fig. 10: ECC defined in the figure text. Make definitions in main text. A general comment that will help the flow.
  • On p. 27 in sentence: … based on (needs a ‘d’)
  • In Fig. 9. Not the best choice of colors. Consider better colors that also works for colorblindness.
  • In Box 5: sentence ‘and it is essential … ‘, misses a word to complete the sentence. Perhaps ‘to’ will fix it.
  • In equation 89: What is g? not defined or explained. If it was defined in the other chapters, (although couldn’t find it), then it would be good to remind the reader here.
  • In Fig. 17. There is barely any labeling. Toffoli, $S_1$, $S_2$, and ‘cold bath’.
  • On p.42: the listing of references is good, but ‘some time’ is too vague when it can be made concrete. Is ‘some time’ = year, months’ time scale. Would be instructive to be specific.
  • On p. 42: NV centers is introduced with its abbreviation. It will be better to define NV = nitrogen vacancy for non-expert readers.
  • On p.42: ‘moving closer’ is vaguely used here, can this be quantified? What are the required lifetimes (roughly)? Are lifetimes the only aspect of ‘moving closer’.
  • On p. 54: above sec 5.3 there is an incomplete parenthesis.
  • In Fig. 18: black lines for ‘plaquette’ and ‘star’ do not provide good contrast. Consider enhancing with for instance color.
  • On p. 68. Reference to lecture and seminar ‘at this school’. Instead reference paper or recorded talk, otherwise should be taken out.
  • On p. 68: there seem to an inconsistency in spelling ‘codewords’. Previous chapters: one word. On p. 68 this is two words: ‘code words’.
  • On p. 69: ‘discussed in lectures at this school’: add reference to notes instead as mentioned above.
  • On p. 71: consider consistency when writing H. Italic in eq. 209, but not in main text, meanwhile X, Z, … are all consistently italic.
  • End of section 7.4. For completeness, it might be instructive to show or quickly derive the driven case (used in many applications). Can it be shown that in a different frame one then obtains the undriven case again?
  • On p. 78: code words -> codewords?
  • On p. 78: misspelling or unitary: … essentially a unary representation..
  • On p. 81, tow-mode bosonic code: How feasible is it to have cavities with the exact same decay rate. Perhaps comment on the realization of this or can it be $\Gamma_1 \approx \Gamma_2$ when making this comparison between single-kitten and two-mode.

---

## Editorial Decision

published